# Enhanced Hsa-miR-181d/p-STAT3 and Hsa-miR-181d/p-STAT5A Ratios Mediate the Anticancer Effect of Garcinol in *STAT3/5A*-Addicted Glioblastoma

**DOI:** 10.3390/cancers11121888

**Published:** 2019-11-27

**Authors:** Heng-Wei Liu, Peter Mingjui Lee, Oluwaseun Adebayo Bamodu, Yu-Kai Su, Iat-Hang Fong, Chi-Tai Yeh, Ming-Hsien Chien, I-Hung Kan, Chien-Min Lin

**Affiliations:** 1Graduate Institute of Clinical Medicine, College of Medicine, Taipei Medical University, Taipei City 11031, Taiwan; henryway0404@hotmail.com (H.-W.L.); yukai.su@gmail.com (Y.-K.S.); ctyeh@s.tmu.edu.tw (C.-T.Y.); mhchien1976@gmail.com (M.-H.C.); 2Department of Neurology, School of Medicine, College of Medicine, Taipei Medical University, Taipei City 11031, Taiwan; 3Division of Neurosurgery, Department of Surgery, Taipei Medical University-Shuang Ho Hospital, New Taipei City 23561, Taiwan; impossiblewasnothing@hotmail.com; 4Taipei Neuroscience Institute, Taipei Medical University, Taipei 11031, Taiwan; 16625@s.tmu.edu.tw; 5Department of Clinical Oncology, College of Medicine, California North State University, Elk Grove, CA 95757, USA; peter100893@gmail.com; 6Department of Hematology and Oncology, Cancer Center, Taipei Medical University—Shuang Ho Hospital, New Taipei City 235, Taiwan; 7Department of Medical Research and Education, Taipei Medical University—Shuang Ho Hospital, New Taipei City 235, Taiwan; 8Department of Medical Laboratory Science and Biotechnology, Yuanpei University of Medical Technology, Hsinchu City 30015, Taiwan

**Keywords:** glioblastoma, GBM, glioma, *STAT3*, *STAT5A*, hsa-miR-181d, microRNA

## Abstract

Background: Glioblastoma (GBM), a malignant grade IV tumor, is the most malignant brain tumor due to its hyper-proliferative and apoptosis-evading characteristics. The signal transducer and activators of transcription (STAT) family genes, including *STAT3* and *STAT5A*, have been indicated to play important roles in GBM progression. Increasing number of reports suggest that garcinol, a polyisoprenylated benzophenone and major bioactive component of *Garcinia indica* contains potent anti-cancer activities. Material and Methods: The present study investigated the anti-GBM effects of garcinol, focusing on the *STAT3/STAT5A* activation, using a combination of bioinformatics, in vitro, and ex vivo assays. Results: Our bioinformatics analysis of The Cancer Genome Atlas (TCGA)–GBM cohort (*n* = 173) showed that *STAT3* and *STAT5A* are preferentially elevated in primary and recurrent GBM, compared to non-tumor brain tissues, and is significantly correlated with reduced overall survival. In support, our immunohistochemical staining of a GBM cohort (*n* = 45) showed an estimated 5.3-fold (*p* < 0.001) elevation in *STAT3* and *STAT5A* protein expression in primary and recurrent GBM versus the non-tumor group. In vitro, garcinol treatment significantly suppressed the proliferative, invasive, and migratory potential of U87MG or GBM8401 cells, dose-dependently. In addition, garcinol anticancer effect significantly attenuated the GBM stem cell-like phenotypes, as reflected by diminished ability of U87MG or GBM8401 to form colonies and tumorspheres and suppressed expression of OCT4 and SOX2. Furthermore, analysis on GBM transcriptome revealed an inverse correlation between the level of *STAT3/5A* and hsa-miR-181d. Garcinol-mediated anti-GBM effects were associated with an increased hsa-miR-181d/*STAT3* and hsa-miR-181d/5A ratio. The results were further verified in vivo using U87MG mouse xenograft model where administration of garcinol significantly inhibited tumor growth. Conclusions: We present evidence of anti-GBM efficacy of garcinol mediated by enhancing the hsa-miR-181d/STAT3 and hsa-miR-181d/5A ratios in GBM cells. Our findings suggest a potential new therapeutic agent for combating aggressive GBM.

## 1. Introduction

Glioblastoma (GBM), a WHO grade IV malignant glioma associated with poor prognosis, is characterized by enhanced cellularity, increased mitotic activity, vascular hyperproliferation, and pseudopallisading necrosis [1,2]. Being highly invasive, GBM cells infiltrate the surrounding brain parenchyma, including the cerebral cortex, cerebellum, brainstem, and the spinal cord [1]. Current treatment options for GBM, namely surgery, chemotherapy, and radiotherapy are characterized by dismal curative success, as evidenced by the limited increase in the median overall survival (OS) from 12 months to 15 months in patients exposed to radiation therapy and temozolomide [2,3]. Symptomatic treatments such as corticosteroids for managing peritumoral edema, antiseizure drugs (ASDs) for reducing seizures, and the anti-angiogenesis monoclonal antibody, Bevacizumab (also known as Avastin), to mitigate brain edema are also used; nevertheless, aside from improving patients’ quality of life (QoL), they confer no survival advantage on patients with GBM [2,4,5]. In addition, contemporary evidence indicates that the one-year-survival rate of patients with de novo GBM is 17–30%, and the two-year survival rate is set at a dismal 3–5% [1]. This dearth of effective anti-GBM therapeutic strategy, against the background of dismal clinical outcomes, necessitates urgent discovery of new actionable molecular targets and/or development of novel efficacious anti-GBM therapy.

The last 20 years has been characterized by accruing evidence of the role of cancer stem cells (CSCs, otherwise known as cancer-initiating cells) in enhanced oncogenicity and metastatic phenotype, resistance to therapy, cancer relapse, and consequently poor clinical outcome in patients with malignancies, including GBM [6,7,8,9,10]. CSCs transcription factors (TFs) are promising anti-cancer therapeutic targets. Contemporary literature is replete with documented association of the signal transducer and activator of transcription (STAT) proteins with enhanced pluripotency, related CSCs-like phenotypes, and cancer progression [11,12,13]. STAT proteins are a family of TFs that facilitate numerous biological processes such as cell proliferation, differentiation, survival, and inflammation [14]. The aberrant expression and/or activities of several members of the STAT protein family, including STAT3 and STAT5A, have been implicated in the initiation, growth, and metastatic dissemination of various cancers, including head and neck, breast and brain cancers [15]. The aberrant expression of STAT3 has been implicated in GBM development and progression and has been suggested to be a master regulator of the mesenchymal cum malignant transformation of gliomas [16]. Li, G et al. demonstrated that lentivirus-mediated silencing STAT3 in GBM cells induces cellular differentiation, indicating its role in keeping an undifferentiated cellular state in malignancy [17]. As with STAT3, STAT5A is a key effector of the Janus Tyrosine Kinase (JAK)/STAT pathway, its phosphorylation being positively correlated with cell invasion and poor prognosis in GBM [18,19,20]. Fan et al. demonstrated that STAT5 signaling drives pro-tumorigenic phenotypes in GBM cells [21]. Thus, in the light of all these evidences suggesting that STAT3 and STAT5A play critical roles in cancer progression, the present study investigated the nature and extent of the involvement of STAT3 and STAT5A in the therapeutic response and disease progression of patients with GBM.

Over the last two decades, there has been increase in the documentation of the critical roles of microRNA (miRNAs/miRs) in the malignantization and progression of many human malignancies, including GBM; by their ability to hamper or facilitate cancer initiation, miRNAs have emerged as therapeutically-relevant actionable biomolecules for anticancer therapy [22,23]. One such miRNA family is miRNA-181, with varied expression of its isoforms (miRNA-181a/b/c/d) being touted as independent predictors of clinical outcome in patients with different cancer types [24]. A notable feature of treatment-resistant GBM is the enzyme O^6^-methylguanin-DNA methyltransferase (MGMT), a DNA-repair protein that removes the alkyl group from the O^6^ position of alkyl groups, and consequently diminish the curative effects of chemotherapeutic agents [25], conversely, the silencing of MGMT through promoter methylation is associated with prolonged OS and disease-free survival (DFS) in patients with GBM [26]. Interestingly, recent studies show that MGMT activity is inversely correlated with the expression of microRNA-181d/miRNA-181d; as higher an expression of miRNA-181d is positively correlated with improved OS in patients with GBM [27]. There is also accruing evidence that the prognostic implications of altered miRNA expression, is connected to their roles in modulation of stemness signaling such as Notch, Hedgehog, and JAK/STAT3 [23,28], and the consequent acquisition of stem cell-like traits by GBM cells; thus, the present study’s rationale for exploring the actionability of miRNA-181d in the context of STAT signaling in GBM.

Due to the accruing adverse reactions to most chemotherapeutic agents, phytochemicals and nutraceuticals have garnered interest as possible safe alternatives or adjuvants in cancer treatment [29]. Building on previous works by our team demonstrating that garcinol inhibits CSCs-like phenotype by suppressing the Wnt/β-catenin/STAT3 signaling axis in human non-small cell lung carcinoma [30], we now examined the probable effect of garcinol on GBM stem cells (GBM-SCs), and the implication of same for sensitivity to conventional chemotherapy and better prognosis. Garcinol, a major bioactive constituent of the fruit *Garcinia indica*, has widely documented antioxidant and anticancer effects, and is chemically similar in structure to the well-known curcumin [29,31]. In fact, Hong et al. [32] demonstrated that garcinol significantly inhibited the growth of the colon cancer cells, and importantly, provided evidence that unlike conventional chemotherapeutics, garcinol preferentially targets cancerous cells; thus, inhibiting cancer growth without adversely affecting the neighboring ‘normal’ non-cancerous cells [32]. In breast cancer, garcinol was shown to inhibit STAT3-NF-kB signaling, resulting in reduced invasiveness, in vitro, and significantly attenuated tumor growth in NOD-SCID mice [33,34]; thus, building a case for the preclinical investigation of the probable anti-GBM effect of garcinol in this stance. This present study, for the first time, to the best of our knowledge, investigated and documents the effect of garcinol on GBM, consistent with current theme-relevant knowledge, especially in the context of the therapeutic effects of garcinol alone or in synergism with conventional anticancer treatment modalities on GBM-SCs through the mediation of STAT3/5A signaling and miRNA-181d.

## 2. Materials and Methods

Ethics approval and consent to participate: The study was approved by the Joint Institutional Review Board (JIRB) of the Taipei Medical University—Shuang Ho Hospital (Approval number: N201903047). Tissue samples from patients with primary and recurrent GBM were obtained from the Taipei Medical University—Shuang Ho Hospital GBM cohort (*n* = 45). The animal study protocol was approved by the Animal Care and User Committee at Taipei Medical University (Affidavit of Approval of Animal Use Protocol # Taipei Medical University- LAC-2017-0512).

### 2.1. Drugs and Chemicals

Garcinol (sc-200891A, HPLC purity ≥95%) and Z-VAD-FMK (sc-3067, HPLC purity ≥95%) purchased from Santa Cruz Biotechnology (Santa Cruz, CA, USA) was dissolved in dimethyl sulfoxide (DMSO) to prepare a 20 mM stock and stored at −20 °C until use. For different assays, the stock was further diluted using cell growth medium as appropriate. Dimethyl sulfoxide (DMSO), served as vehicle and negative control. BD Pharmingen™ PE Annexin V apoptosis detection kit I (#559763) was purchased from BD Biosciences (San Jose, CA, USA). Unless otherwise indicated, all reagents were obtained from Gibco (Thermo Fisher Scientific, Life Technologies, Foster City, CA, USA).

### 2.2. Analyses of Cancer RNAseq Dataset

The Cancer Genome Atlas (TCGA) GDC-TCGA glioblastoma (GBM) cohort (*n* = 173) used for *STAT3* and *STAT5A* gene expression profiling and correlative studies, was accessed, downloaded and analyzed using the University of California Santa Cruz (UCSC) Xena functional genomics explorer platform (https://xenabrowser.net/heatmap/#). The dataset consists of non-tumor (*n* = 5), primary GBM (*n* = 155) and recurrent GBM (*n* = 13).

### 2.3. Cell lines and Primary Culture Cell Culture

The human U-87 MG (ATCC^®^ HTB-14™) (ATCC, Manassas, VA, USA) and GBM8401 GBM cell lines used in the study were purchased from (Bioresource Collection Research Center, Hsinchu, Taiwan). The cell lines were cultured in Gibco DMEM (Cat. No. 11965175, Thermo Fisher Scientific, Inc. Waltham, MA, USA), supplemented with 10% fetal bovine serum (FBS) and 1% penicillin/streptomycin (Invitrogen, Life Technologies, Carlsbad, CA, USA) and incubated in 5% humidified CO2 incubator at 37 °C. The cells were sub-cultured when they reached 80–90% confluency and the media changed every 48–72 h. Patient-derived CD133 + GBM spheres were kindly provided by our collaborator Dr. Alexander T.H. Wu at Taipei Medical University. In brief, the patient-derived GBM cells were first sorted using the established flow cytometric method. Once CD133+ cells were sorted, they were expanded in advanced DMEM/F12 (Gibco) mixed with Neurobasal TM-A medium (Gibco) (1:1) supplemented with B-27 (1×), FGF (20 ng/mL) and EGF (20 ng/mL); culturing under these conditions maintained CD133+ cell population and stemness (as well as TMZ-resistant), the tumor-initiating ability was demonstrated in vivo as described previously [35].

### 2.4. Sulforhodamine B (SRB) Viability Assay

GBM8401 and U87MG cells were seeded in 96-well plates in triplicates at a concentration of 3.5 × 10^3^ cells per well. After 24 h incubation in a 5% CO_2_ humidified incubator at 37 °C, the cells were treated with varying concentrations of 2.5–40 μM garcinol as indicated for 24 h. Thereafter, cells were washed in cold PBS, fixed in 10% trichloroacetic acid (TCA) for 1h, washed with distilled water, and then incubated in 0.4 SRB (*w*/*v*) in 1% acetic acid at room temperature for 1 h. After washing of unbound SRB dye with 1% acetic acid thrice, the plates were air-dried, and bond SRB dye dissolved in 20 mM Tris base under gentle agitation for 5 min and the absorbance was read at 570 nm wavelength in a microplate reader (Molecular Devices, Sunnyvale, CA, USA).

### 2.5. Western Blot Analysis

U87MG and GBM8401 cells pre-treated with or without 2.5 μM or 5 μM of garcinol for 24 h were lysed, and the cellular protein lysates extracted using the Protein Extraction Kit (QIAGEN, Germantown, MD, USA), followed by quantification using the Bradford Protein Assay Kit. 20 μg protein samples were then loaded per lane and protein separated in 10% sodium dodecylsulfate polyacrylamide gel electrophoresis (SDS-PAGE) gels, blots were transferred onto polyvinylidene fluoride (PVDF) membranes, and then non-specific binding blocked in 5% skimmed milk in Tris-buffered saline with Tween20 (TBST) for 1h. Thereafter, blot-bearing PVDF membranes were incubated overnight at 4 °C with the following primary antibodies: p-STAT3 (#9145; 1:1000 dilution), STAT3 (#9132; 1:1000 dilution), p-ERK (#4370; 1:1000), ERK1/2 (#4695; 1:1000), p-Akt1/2/3 (#4060, 1:1000), Akt1/2/3 (#2920, 1:1000), Bax (#5023; 1:1000), and Bcl-xl (#2764; 1:1000) from Cell Signaling Technology (cell signaling, Danvers, MA, USA), p-STAT5 (ab32364; 1:1000), and STAT5 (ab227687; 1:1000 dilution) from Abcam (Abcam, MA, USA) and GAPDH (10494-1-AP; 1:10000) from Proteintech Group (Proteintech, IL, USA). Please see the Appendix A. This was followed by washing membranes thrice in TBST for 5 min each, before incubition with appropriate goat anti-mouse or anti-rabbit horseradish peroxidase (HRP)-linked secondary antibodies for 1h and TBST washing three times for 5 min each. The protein bands were visualized using enhanced chemiluminescence (ECL) detection system (ECL, Amersham Pharmacia Biotech, NJ, USA), and quantified using ImageJ software (https://imagej.nih.gov/ij/).

### 2.6. Immunohistochemical (IHC) Staining

The study was approved by the Joint Institutional Review Board (JIRB) of the Taipei Medical University–Shuang Ho Hospital (Approval number: N201903047). Tissue samples from patients with primary and recurrent GBM were obtained from the Taipei Medical University–Shuang Ho Hospital GBM cohort (*n* = 45). After de-waxing the paraffin-embedded 4 μm tissue sections using xylene for 5 min twice and re-hydrating with 100% ethanol twice for 5 min, 95% ethanol for 5 min, and 80% ethanol for 5 min, 3% hydrogen peroxide (H_2_O_2_) (TA-125-H2O2Q, Thermo Fisher Scientific, Waltham, MA, USA) was used to block endogenous peroxidase activity for 10 min. The sections were then immersed in 10 mmol/L ethylenediaminetetraacetic acid (EDTA) for 3 min in a pressure cooker, then blocked with 10% normal serum. Thereafter, tissue samples were incubated with primary antibody against STAT3 (1:200) or STAT5 (1:200) at 4 °C overnight, and then with biotin-labeled secondary antibody at room temperature for 1h. Sections were incubated in diaminobenzidine (DAB) and then counterstained with hematoxylin. Visualization was done under a light microscope. For staining determination, we used the Quick (Q) score formula: Q = [percentage of stained/positive cells] × [intensity of staining] The maximum score for percentage/distribution of stained/positive cells was 100, while the intensity of staining was delineated into weak (1), moderate (2), or strong (3), making the obtainable maximum Q score = 300.

### 2.7. Immunofluorescence Staining and Quantification

U87MG and GBM8401 cell lines pre-treated with or without 2.5 μM or 5 μM of garcinol were seeded in 6-well chamber slides (Nunc, Thermo Fisher Scientific, Taipei, Taiwan), incubated at 4 °C overnight, fixed with 2% paraformaldehyde for 10 min at room temperature, and permeabilized in 0.01 M phosphate-buffered saline (PBS) with 0.1% Triton X-100 and 0.2% bovine serum albumin (BSA). The slides were air-dried and rehydrated with PBS before incubation with primary antibodies against OCT4 (#2840; Cell Signaling Technology) and/or SOX2 (#3579; Cell Signaling Technology) at 1:500 dilution in PBS for 2 h at room temperature. The slides were washed with PBS twice for 10 min each, then incubated with anti-rabbit IgG fluorescein isothiocyanate (FITC)-conjugated secondary antibody (diluted 1:500; Jackson Immunoresearch Lab. Inc., West Grove, PA, USA) in PBS for 1 h. The slides were mounted with Vectashield mounting medium and counter stained with 4′, 6′-diamidino-2-phenylindole (DAPI) for nucleus visualization.

### 2.8. Tumorsphere Formation Assay

For tumorsphere formation, U87MG and GBM8401 cells were cultured in Chemicon^®^ serum-free HEScGRO medium for human embryonic stem cell culture (CAT. No. SCM020, Merck KGaA, Darmstadt, Germany) supplemented with 10 ng/mL human recombinant basic fibroblast growth factor (hbFGF; Invitrogen, Carlsbad, CA, USA), 20 ng/mL human epithelial growth factor (hEGF; Millipore, Bedford, MA, USA), B27 supplement (Invitrogen, Carlsbad, CA, USA), heparin (CAT. No. 07980; STEMCELL Technologies Inc., Interlab Co., Ltd., Taipei, Taiwan), and NeuroCult^TM^ NS-A proliferation supplement (CAT. No. 05753, STEMCELL Technologies Inc., Interlab Co., Ltd., Taipei, Taiwan). The cells were seeded at a concentration of 1 × 10^3^ cells/mL/well in 6-well ultra-low adhesion plates (Corning Inc., Corning, NY, USA) with or without 5 µM of garcinol and incubated at 37 °C in 5% humidified CO_2_ atmosphere for 7–10 days. The anchorage-independent tumorspheres (≥90 µm in diameter) were counted under inverted phase contrast microscope at a magnification of 40× and photographed. Tumorsphere sizes were determined from 5 randomly-selected fields of the digital images acquired, using NIH ImageJ software (https://imagej.nih.gov/ij/). We calculated our tumorsphere formation efficiency (MFE%) using the formula: MFE (%) = [No. of tumorspheres per well]/(No. of seeded cells per well) × 100.

### 2.9. Colony Formation Assay

U87MG and GBM8401 cells were seeded in triplicates at a density of 2 × 10^4^ cells per well in 6-well culture plates (Corning, Corning, NY, USA) with complete growth media containing 0 µM, 2.5 µM, or 5 µM of garcinol and incubated at 37 °C for 12–14 days. Culture plates with colonies with ≥100 µm in diameter and ≥50 cells were washed with PBS twice, fixed with methanol for 15 min, and stained with 0.005% crystal violet for 15 min at room temperature. The colonies formed were then visualized under microscope and counted using the ChemiDoc-XRS imager (QuantityOne software package; Bio-Rad, Hercules, CA, USA)

### 2.10. Invasion Assay

3 × 10^4^ U87MG and GBM8401 cells were seeded per well onto 8 µm pore membrane coated with Matrigel in the upper chamber of 24-well Transwell chambers containing serum-free DMEM medium with 0, 2.5, or 5 µM of garcinol, while the lower chambers contained growth media with 10% FBS serving as chemo-attractant. After 24 h incubation the media was discarded, and the non-invaded cells on the upper surface of the insert were removed with sterile cotton swabs, while the invaded GBM cells on lower surface of the filter membrane were fixed with 3.7% formaldehyde for 1 h, and stained with crystal violet dye for 20 min. The stained cells were visualized under microscope and then analyzed using NIH ImageJ software (https://imagej.nih.gov/ij/).

### 2.11. Wound-Healing Migration Assay

After U87MG and GBM8401 cells were seeded in 24-well plates (Corning, Corning, NY, USA) with DMEM with 10% FBS and incubated until 100% confluency, scratch-wounds were made along the median axis of the adherent monolayer cells using sterile 200 µL micropipette tips. The wells were carefully washed with PBS to remove detached cells and then incubated in new growth media containing 0, 2.5, or 5 µM of garcinol for 24 h or 48 h. Photographs of scratch-wound healing were taken at indicated time-points and under microscope with a 10× objective lens, and analyzed with the NIH ImageJ software (https://imagej.nih.gov/ij/). The percentage migration (M%) was calculated using the formula: M (%) = [denuded area at time ‘x’]/(denuded area at time 0) × 100.

### 2.12. Real-Time Polymerase Chain Reaction (qRT-PCR) Analysis

Total RNA extracted from U87MG or GBM8401 cells treated with or without 5 µM of garcinol using Trizol reagent (Invitrogen, Carlsbad, CA, USA) following manufacturer’s instruction. 2 µg of RNA was added to the real time PCR, with the final primer concentration being 0.5 µM. The PCR was performed under the following condition: reverse transcription at 42 °C for 60 min, amplification for 30 cycles at 94 °C for 30 s, 58 °C for 50 s, and 72 °C for 50 s.

### 2.13. PE-Annexin V/7-AAD Cell Death Assay

PE-Annexin V/7-AAD staining was used for detection of cell death using the BD FACSCanto™ II flow cytometry system (BD Biosciences, San Jose, CA, USA) following the manufacturer’s instructions. Briefly, after washing 5 × 10^5^ wild type, Z-VAD-FMK-treated or Garcinol-treated U87MG cells twice with PBS, and once with annexin V binding buffer, the cells were incubated with PE-labeled annexin V and 7-AAD at room temperature for 15 min and then analyzed using flow cytometry. The mitochondrial transmembrane potential (DΨm) was evaluated using the cationic dye JC-1 (Mitochondrial Membrane Potential Assay Kit, #ab113850, Abcam plc., Cambridge, UK) in accordance with the manufacturer’s instructions (BD Pharmingen); 1 × 10^6^ U87MG cells were incubated with 10 µg/mL JC-1 at 37 °C in the dark for 15 min, and then analyzed using flow cytometry. All samples were assayed three times in triplicate.

### 2.14. Mir-181 Transfection Assay

U87MG cells (5 × 10^4^) were seeded in 24-well plates 24 h before transfection. Transfection with mi-181d control, Syn-mir, inhibitor or mimic was performed using Hiperfect Transfection Reagent (#301705, QiIAGEN Inc., Germantown, MD, USA) following the manufacturer’s instructions. miScript inhibitor negative control (#1027272), Syn-hsa-miRNA-181d-5p miScript miRNA mimic (#MSY0002821), Anti-hsa-miRNA-181d-5p miScript miRNA inhibitor (#MIN0002821), and Hs_miR-181d_2 miScript Primer assay (#MS00031500) were purchased from QIAGEN Inc. (Germantown, MD, USA). Briefly, after adding 3 µL Hiperfect transfection reagent to 100 µL FBS-free DMEM containing miR-181d mimic or negative control to a final concentration of 50 nmol/L, culture media volume in the wells were adjusted to 600 µL using medium supplemented with 10% FBS. For evaluation of transfection efficiency, green fluorescent protein (GFP) expression was monitored under fluorescence microscope. We also measured miR181d expression in transfected U87MG cells by real-time PCR.

### 2.15. Mice Tumor Xenograft Study

The animal study protocol was approved by the Animal Care and User Committee at Taipei Medical University (Affidavit of Approval of Animal Use Protocol # Taipei Medical University-LAC-2017-0512) consistent with the U.S. National Institutes of Health Guide for the Care and Use of Laboratory Animals. NOD/SCID mice purchased from BioLASCO (BioLASCO Taiwan Co., Ltd. Taipei, Taiwan) were inoculated with 1 × 106 U87MG cells in the flank subcutaneously, and then randomly placed into untreated control (0.1% DMSO, 100 µL daily) or 1 mg/kg garcinol-treated (1 mg/kg body weight, suspended in 0.1% DMSO, intraperitoneal (i.p.) injection, five times/week) group. Tumor growth was measured using standard caliper and the tumor volume (v) was calculated using the formula: v = l × w2 × 0.5, where l is the longest diameter, and w is the shortest diameter of the tumor. The survival ratio of the two groups were tracked and represented using a Kaplan–Meier survival curve, using GraphPad Prism 5 software. Post experiment, the mice were humanely sacrificed, and tumor samples were collected for further comparative immunohistochemistry (IHC) and miRNA analyses.

### 2.16. Statistical Analysis

All experiments were performed at least 3 times in triplicates. Data presented represent means ± SD of all results. The comparison between two groups was done using the 2-sided Student’s *t*-test, while one-way analysis of variance (ANOVA) was used to compare ≥3 groups. A *p*-value < 0.05 was considered statistically significant.

## 3. Results

### 3.1. STAT-3 and STAT-5 Are Highly Expressed in Primary and Recurrent Glioblastoma, and Their Expression Negatively Correlates with Overall Survival Rates

Firstly, against the background that *STAT3* and *STAT5* to be highly expressed in GBM cell lines [20], we evaluated the expression profiles of STAT proteins GDC TCGA-GBM cohort of 173 samples. Our results showed a 2.226-fold (*t*-Test = 13.114, *p* = 5.83 × 10^−8^) or 2.681-fold (*t*-Test = 14.037, *p* = 5.19 × 10^−8^) increase in the gene expression of *STAT3* in the primary and recurrent GBM samples, respectively, compared to non-tumor samples; similarly, compared to the non-tumor control, the median expression of *STAT5A* was elevated by 1.492-fold (*t*-Test = 2.211, *p* = 0.036) or 2.453-fold (*t*-Test = 4.081, *p* = 0.001) in the primary and recurrent GBM samples, respectively (Figure 1A). In line with the above, using the Kaplan–Meier plots and median-based high/low dichotomization of gene expression, our survival analyses showed a significantly strong association between worse OS and high *STAT3* (*p* < 0.015) or *STAT5* (*p* < 0.008) expression levels, as demonstrated by ~15.8% or ~11.3% survival advantage in patients with high *STAT3* or *STAT5*, respectively, compared to the low group by year two after diagnosis (Figure 1B, upper). Interestingly, we also demonstrated that compared with the ~20%, 11%, or 10% OS amongst patients with *STAT3^low^STAT5A^low^, STAT3^low^STAT5A^high^*, or *STAT3^high^STAT5A^low^* GBM, respectively, no patient with *STAT3^high^STAT5A^high^* GBM was alive by year three after diagnosis, suggesting a strong association between *STAT3^high^STAT5A^high^* and GBM-specific mortality (*p* < 0.007) (Figure 1B, lower). Moreover, consistent with the RNA expression, results of our immunohistochemical staining confirmed elevated STAT3, pSTAT3, STAT5A, and pSTAT5A protein expression levels in primary (*n* = 31) and recurrent (*n* = 14) GBM tissues compared to non-tumor tissues (STAT3/5A vs. non-tumor: ~5.3-fold, *p* < 0.001; pSTAT3/5A vs. non-tumor: ~9.0-fold, *p* < 0.001) (Figure 1C,D). These results indicate, at least in part, that enhanced expression and/or activation of *STAT3* and *STAT5A* play a critical role in the development and recurrence of GBM.

### 3.2. Garcinol Significantly Inhibits GBM Cell Viability and Oncogenicity through Induction of STAT3/5A Signaling and Enhanced Apoptosis

Against the background of recent work demonstrating that garcinol inhibits CSCs-like phenotype of human non-small cell lung carcinoma by suppressing the Wnt/β-catenin/STAT3 signaling axis [30], we investigated the probable STAT signaling-mediated anti-GBM effect of garcinol (Figure 2A). Firstly, to provide some mechanistic insight, we demonstrated that treatment of U87MG or GBM8401 cells with 2.5 µM or 5 µM garcinol significantly downregulated the expression of p-STAT3, p-STAT5, p-ERK, and p-AKT (Figure 2B), synchronous with the observed inhibition of *STAT3*, *STAT5*, and *AKT* signaling, garcinol significantly suppressed the viability of GBM4801 and U87MG cells, with 10 µM eliciting 51% or 25% reduced viability of U87MG or GBM8401 cells, respectively, and 40 µM eliciting 94.7% reduction of U87MG and GBM8401 cell viability, indicating a dose-dependent GBM cell killing effect (Figure 2C), and this reduced viability was associated with markedly enhanced Bax/Bcl-xL apoptotic ratio, as 2.5 µM induced a 1.67-fold (*p* < 0.05) or 2.7-fold (*p* < 0.05) increase in U87MG or GBM8401 apoptotic ratio, while 5 µM increased the apoptotic ratio by 2.83-fold (*p* < 0.001) or 2.92-fold (*p* < 0.001) in the U87MG or GBM8401 cells, respectively (Figure 2D). In addition, using the Phycoerythrin (PE)-conjugated Annexin V/7-Amino-Actinomycin (7-AAD) staining, we demonstrated that compared to the cell death in the untreated control (U87MG: 0.379%, GBM8401: 0.208%) or 20 µM pan-caspase inhibitor benzyloxycarbonyl-Val-Ala-Asp-fluoromethyl ketone (Z-VAD-FMK)-treated negative control (U87MG: 0.157%, GBM8401: 0.071%), treatment with 2.5 µM garcinol enhanced cell death (U87MG: 18.76%, GBM8401: 17.92%), while 2.5 µM or 5 µM garcinol in the presence of 20 µM Z-VAD-FMK elicited 5.46% or 10.96% apoptosis of the GBM8401 cells, respectively, and 7.97% or 14.11% apoptosis of the U87MG cells (Figure 2E), indicating that the garcinol-induced cell death was apoptotic and caspase-dependent. Since the highly invasive GBM spreads fast to surrounding brain tissue, thus, contributing to its documented lethality [2], we sought to understand if and how garcinol affects this invasive trait. We demonstrated that treatment with 2.5 µM or 5 µM dose-dependently suppressed the migration of the U87MG (~59%, *p* < 0.01 or 81%, *p* < 0.001, respectively) and GBM8401 (~48%, *p* < 0.01 or 76%, *p* < 0.001, respectively) cells at the 24 h time-point (Figure 2F). Similarly, 2.5 µM or 5 µM garcinol induced a 60% (*p* < 0.01) or ~80% (*p* < 0.001) reduction of U87MG invasive capacity, respectively, and 39% (*p* < 0.01) or 60% (*p* < 0.001) reduction in number of invaded GBM8401 cells (Figure 2G). Furthermore, in parallel assays to confirm the anticancer role of garcinol, consistent with earlier results, we demonstrated that treatment with 2.5 µM or 5 µM garcinol, significantly suppressed the expression of N-cadherin, vimentin, and slug proteins, while conversely upregulating the expression of E-cadherin protein (Figure 2H), thus indicating that garcinol attenuates epithelial-to-mesenchymal transition (EMT) and the metastatic phenotype of GBM cells. Together, these data suggest that garcinol significantly inhibits GBM cell viability and oncogenicity through induction of STAT3/5A and associated signaling with enhanced apoptosis.

### 3.3. Garcinol Negatively Impacts GBM Stem Cell-Like Phenotypes

Understanding that the highly prevalent and malignant GBM harbors self-renewing, tumorigenic GBM-SCs that facilitate tumor initiation and resistance to therapy [36,37]. To assess the effects of garcinol on GBM-SCs, we performed tumorsphere and colony formation assays on the GBM8401 and U87MG cell lines. The results from the tumorsphere assay demonstrated that 5 µM garcinol significantly caused both cell lines to lose their ability to form GBM tumorspheres, quantitatively and qualitatively, with ~88% (*p* < 0.01) reduction in the number of U87MG or GBM8401 tumorspheres formed, and ~96% (*p* < 0.01) or 89% (*p* < 0.001) reduction in the U87MG or GBM8401 tumorsphere sizes, respectively (Figure 3A). Furthermore, because of the biological relevance of clonality in GBM-SCs origin [36], we demonstrated that 2.5–5 µM garcinol significantly inhibited the ability of the GBM cells to form colonies, dose-dependently, as 2.5 µM reduced the number of formed U87MG or GBM8401 colonies by 49% (*p* < 0.05) or 36% (*p* < 0.05), respectively, while 5 µM induced a 75% (*p* < 0.01) or 72% (*p* < 0.01) reduction, respectively (Figure 3B). Contextually, the garcinol-induced inhibition of tumorsphere and colony formation potential was associated with significant and dose-dependent downregulation of the nuclear expression of stemness proteins SOX2 and OCT4 (Figure 3C) as shown with immunofluorescence (IFC) assay, and this inhibitory effect was associated with significantly suppressed p-STAT3 and p-STAT5A immunofluorescence, in a dose-dependent manner (Figure 3D). Consistently, akin to the IFC results, western blot analyses also showed that treatment with garcinol downregulated SOX2 (2.5 µM: 1.92-fold, *p* < 0.05; 5 µM: 2.78-fold, *p* < 0.01) and OCT4 (2.5 µM: 2.63-fold, *p* < 0.05; 5 µM: 5.88-fold, *p* < 0.01) protein expression levels in U87MG tumorspheres with similar trend in the GBM8401 tumospheres (Figure 3E). These data are indicative of the negative influence of garcinol on the stem cell-like phenotypes of GBM cells.

### 3.4. Garcinol Increases the Expression of Hsa-miR181d, which Has Inhibitory Effects on STAT3 and STAT5 Activation

Having established that garcinol impairs STAT3 and STAT5A activation, we probed for likely modulators and/or mediators of the interaction between garcinol and the STAT proteins. Hsa-miR-181d shown in Figure 4A, has been implicated in the worse OS of patients with GBM [27]. Consistent with this, using the Schrodinger PyMOL 2.3 molecular docking and visualization software (http://pymol.org) we demonstrated that hsa-miR-181d interacts with and binds directly to STAT3 (docking score = −254.49, ligand root mean square deviation (RMSD) = 195.10 Å) or STAT5A (docking score = −234.19, ligand RMSD = 143.04 Å), complementing earlier prediction that hsa-miR-181d binds with STAT3 with a mirSVR or PhastCons score of −0.26 or 0.69, respectively, while it binds with STAT5A with a mirSVR or PhastCons score of −0.21 or 0.49, respectively (Figure 4A). Concomitantly, as shown in Figure 4A, results of our nucleotide complementarity analysis indicate that the 5′-UTR (untranslated region) of hsa-miR-181d binds to the 3′ UTR of JAK2 with a mirSVR or PhastCons score of −0.79 or 0.61, respectively. Where the mirSVR shows the likelihood of hsa-miR-181d down-regulating the target mRNA STAT3 and STAT5A based on the sequence and structure features in the miRNA/mRNA predicted target sites. Moreover, the PhastCons score shows the likelihood that the predicted miRNA/mRNA binding nucleotides are conserved. In concordance, our qRT-PCR analysis of 5 µM garcinol-treated U87MG and GBM8401 cells showed that garcinol significantly induced higher expression of miR-181d in the U87MG (2.7-fold, *p* < 0.01) and GBM8401 (2.1-fold, *p* < 0.01) cells (Figure 4B). Furthermore, having implicated STAT3/5A in enhanced migration and invasiveness of U87MG or GBM8401 cells, we examined the probable effect of hsa-miR-181d on these metastatic phenotypes of GBM cells. Using the scratch wound-healing assay, we demonstrated that compared to the untreated control or syn-mir-treated cells, treatment with mir-181d inhibitor significantly enhanced the ability of the U87MG cells to migrate (4.64-fold, *p* < 0.01), while treatment with the mir-181d-mimic elicited marked attenuation of migration (3.80-fold, *p* < 0.01) (Figure 4C), which is reminiscent of suppressed migration induced by 5 µM garcinol (4.17-fold, *p* < 0.01) earlier. Similarly, while treatment with mir-181d inhibitor significantly enhanced the invasiveness of the U87MG cells (4.63-fold, *p* < 0.01), treatment with the mir-181d-mimic elicited profound suppression of invasion (4.15-fold, *p* < 0.01) (Figure 4D), and this was akin to the effect of 5 µM garcinol (4.22-fold, *p* < 0.01). To confirm a direct relationship between the STAT proteins and miR-181d expression, western blot analysis was done comparing samples exposed to mir-181d inhibitor, mir-181d-mimic, or mir-181d inhibitor/garcinol combination. The results showed that mir-181d inhibitor significantly enhanced the expression of p-STAT3, p-STAT5, N-cadherin and vimentin proteins, but suppressed E-cadherin protein expression compared to the control group, while for the mir-181d-mimic-treated cells, the p-STAT3, p-STAT5, N-cadherin, and vimentin protein expression levels were significantly lower, but E-cadherin was upregulated; For cells incubated with mir-181d inhibitor and 5 µM garcinol concomitantly, p-STAT3 and p-STAT5 protein expression levels were markedly higher than in the mir-181d-mimic group but lower than the mir-181d-inhibitor group (Figure 4E). Concomitantly, we observed that compared with the control group, syn-mir-treated cells, or even mir-181d inhibitor-treated cells, treatment with mir-181d-mimic markedly suppressed the expression of JAK2 protein (2.22-fold, *p* < 0.01), akin to the effect elicited by treatment with concurrently with mir-181d inhibitor and 5 µM garcinol (2.94-fold, *p* < 0.01), which is consistent with the results above and suggestive of a miR-181d-mediated JAK2-modulated phosphorylation of STAT3/5A. These results indicate that garcinol can activate mir-181d activity which suppresses JAK2-modulated STAT3/5A activation.

### 3.5. Garcinol Inhibits Tumor Growth in GBM Mice Models through Inversely Correlated STAT3/5A and hsa-miR-181d Expressions

Having shown that treatment with garcinol suppresses the cancer stem cell-like phenotype of U87MG and GBM8401 cells in vitro, to determine the probable suppressive effect of garcinol on the formation and growth of tumor, in vivo, we generated NOD/SCID mice xenograft models derived by inoculation with 1 × 10^6^ U87MG cells subcutaneously in the hind-flank. Mice were randomly placed into control or garcinol treatment group. We demonstrated that treatment with 1 mg/kg garcinol significantly reduced the size of tumors formed in the treated mice, compared to the untreated control group (U87MG: ~7.1-fold smaller, *p* < 0.001 by week 4) (Figure 5A), without adversely affecting the mice body weight (Figure 5B). We also observed that mice treated with garcinol showed 100% survival as compared to 60% in the control counterparts, over the four-week treatment period. (Figure 5C). In subsequent experiments using protein lysates derived from the tumors extracted from the untreated and garcinol-treated mice, we demonstrated that compared to the untreated control group, STAT3, pSTAT3, STAT5A, p-STAT5A, Ki-67, and Bcl-xL protein expression levels were concomitantly suppressed, while Bax expression was significantly enhanced (Figure 5D). Moreover, for tumors extracted from the 1 mg/kg garcinol-treated U87MG tumor-bearing mice, compared to the untreated control group, *STAT3*, or *STAT5A* mRNA expression were suppressed by four-fold (*p* < 0.01) or 3.87-fold (*p* < 0.01), while miR-181d expression was enhanced by 3.52-fold (*p* < 0.01) in the U87MG mice treated with 1 mg/kg garcinol (Figure 5E). These findings indicate that garcinol inhibits tumorigenicity and growth of GBM by abrogating STAT3/5A signaling, and upregulating hsa-miR-181d, with concomitant suppression of Ki-67 proliferation index and enhancement of Bax/Bcl-xL apoptotic ratio, in vivo.

### 3.6. Garcinol, Akin to Stattic, a Selective Inhibitor of STAT3/5A Activation, Inhibits the Metastatic and Cancer Stem Cell-Like Phenotypes of Primary GBM Culture Cells

Sequel to our data demonstrating that garcinol suppresses the cancer stem cell-like phenotype of U87MG and GBM8401 cells in vitro, and that garcinol also suppresses the formation and growth of tumor in mice GBM models by upregulating hsa-miR-181d expression and inhibiting STAT3/5A activation, we further investigated if indeed these findings could be replicated in GBM primary culture cells. Comparative analyses of the anti-GBM therapeutic effects of garcinol and stattic, a selective inhibitor of STAT3/5A activation and dimerization, revealed that akin to the 25–98% reduction in cell viability of the primary GBM culture cells by 2.5–40 µM static, equimolar garcinol treatment dose-dependently elicited a 13–96.8% reduced viability of the primary GBM culture cells (Figure 6A). We also observed that concurrent with reduced cell viability, 2.5 and 5 µM garcinol or stattic induced significant downregulation of p-STAT3 (garcinol: 40% and 85%, *p* < 0.01; stattic: 56% and 89%, *p* < 0.01) and p-STAT5A (garcinol: 35% and 83%, *p* < 0.01; stattic: 48% and 74%, *p* < 0.01) in the primary GBM culture cells (Figure 6B). Furthermore, while treatment with 2.5 and 5 µM stattic attenuated the migration of the primary culture cells by 49.3% (*p* < 0.05) and 88.9% (*p* < 0.001), 2.5 and 5 µM garcinol elicited 48% (*p* < 0.05) and 87.5% (*p* < 0.001) (Figure 6C). Similarly, suppression of cell invasion was demonstrated, with 2.5 and 5 µM stattic causing 64% (*p* < 0.01) and 91% (*p* < 0.001) reduction in migration, respectively, and equimolar garcinol induced 51% (*p* < 0.01) and 84% (*p* < 0.001), respectively (Figure 6D). We also showed that comparable to stattic, 2.5 and 5 µM garcinol reduced the number of colonies formed by 50.7% (*p* < 0.01) and 91% (*p* < 0.001), respectively (Figure 6E). Moreover, akin to stattic, 5 µM garcinol reduced the number of primary GBM culture tumorspheres formed by 91.7% (*p* < 0.001) and the tumorsphere sizes by ~90% (*p* < 0.001) (Figure 6F). These findings not only validate the replicability of garcinol anti-GBM effect in vitro, in vivo and ex vivo using primary GBM culture cells, and enhance clinical or translational relevance, but also demonstrate, at least in part, that akin to stattic, garcinol inhibits the metastatic and stemness phenotypes of primary GBM culture cells by inhibiting the activation of STAT3/5A, which is consistent with results in Figure 4 and Figure 5 showing that garcinol inhibits metastasis, cancer stemness and tumor growth through enhanced hsa-miR-181d/STAT3 or hsa-miR-181d/STAT5A.

## 4. Discussion

STAT3 and STAT5 are transcription factors implicated in various tumor cell proliferation, migration, and invasion [20]. STAT3 and STAT5A, in response to cytokines and growth factors, are activated by phosphorylation of their Y705 position and Y695 residues, respectively, followed by nuclear translocation of the phosphorylated proteins with subsequent activation of specific downstream gene transcription [38]. Activation of STAT3 and STAT5 serve different purposes. STAT3 has been shown to promote the proliferation and pathobiology of GBM tumor cores, while STAT5 affects its local invasion capabilities [20]. Evidence also abound that STAT3 signaling plays a role in maintaining the self-renewal capabilities and multilineage differentiation potential of glioma stem cells (GSMs) [15]. As with *STAT3*, genetic alterations in the *STAT5A* gene has been implicated in myeloproliferative disorders and linked to hematopoietic stem cell proliferation [39].

Past works from our team demonstrated that garcinol inhibited *STAT3* activation and suppressed the lung cancer stem cell population [30]. In the present study, investigating if glioblastoma patients also exhibit high levels of *STAT3* and the associated *STAT5A*, we performed RNAseq analysis of the GDC-TCGA glioblastoma cohort and demonstrated that both primary and recurrent glioblastoma patients have higher expression of *STAT3* and *STAT5*. Consistent with past evidence showing aberrant STAT activity in glioblastoma cell lines, immunohistochemical analysis revealed amplified *STAT3/5A* expression in primary and recurrent GBM. The increased RNA and protein expression indicated *STAT3* and *STAT5A* play a critical role in driving GBM tumorigenesis. Furthermore, recurrent glioblastoma displayed a general trend of increased *STAT5A* expression, suggesting a specific role for *STAT5A* in maintaining the GBM-SC phenotype. Upon analysis of the differential *STAT3/5A* expression-based survival rates, high expression of both *STAT3* and *STAT5A* was found to be strongly associated with the poorest OS time; and may relate to the enhanced tumorigenicity and lethality of GBM in oncology clinics. Of translational relevance, we observed no significant difference in OS time between the *STAT3^high^STAT5A^low^* cohort and the *STAT3^low^STAT5A^high^* cohort, suggesting there is no specific advantage in inhibiting the expression of either transcription factor, and highlighting probable therapeutic efficacy of parallel inhibition of both *STAT3* and *STAT5A*. Thus, as indicated by the dismal OS of patients with *STAT3^high^STAT5^high^* compared to the other groups, we, for the first time to the best of our knowledge demonstrate that concerted targeting of the aberrant expression and/or activity of *STAT3/5A* in GBM, both at protein and mRNA levels, is essential for any meaningful curative effect in STAT-based targeted therapy for patients with primary or even recurrent GBM. This would be consistent with recent evidence indicating that combined targeting of *STAT3* and *STAT5* effectively reversed various forms of tyrosine kinase inhibitor (TKI)-resistance in highly resistant BCR-ABL1^T315I^ chronic myeloid lymphoma (CML) cells [40].

In the context of the GBM SCs-phenotypes including enhanced proliferation, oncogenicity, therapy resistance, recurrence and poor prognosis, we further demonstrated that garcinol significantly inhibited GBM8401, U87MG, and GBM primary culture cell viability in a dose-dependent manner, as well as suppresses the cell invasive and migratory potentials, thus, demonstrating the anti-proliferative and anti-metastatic efficacy of garcinol in GBM. This is consistent with documented robust growth-inhibitory effects of garcinol demonstrated against colon cancer and immortalized intestinal cells [32]. Since *STAT3* and *STAT5A* are implicated in the maintenance of the stem cell-like characteristics of GBM, we examined garcinol’s inhibitory effect on the GBM-SCs profile. Interestingly, low dose garcinol (≤5 µM) deregulated STAT3/5A signaling with repressed AKT and ERK crosstalk, and this was sufficient to significantly impeded GBM cell migration, invasion, clonogenicity, and tumorsphere formation, with associated increase in apoptotic index and nuclear expression of SOX2 and OCT4. Our findings are corroborated by recent evidence that *STAT3* and *STAT5* are constitutively activated in malignant cells, and that their persistent activation facilitates cancer development and progression by altering downstream gene expression through epigenetic modification, EMT induction, oncogenic modification of the tumor microenvironment, and enhancing of CSCs self-renewal and differentiation [41], as well as evidence implicating high ERK1/2 activity in the acquisition and maintenance of SOX2-expressing Glioma stem cells [42].

This study also demonstrated for the first time to the best of our knowledge, that garcinol-induced suppression of *STAT3* and *STAT5A* is associated with significant upregulation of hsa-miR-181d expression, in vitro, ex vivo and in vivo; interestingly we also showed direct interaction between hsa-miR-181d and STAT3 or STAT5A protein. We posit that upon treatment with garcinol, miR-181d canonically represses the activation/phosphorylation of STAT3/5A in a JAK2-mediated manner, or as also documented herein, miR-181d non-canonically binds directly to the coding region of *STAT3* or *STAT5A* mRNA, eliciting STAT3/5A degradation, and consequently impair activation of *STAT3* or *STAT5A* in the GBM cells. This demonstrated tumor suppressor role of hsa-miR-181d is consistent with findings showing that overexpression of miR-181d significantly suppressed esophageal squamous cell carcinoma (ESCC) by downregulating Derlin-1, inhibiting cancerous cell proliferation, migration and cell cycle progression in vitro, as well as inhibiting tumorigenicity in vivo [43], as well as in glioma samples and cell lines, where ectopic expression of miR-181d suppressed proliferation and induced cell cycle arrest and apoptosis by targeting K-ras and Bcl-2 [44]. This demonstrated garcinol-modulated miR-181d/STAT3/5A signaling axis is of therapeutic relevance, considering that well documented role of the JAK–STAT signaling pathway, and more particularly its molecular effectors namely *STAT3* and *STAT5* which act as a point of convergence for several signaling pathways in cancerous cells and oncogenic processes [45]. It is worth mentioning however, that while we cannot fully explain how garcinol induced hsa-miR-181d inhibited the activation of *STAT3* and *STAT5*, our data finds some corroboration in increasingly documented role of miRs in the (de)activation of the JAK-STAT signaling [46,47,48], and of particular interest is miR-204 which similarly had very insignificant effect on total *STAT3* expression, but impaired *STAT3* phosphorylation, consequently inducing cancerous cell apoptosis and suppressed cell proliferation, migration in vitro and tumor growth in vivo [48,49]. Moreover, recent report that the Kaposi’s sarcoma-associated herpes virus (KSHV) miRNAs impair the activation/phosphorylation of *STAT3/5*, and inhibit the activation of *STAT3*-dependent reporter upon IL6-treatment, also lend credence to our finding that hsa-miR-181d may bind directly with *STAT3/5A* and impair activation of the later [50,51].

## 5. Conclusions

Taken together, as depicted in our graphical abstract in Figure 7, the present study provides evidence that the constitutive activation of *STAT3/5A* in GBM is inversely correlated with suppressed hsa-miR-181d expression, and that JAK2-mediated garcinol-induced upregulation of hsa-miR-181d/STAT3 and hsa-miR-181d/5A ratios underlies the anti-GBM-SCs effect of garcinol in *STAT3/5A*-addicted GBM. These findings are of translational relevance as they highlight the therapeutic efficacy of a relatively novel small molecule inhibitor of *STAT3/5A* in the highly invasive and often therapy resistant GBM. In addition, findings documented in the present pilot study form a basis for further large cohort exploration of the preclinical feasibility and subsequent clinical applicability of garcinol-modulated hsa-miR-181d/STAT ratio as a therapeutic strategy in GBM.

## Figures and Tables

**Figure 1 cancers-11-01888-f001:**
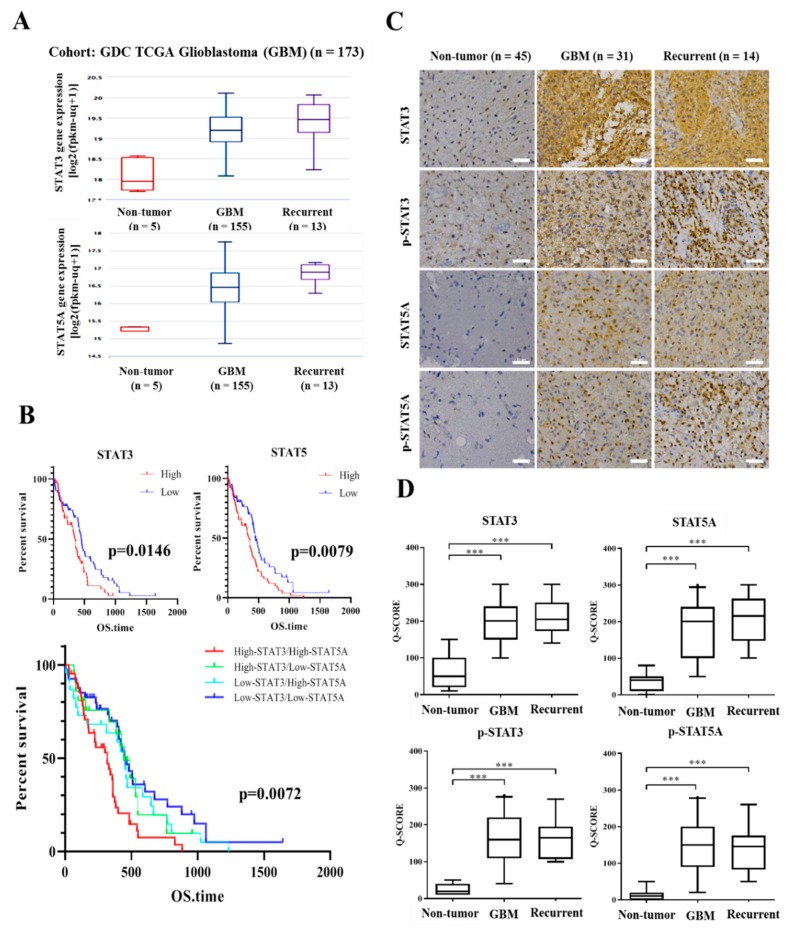
*STAT3* and *STAT5* are highly expressed in primary and recurrent glioblastoma, and their expression negatively correlates with overall survival rates. (**A**) Graphical representation of the differential expression of *STAT3* and *STAT5A* in primary GBM, recurrent GBM or normal brain tissues from the Genomic Data Commons and The Cancer Genome Atlas (GDC-TCGA)-GBM cohort. (**B**) Kaplan–Meier plots of the effect of differential *STAT3* or *STAT5A* expression on OS of patients with GBM. (**C**) Representative immunohistochemistry (IHC) images showing the differential expression of STAT3, pSTAT3, STAT5A, and pSTAT5A proteins in primary GBM, recurrent GBM or normal brain tissues. Scale bar: 25 µm. (**D**) Graphical representation of the differential expression of STAT3, pSTAT3, STAT5A, pSTAT5A proteins in primary GBM, recurrent GBM or normal brain tissues. *** *p* < 0.001; OS, overall survival, GM, glioblastoma.

**Figure 2 cancers-11-01888-f002:**
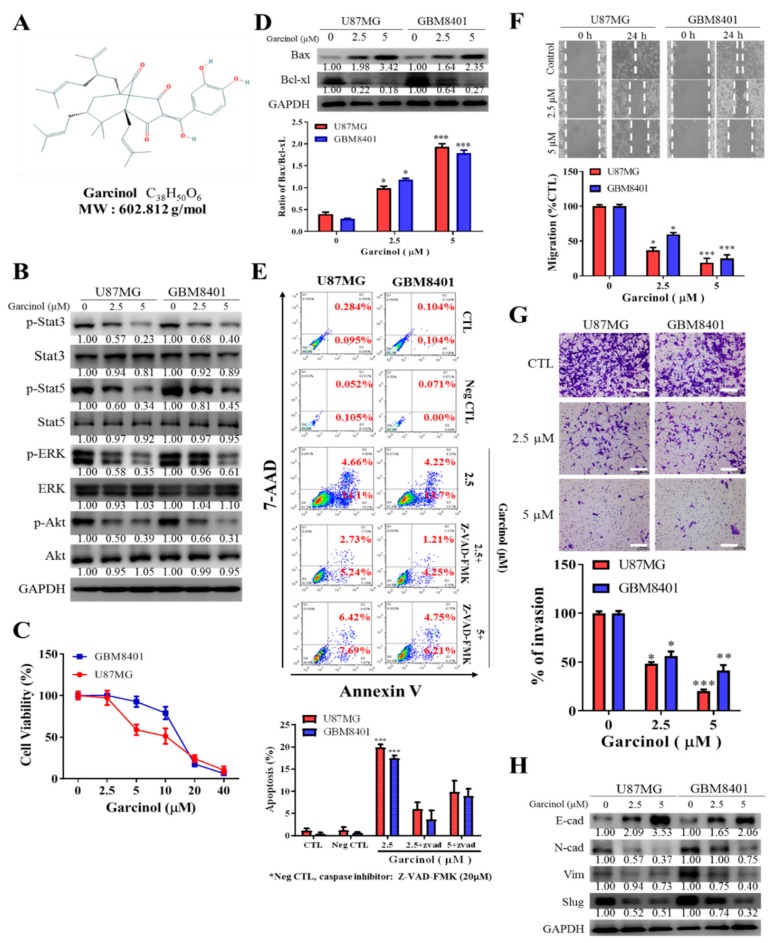
Garcinol significantly inhibits GBM cell viability and oncogenicity through induction of STAT3/5A signaling and enhanced apoptosis. (**A**) Chemical structure of garcinol with molecular formula C_38_H_50_O_6_ and molecular weight 602.80 g/mol. (**B**) Representative western blot photo-images of the effect of 2.5–5 µM on the expression of p-STAT3, STAT3, p-STAT5, STAT5, p-ERK, ERK, p-AKT, and AKT proteins in GBM8401 or U87MG cells. (**C**) Graphical representation of the effect of 2.5–40 µM on the viability of GBM8401 or U87MG cells. (**D**) Representative western-blot photo-images showing the effect of 2.5–5 µM on the expression of Bax and Bcl-xL proteins in GBM8401 or U87MG cells. (**E**) Flow-cytometry data (upper) and graphical representation (lower) showing the effect of Garcinol, alone or in presence of Z-VAD-FMK, on U87MG or GBM8401 cells co-stained with PE-conjugated Annexin V and 7-AAD, compared with untreated control or Z-VAD-FMK-treated negative control groups. Annexin V-stained Q4 cells are early apoptotic cells, whereas Q2 cells are late apoptotic (necrotic) cells. Apoptosis (%), sum of Q4 + Q2; CTL, vehicle-treated; Neg CTL, pan-caspase inhibitor benzyloxycarbonyl-Val-Ala-Asp-fluoromethyl ketone (Z-VAD-FMK). Representative photo-images (upper) and graphical representation (lower) of the effect of 2.5 µM or 5 µM on the (**F**) migration and (**G**) invasion of GBM8401 or U87MG cells. Scale bar: 100 μm. (**H**) Representative western-blot photo-images showing the effect of 2.5–5 µM on the expression of E-cadherin, N-cadherin, vimentin, and slug proteins in GBM8401 or U87MG cells. * *p* < 0.05, ** *p* < 0.01, *** *p* < 0.001; GAPDH is loading control. Detailed information of western blot figures can be found at Appendix A.

**Figure 3 cancers-11-01888-f003:**
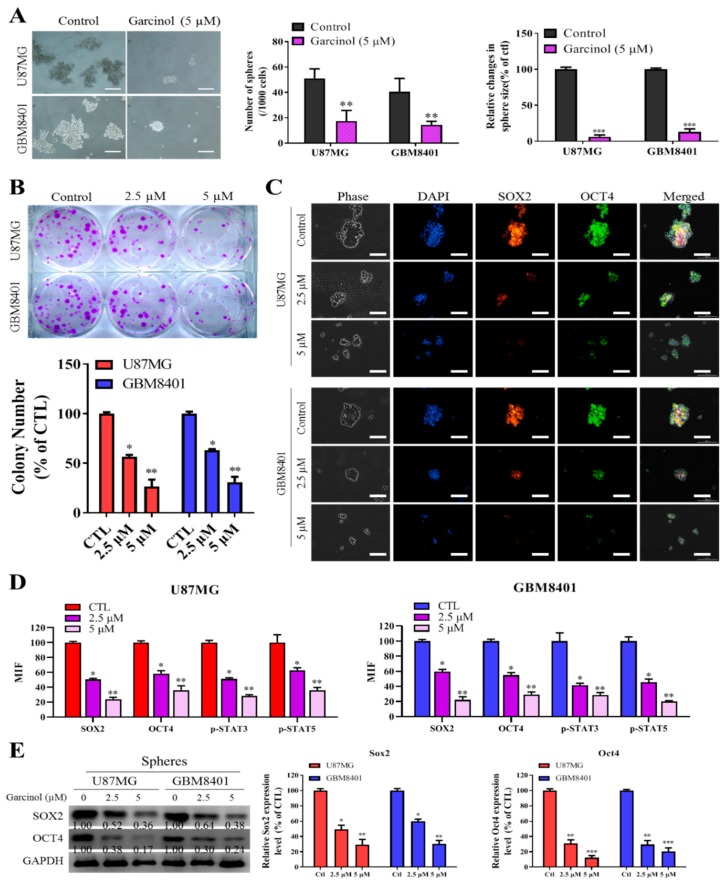
Garcinol negatively impacts GBM stem cell-like phenotypes. (**A**) Representative photo-images (left) and histograms (middle, right) of the effect of 5 µM garcinol on the number and size of tumorspheres formed by U87MG or GBM8401 cells. Scale bar: 100 μm. (**B**) Representative images (upper) and graph (lower) of the effect of 2 µM or 5 µM garcinol on the number of colonies formed by U87MG or GBM8401 cells. (**C**) Representative photo-images showing the effect of 2 µM or 5 µM garcinol on the tumorsphere size and sub-cellular localization of SOX2 and OCT4 proteins in U87MG or GBM8401 cells. Scale bar: 100 μm. (**D**) Graphs showing how 2 µM or 5 µM affect the MIF of SOX2, OCT4, p-STAT3 and p-STAT5A in U87MG or GBM8401 cells. (**E**) Representative western blot images (*left*) and histograms (*right*) showing the effect of 2 µM or 5 µM garcinol on the expression of SOX2 and OCT4 proteins. * *p* < 0.05, ** *p* < 0.01, *** *p* < 0.001; MIF, median immunofluorescence; DAPI served as nuclear marker. GAPDH served as loading control. Detailed information of western blot figures can be found at Appendix A.

**Figure 4 cancers-11-01888-f004:**
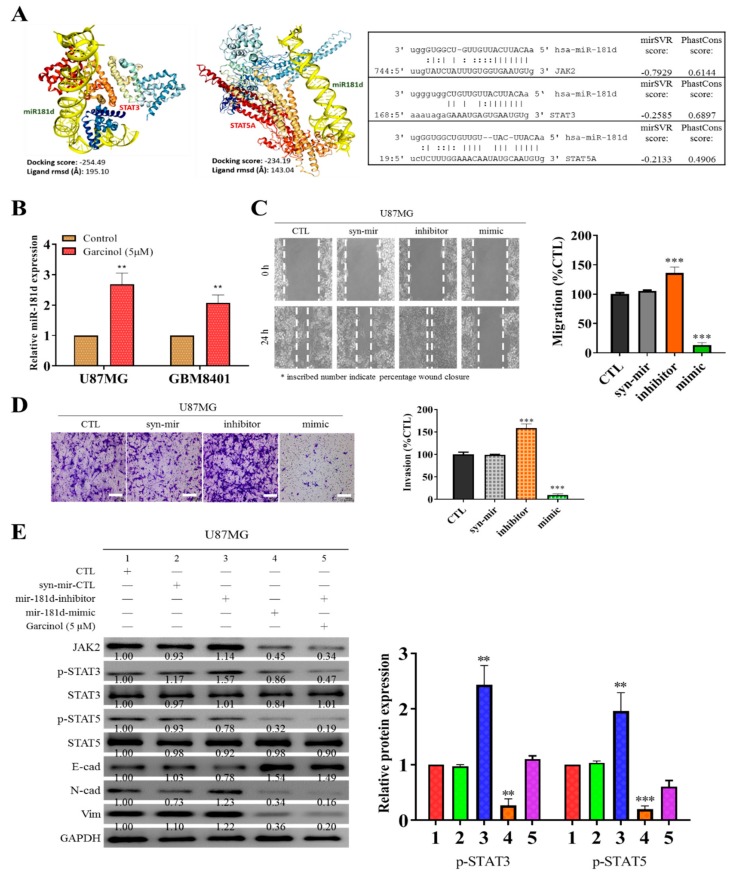
Garcinol increases the expression of hsa-miR181d, which has inhibitory effects on STAT3 and STAT5 activation. (**A**) 3-dimensional visualization of direct interaction between hsa-miR-181d and STAT3 (left) or STAT5A (middle) and images showing the complementary sequence alignment of hsa-miR-181d with JAK2 (upper right), STAT3 (middle right) or STAT5A (lower right). The mirSVR and PhastCons scores are indicated. (**B**) Histograms of the effect of 5 µM on hsa-miR-181d expression in U87MG or GBM8401 cells. (**C**) Representative images (left) and histograms (right) showing the effect of Syn-mir-CTL, mir-181d-inhibitor, or mir-181d-mimic on the migration of U87MG cells over 24 h, as determined by wound-healing assay. (**D**) Representative images (left) and histograms (right) showing the effect of Syn-mir-CTL, mir-181d-inhibitor, or mir-181d-mimic on the invasion of U87MG cells. Scale bar: 100 μm. (**E**) Representative western-blot photo-images (left) and histograms (right) comparing the effect of Syn-mir-CTL, mir-181d-inhibitor, mir-181d-mimic, or garcinol on the expression level of JAK2, p-STAT3, STAT3, p-STAT5, STAT5, E-cadherin, N-cadherin, and vimentin proteins in U87MG or GBM8401 cells. ** *p* < 0.01, *** *p* < 0.001; GAPDH is loading control. Detailed information of western blot figures can be found at Appendix A.

**Figure 5 cancers-11-01888-f005:**
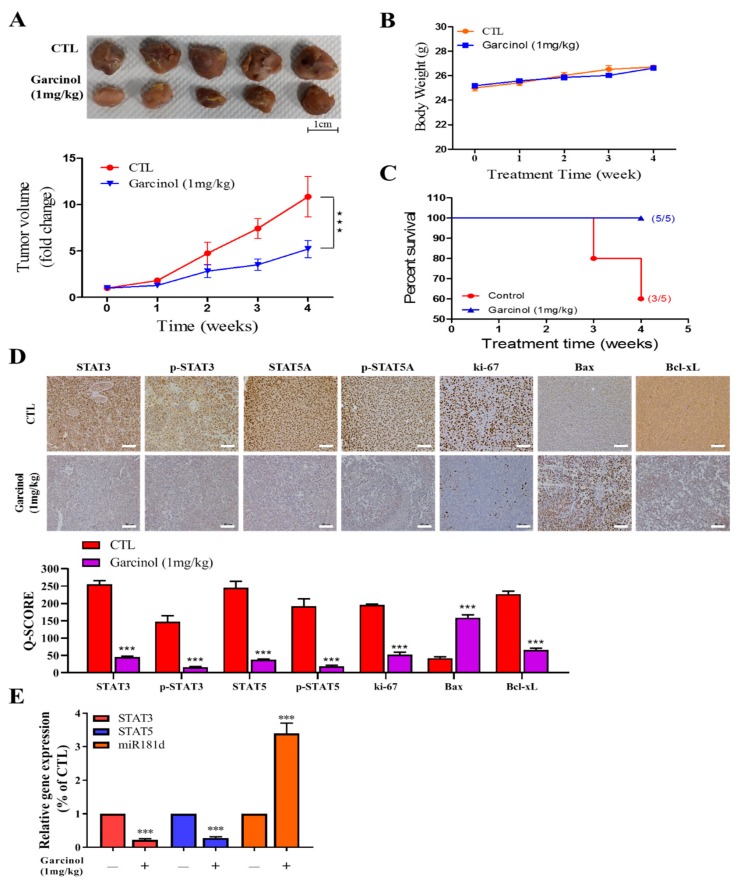
Garcinol inhibits tumor growth in GBM mice models through inversely correlated STAT3/5A and hsa-miR-181d expressions. Representative image and graph showing the effect of garcinol on the (**A**) tumor volume, and (**B**) body weight of U87MG-tumor-bearing mice. *p*-values were determined by 2-way ANOVA. (**C**) Kaplan–Meier Survival curve. Mice treated with garcinol showed 100% survival as compared to 60% in the control counterparts, over the four-week treatment period. (**D**) Representative images and histograms of the differential expression of STAT3, pSTAT3, STAT5A, pSTAT5A, Ki-67, Bax, and Bcl-xL proteins level in tumors extracted from mice bearing U87MG cell-derived tumors, treated with or without garcinol. Scale bar: 50 μm. (**E**) Histograms showing the effect of garcinol treatment on STAT3, STAT5A, and miR-181d expression levels in U87MG-tumor-bearing mice. *** *p* < 0.001; GAPDH is loading control.

**Figure 6 cancers-11-01888-f006:**
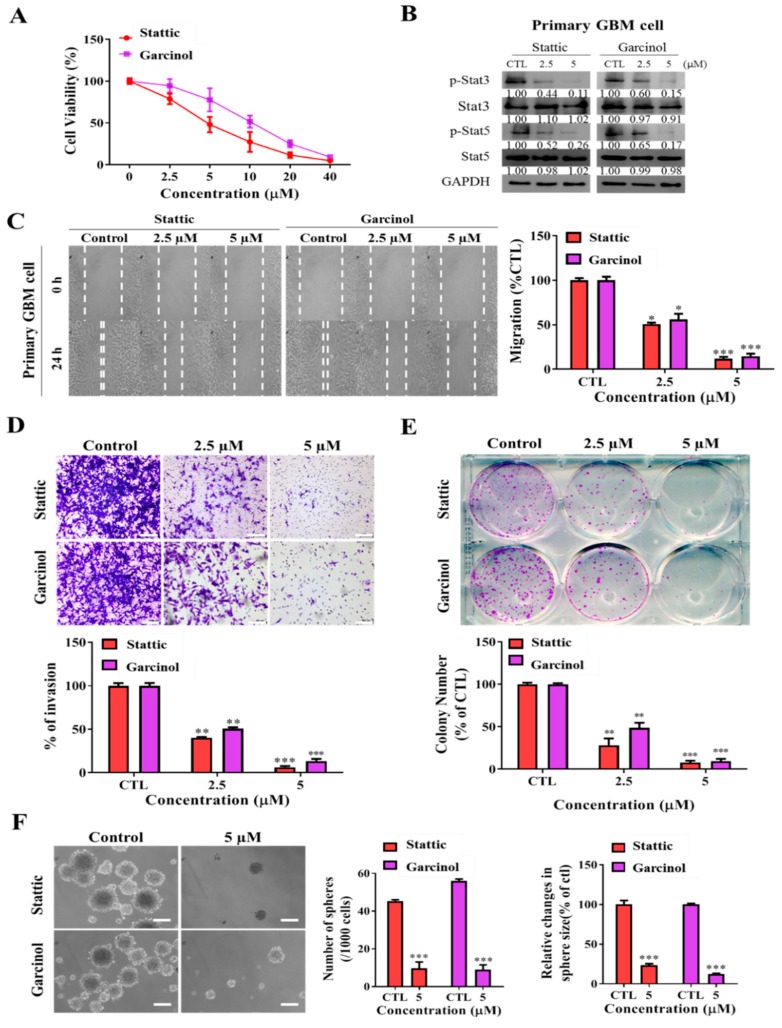
Garcinol, akin to Stattic, a selective inhibitor of STAT3/5A activation, inhibits the metastatic and cancer stem cell-like phenotypes of GBM primary culture cells. (**A**) Graphical representation of the effect of 2.5 µM–40 µM on the viability of GBM primary culture cells. (**B**) Representative western-blot photo-images comparing the effect of stattic and garcinol on the expression level of p-STAT3, STAT3, p-STAT5, and STAT5 proteins in primary culture cells. Detailed information of western blot figures can be found at Appendix A. (**C**) Representative images (left) and histograms (right) comparing the effect of 2.5 and 5 µM stattic or garcinol on the migration of U87MG cells over 24 h, as determined by wound-healing assay. Representative images (upper) and histograms (lower) comparing the effect of 2.5 and 5 µM stattic or garcinol on the (**D**) invasion and (**E**) colony formation capacity of GBM primary culture cells. Scale bar: 100 μm. (**F**) Representative photo-images (left) and histograms comparing the effect of 5 µM stattic and garcinol on the number and size of tumorspheres formed by the GBM primary culture cells. Scale bar: 100 μm. * *p* < 0.05, ** *p* < 0.01, *** *p* < 0.001; GAPDH is loading control.

**Figure 7 cancers-11-01888-f007:**
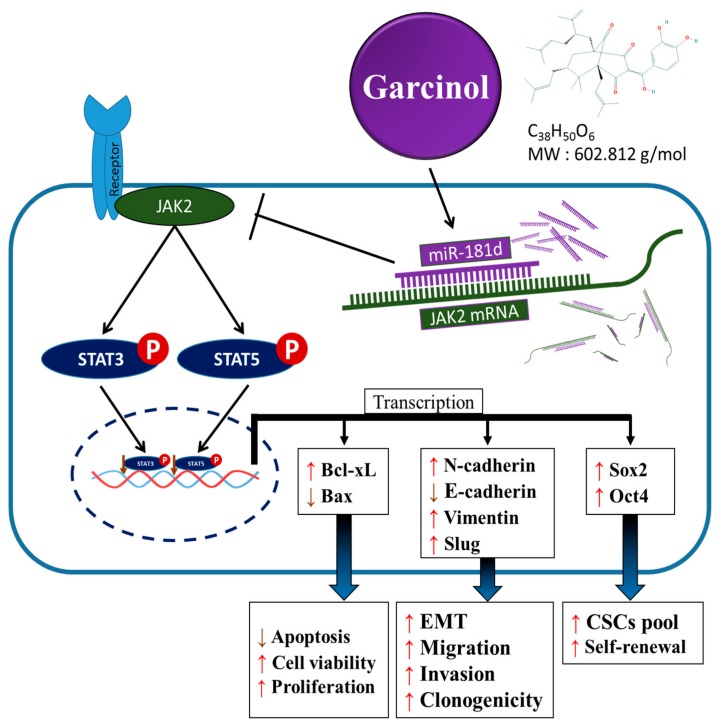
Graphical abstract showing how activation of *STAT3/5A* in GBM is inversely correlated with suppressed hsa-miR-181d expression, and that JAK2-mediated garcinol-induced upregulation of hsa-miR-181d/STAT3 and hsa-miR-181d/5A ratios underlies the anti-GBM-SCs effect of garcinol in STAT3/5A-addicted glioblastoma.

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
