# Peer review of "Enhanced Hsa-miR-181d/p-STAT3 and Hsa-miR-181d/p-STAT5A Ratios Mediate the Anticancer Effect of Garcinol in STAT3/5A-Addicted Glioblastoma"

_cancers, 2019, doi:10.3390/cancers11121888_

Round 1

Reviewer 1 Report

The authors answered almost of my questions.

However, one point remains to be clarified;

The Annexin V/7AAD staining represented in figure 2E is not complete. The dot plots were not so nice and the labeling of the axes is missing. Second the authors represent only double positive cells while usually the representation show both AnnV+/7AAD- and AnnV+/7AAD+ to differentiate apoptosis and late apoptosis/necrosis. Finally the authors used zVAD only in untreated cells. However the interest to use it is to check whether garcinol-induced cell death is mediated by caspases or not, so the authors have to use it in combination with garcinol.

Author Response

Answers to the comments:

Point-by-point responses to reviewer’s comments:

We would like to thank the reviewer for the thorough reading of our manuscript as well as their valuable comments. We have followed their comments closely and feel that they have further improved the readability and appeal of our work, as well as strengthened the manuscript. Below are our point-by-point responses.

Q1: Reviewer #1: The Annexin V/7AAD staining represented in figure 2E is not complete. The dot plots were not so nice and the labeling of the axes is missing. Second the authors represent only double positive cells while usually the representation show both AnnV+/7AAD- and AnnV+/7AAD+ to differentiate apoptosis and late apoptosis/necrosis. Finally the authors used zVAD only in untreated cells. However the interest to use it is to check whether garcinol-induced cell death is mediated by caspases or not, so the authors have to use it in combination with garcinol.

A1: We are grateful for the reviewer’s insightful comment. As requested by the reviewer, we have now provided more representation data of updated figure 2E for the ensured that the Annexin V/7-AAD staining, ensuring that the quality of the dot plots are improved, and that the axes labels are included. We have also indicated data for the early and late apoptosis. In addition, we have also added data for Z-VAD-fmk combined with Garcinol ‘to check whether garcinol-induced cell death is mediated by caspases or not’. Please see the updated figure 2 E.

Please kindly see our revised Results section, Lines 354-391.

3.2. Garcinol significantly inhibits GBM cell viability and oncogenicity through induction of STAT3/5A signaling and enhanced apoptosis

Against the background of recent work demonstrating that garcinol inhibits CSC-like phenotype of human non-small cell lung carcinoma by suppressing the Wnt/β-catenin/STAT3 signaling axis (Huang et al., 2018), we investigated the probable STAT signaling-mediated anti-GBM effect of garcinol (Figure 2A). Firstly, to provide some mechanistic insight, we demonstrated that treatment of U87MG or GBM8401 cells with 2.5 μM or 5 μM garcinol significantly downregulated the expression of p-STAT3, p-STAT5, p-ERK, and p-AKT (Figure 2B), Synchronous with the observed inhibition of STAT3, STAT5 and AKT signaling, garcinol significantly suppressed the viability of GBM4801 and U87MG cells, with 10 μM eliciting 51% or 25% reduced viability of U87MG or GBM8401 cells, respectively, and 40 μM eliciting  94.7% reduction of U87MG and GBM8401 cell viability, indicating a dose-dependent GBM cell killing effect (Figure 2C), and this reduced viability was associated with markedly enhanced  Bax/Bcl-xL apoptotic ratio, as 2.5 μM induced a 1.67-fold (p < 0.05) or  2.7-fold (p < 0.05) increase in U87MG or GBM8401 apoptotic ratio, while  5 μM increased the apoptotic ratio by 2.83-fold (p < 0.001) or 2.92-fold (p < 0.001) in the U87MG or GBM8401 cells, respectively (Figure 2D). In addition, using the Phycoerythrin (PE)-conjugated Annexin V/7- Amino-Actinomycin (7-AAD) staining, we demonstrated that compared to the cell death in the untreated control (U87MG: 0.379%, GBM8401: 0.208%) or 20 mM pan-caspase inhibitor benzyloxycarbonyl-Val-Ala-Asp-fluoromethyl ketone (Z-VAD-FMK)-treated negative control (U87MG: 0.157%, GBM8401: 0.071%), treatment with 2.5 μM garcinol enhanced cell death (U87MG:  18.76%, GBM8401: 17.92%), while 2.5 μM or 5 μM garcinol in the presence of 20 μM Z-VAD-FMK elicited 5.46% or 10.96% apoptosis of the GBM8401 cells, respectively, and 7.97% or 14.11% apoptosis of the U87MG cells (Figure 2E), indicating that the Garcinol-induced cell death was apoptotic and caspase-dependent. Since the highly invasive GBM spreads fast to surrounding brain tissue, thus, contributing to its documented lethality (Omuro & DeAngelis, 2013), we sought to understand if and how garcinol affects this invasive trait. We demonstrated that treatment with 2.5 μM or 5 μM dose-dependently suppressed the migration of the U87MG (~59%, p < 0.01 or 81%, p < 0.001, respectively) and GBM8401 (~48%, p < 0.01 or 76%, p < 0.001, respectively) cells at the 24 h time-point (Figure 2F). Similarly, 2.5 μM or 5 μM garcinol induced a 60% (p < 0.01) or ~80% (p < 0.001) reduction of U87MG invasive capacity, respectively, and 39% (p < 0.01) or 60% (p < 0.001) reduction in number of invaded GBM8401 cells (Figure 2G). Furthermore, in parallel assays to confirm the anticancer role of garcinol, consistent with earlier results, we demonstrated that treatment with 2.5 μM or 5 μM garcinol, significantly suppressed the expression of N-cadherin, vimentin and slug proteins, while conversely upregulating the expression of E-cadherin protein (Figure 2H), thus indicating that garcinol attenuates epithelial-to-mesenchymal transition (EMT) and the metastatic phenotype of GBM cells. Together, these data suggest that garcinol significantly inhibits GBM cell viability and oncogenicity through induction of STAT3/5A and associated signaling with enhanced apoptosis.

Please also kindy see our updated Figure 2 and its legend, Lines 393-410.

Figure 2. Garcinol significantly inhibits GBM cell viability and oncogenicity through induction of STAT3/5A signaling and enhanced apoptosis. (A) Chemical structure of garcinol with molecular formula C38H50O6 and molecular weight 602.80 g/mol. (B) Representative western blot photo-images of the effect of 2.5 μM – 5 μM on the expression of p-STAT3, STAT3, p-STAT5, STAT5, p-ERK, ERK, p-AKT, and AKT proteins in GBM8401 or U87MG cells. (C) Graphical representation of the effect of 2.5 μM – 40 μM on the viability of GBM8401 or U87MG cells. (D) Representative western-blot photo-images showing the effect of 2.5 μM – 5 μM on the expression of Bax and Bcl-xL proteins in GBM8401 or U87MG cells. (E) Flow-cytometry data (upper) and graphical representation (lower) showing the effect of Garcinol, alone or in presence of Z-VAD-FMK, on U87MG or GBM8401 cells co-stained with PE-conjugated Annexin V and 7- AAD, compared with untreated control or Z-VAD-FMK -treated negative control groups. Annexin V-stained Q4 cells are early apoptotic cells, whereas Q2 cells are late stage apoptotic (necrotic) cells. Apoptosis (%), sum of Q4+Q2; CTL, vehicle-treated; Neg CTL, pan-caspase inhibitor benzyloxycarbonyl-Val-Ala-Asp-fluoromethyl ketone (Z-VAD-FMK). Representative photo-images (upper) and graphical representation (lower) of the effect of 2.5 μM or 5 μM on the (F) migration and (G) invasion of GBM8401 or U87MG cells. (H) Representative western-blot photo-images showing the effect of 2.5 μM – 5 μM on the expression of E-cadherin, N-cadherin, vimentin and slug proteins in GBM8401 or U87MG cells. *p<0.05, **p<0.01, ***p<0.001; GAPDH is loading control.

Reviewer 2 Report

I am satisfied that the authors have addressed all my comments made regarding the original manuscript.

Author Response

Q1: Reviewer #2: I am satisfied that the authors have addressed all my comments made regarding the original manuscript.

A1: We thank the reviewer for the time taken to review our work, the critical assessment of our findings and the final satisfactory predisposition towards our revised work. The reviewer’s helpful suggestions has helped us improve the acceptability and appeal of our work.

Reviewer 3 Report

Most of the critics raised during the first revision was answered.

In particular  the diagram of the Figure 6 should be modified: hsa-miR-181 does not directly block STAT3 phosphorylation (because this is not the role of miRNAs). Some intermediate molecule (maybe a protein Kinase) is the direct target of hsa-miR-181. Then lacking of this "Kinase" avoids STAP3 phosphorylation.

Author Response

Q1: Reviewer #3: Most of the critics raised during the first revision was answered.

A1: We thank the reviewer for the time taken to review our work, the assessment of our findings and the helpful suggestions given to help us improve the acceptability and appeal of our work.

Q2: Reviewer #3: In particular the diagram of the Figure 6 should be modified: hsa-miR-181 does not directly block STAT3 phosphorylation (because this is not the role of miRNAs). Some intermediate molecule (maybe a protein Kinase) is the direct target of hsa-miR-181. Then lacking of this "Kinase" avoids STAP3 phosphorylation.

A2: We sincerely thank the reviewer for this insightful comments. Making use of the reviewer’s suggestion, we have modified the graphical abstract in updated Figure 7.  Please kindly see our updated Figure 7 and its legend, Lines 675-678.

Figure 7. Graphical abstract showing how activation of STAT3/5A in GBM is inversely correlated with suppressed hsa-miR-181d expression, and that JAK2-mediated garcinol-induced upregulation of hsa-miR-181d/STAT3 and hsa-miR-181d/5A ratios underlies the anti-GBM-SCs effect of garcinol in STAT3/5A-addicted glioblastoma.

Reviewer 4 Report

The authors have made significantly effort to address my questions and comments. The manuscript they have re-submitted is significantly improved.

There remain however several points of discussion:

The GDC-TCGA glioblastoma (GBM) cohort indeed contains 173 samples- however, the annotation of this cohort mentions only 6 recurrent tumor samples and not 13 as mentioned in the text- how valid are the statistics in these conditions? The authors have added some in vivo data performed on ex situ xenografts (sub-cutaneous U87 xenografts in the flanks of SCID mice), but have not performed any in situ xenografts and survival analyses, which would be required to get an idea of the potential development of resistance mechanisms to their drug. Furthermore, the data remain limited to cell lines (of which U87 remains controversial) and have not been confirmed in primary cultures of GBM; As such, the results remain of very limited relevance for the clinic; The rationale to look at miR181-d as the link between garcinol, STAT3/5 phosphorylation remains obscure – the authors do not explain why they have looked at this miR and how garcinol is supposed to alter the expression of this miR. Theseare major shortcomings of this work; The result section toitle “4. Garcinol increases the expression of hsa-miR181d, which has inhibitory effects on STAT3 and STAT5” is absolutely not supported by the results: mir181d affects the PHOSPHORYLATION of STAT3 &5, not their expression- it also alters the expression of JAK2, ERK and Akt, which aall alter the phosphorylation of these STAT transcription factors- In view of these results, the authors should look at the interaction of mir181d with the genes of these kinases rather than between this miR and STAT genes. In light of 4., the statement in the discussion that “We posit that upon treatment with garcinol, miR-181d binds directly to the coding region of STAT3 or STAT5A mRNA, eliciting STAT3/5A degradation, and consequently reduce the expression and/or activation of STAT3 or STAT5A in the GBM cells. “ is dfinitely not supported by the results. The title of the manuscript remains unsupported by the results and should be modified: “Aberrant Activation of Hsa-miR-181d/STAT3 and Hsa-miR-181d/STAT5A Ratios Mediate the Anticancer Effect of Garcinol in STAT3/5A-Addicted GBM” as The authors have not demonstrated any ‘aberrant activation of Has-miR-181-d/STAT3-5’ in GBM; Such an ‘activation’ certainly does not explain the effect of Garcinol (at best, an inhibiton of the phosphorylation of these STAT proteins, possibly mediated by a down regulation of several kinases (JAK2, ERK, Akt); The authors fail to demonstrate tha GBM are ‘addicted’ to STAT3/5 activation (in particular, given their lack of survival experiments using xenografts to demonstrate the absence of resistance mechanisms.

Author Response

Q1: Reviewer #4: The authors have made significantly effort to address my questions and comments. The manuscript they have re-submitted is significantly improved.

A1: We sincerely thank the reviewer for the time taken to review our work, for the encouraging words, and the important suggestions given to help us improve the quality of our work.

Q2: Reviewer #4: The GDC-TCGA glioblastoma (GBM) cohort indeed contains 173 samples- however, the annotation of this cohort mentions only 6 recurrent tumor samples and not 13 as mentioned in the text- how valid are the statistics in these conditions? (Photo evidence? - xena)

A2: We thank the reviewer for this comments, however, we humbly believe the reviewer is mistaken, as we have double checked our data and do confirm its correctness. We therefore provide a photo-evidence to support our claim. Please kindly see the original screenshot from University of California Santa Cruz Xena Functional Genomics Explorer below.

Q3: Reviewer #4: The authors have added some in vivo data performed on ex situ xenografts (sub-cutaneous U87 xenografts in the flanks of SCID mice), but have not performed any in situ xenografts and survival analyses, which would be required to get an idea of the potential development of resistance mechanisms to their drug.

A1: We sincerely appreciate the reviewer’s comment. We do agree with the reviewer that an orthotopic transplant of GBM cells would have been an ideal experimental design.  We also humbly provide corroborating evidence of the validity and clinical-relevance of the method used in this current study, i.e. sub-cutaneous “flank model”.

William D, et al. Optimized creation of glioblastoma patient derived xenografts for use in preclinical studies. J. Transl. Med 2017; 15 (1):27. doi: 10.1186/s12967-017-1128-5. Husain SR, et al. Complete regression of established human glioblastoma tumor xenograft by interleukin-4 toxin therapy. Cancer Res. 1998; 58:3649-3653. Carlson BL, et al. Establishment, maintenance and in vitro and in vivo applications of primary human glioblastoma multformis (GBM) xenograft models for translational biology studies and drug discovery. Curr Protoc Pharmacol. 2011; 52(14):1-14. doi: 10.1002/0471141755.ph1416s52. Palanichamy K, et al. Clinically relevant brain tumor model and device development for experimental therapeutics. J. Analyt. Oncol. 2015; 4(1):5-12.

Please see the updated figure 5C of the effect of garcinol on the survival of mice GBM xenograft models.

Please kindly see our revised Results section, Lines 506-527.

3.5. Garcinol inhibits tumor growth in GBM mice models through inversely correlated STAT3/5A and hsa-miR-181d expressions

Having shown that treatment with garcinol suppresses the cancer stem cell-like phenotype of U87MG and GBM8401 cells in vitro, to determine the probable suppressive effect of garcinol on the formation and growth of tumor, in vivo, we generated NOD/SCID mice xenograft models derived by inoculation with 1 × 106 U87MG cells subcutaneously in the hind-flank. Mice were randomly placed into control or garcinol treatment group. We demonstrated that treatment with 1 mg/kg garcinol significantly reduced the size of tumors formed in the treated mice, compared to the untreated control group (U87MG: ~7.1-fold smaller, p < 0.001 by week 4) (Figures 5A), without adversely affecting the mice body weight (Figure 5B). We also observed that mice treated with garcinol showed 100% survival as compared to 60% in the control counterparts, over the 4-week treatment period. (Figure 5C). In subsequent experiments using protein lysates derived from the tumors extracted from the untreated and garcinol-treated mice, we demonstrated that compared to the untreated control group, STAT3, pSTAT3, STAT5A, p-STAT5A, Ki-67, and Bcl-xL protein expression levels were concomitantly suppressed, while Bax expression was significantly enhanced (Figure 5D). Moreover, for tumors extracted from the 1 mg/kg garcinol-treated U87MG tumor-bearing mice, compared to the untreated control group, STAT3, or STAT5A mRNA expression were suppressed by 4-fold (p < 0.01) or 3.87-fold (p < 0.01), while miR-181d expression was enhanced by 3.52-fold (p < 0.01) in the U87MG mice treated with 1 mg/kg garcinol (Figure 5E). These findings indicate that garcinol inhibits tumorigenicity and growth of GBM by abrogating STAT3/5A signaling, and upregulating hsa-miR-181d, with concomitant suppression of Ki-67 proliferation index and enhancement of Bax/Bcl-xL apoptotic ratio, in vivo.

Please kindly see our updated Figure 5 and its legend, Lines 529-538.

Figure 5. Garcinol inhibits tumor growth in GBM mice models through inversely correlated STAT3/5A and hsa-miR-181d expressions. Representative image and graph showing the effect of garcinol on the (A) tumor volume and (B) body weight of U87MG-tumor-bearing mice. p-values were determined by 2-way ANOVA. (C) Kaplan-Meier Survival curve.  Mice treated with garcinol showed 100% survival as compared to 60% in the control counterparts, over the 4-week treatment period.  (D) Representative images and histograms of the differential expression of STAT3, pSTAT3, STAT5A, pSTAT5A, Ki-67, Bax, and Bcl-xL proteins level in tumors extracted from mice bearing U87MG cell-derived tumors, treated with or without garcinol. (E) Histograms showing the effect of garcinol treatment on STAT3, STAT5A and miR-181d expression levels in U87MG-tumor-bearing mice. *p<0.05, **p<0.01, ***p<0.001; GAPDH is loading control.

Q4: Reviewer #4: Furthermore, the data remain limited to cell lines (of which U87 remains controversial) and have not been confirmed in primary cultures of GBM; As such, the results remain of very limited relevance for the clinic.

A2: We thank the reviewer for this important comment. As suggested by the reviewer, we have now included results from our primary GBM culture studies data in our revised manuscript. Pleas see the updated figure 6.

 Please kindly see our revised Results section, Lines 539-566.

3.6. Garcinol, akin to Stattic, a selective inhibitor of STAT3/5A activation, inhibits the metastatic and cancer stem cell-like phenotypes of primary GBM culture cells.

Sequel to our data demonstrating that garcinol suppresses the cancer stem cell-like phenotype of U87MG and GBM8401 cells in vitro, and that garcinol also suppresses the formation and growth of tumor in mice GBM models by upregulating hsa-miR-181d expression and inhibiting STAT3/5A activation, we further investigated if indeed these findings could be replicated in GBM primary culture cells. Comparative analyses of the anti-GBM therapeutic effects of garcinol and stattic, a selective inhibitor of STAT3/5A activation and dimerization, revealed that akin to the 25% - 98% reduction in cell viability of the primary GBM culture cells by 2.5 – 40 μM static,  equimolar garcinol treatment dose-dependently elicited a 13% - 96.8% reduced viability of the primary GBM culture cells (μM).  We also observed that concurrent with reduced cell viability, 2.5 and 5 μM garcinol or stattic induced significant downregulation of p-STAT3 (garcinol: 40% and 85%, p < 0.01; stattic: 56% and 89%, p < 0.01) and p-STAT5A (garcinol: 35% and 83%, p < 0.01; stattic: 48% and 74%, p < 0.01) in the primary GBM culture cells (Figure 6B). Furthermore, while treatment with 2.5 and 5 μM stattic attenuated the migration of the primary culture cells by 49.3% (p <0.05) and 88.9% (p<0.001), 2.5 and 5 μM garcinol elicited 48% (p<0.05) and 87.5% (p<0.001) (Figure 6C). Similarly, suppression of cell invasion was demonstrated, with 2.5 and 5 μM stattic causing 64% (p<0.01) and 91% (p<0.001) reduction in migration, respectively, and equimolar garcinol induced 51% (p < 0.01) and 84% (p<0.001), respectively (Figure 6D). We also showed that comparable to stattic, 2.5 and 5 μM garcinol reduced the number of colonies formed by 50.7% (p<0.01) and 91% (p<0.001), respectively (Figure 6E). Moreover, akin to stattic, 5 μM garcinol reduced the number of primary GBM culture tumorspheres formed by 91.7% (p<0.001) and the tumorsphere sizes by ~90% (p<0.001) (Figure 6F). These findings not only validate the replicability of garcinol anti-GBM effect in vitro, in vivo and ex vivo using primary GBM culture cells, and enhance clinical or translational relevance, but also demonstrate, at least in part, that akin to stattic, garcinol inhibits the metastatic and stemness phenotypes of primary GBM culture cells by inhibiting the activation of STAT3/5A, which is consistent with results in Figures 4 and 5 showing that garcinol inhibits metastasis, cancer stemness and tumor growth through enhanced hsa-miR-181d/STAT3 or hsa-miR-181d/STAT5A.

Also see our newly included Figure 6 and its legend, Lines 568-579.

Figure 6. Garcinol, akin to Stattic, a selective inhibitor of STAT3/5A activation, inhibits the metastatic and cancer stem cell-like phenotypes of GBM primary culture cells. (A) Graphical representation of the effect of 2.5 μM – 40 μM on the viability of GBM primary culture cells. (B) Representative western-blot photo-images comparing the effect of stattic and garcinol on the expression level of p-STAT3, STAT3, p-STAT5, and STAT5 proteins in primary culture cells. (C) Representative images (left) and histograms (right) comparing the effect of 2.5 and 5 μM stattic or garcinol on the migration of U87MG cells over 24 h, as determined by wound-healing assay.  Representative images (upper) and histograms (lower) comparing the effect of 2.5 and 5 M stattic or garcinol on the (D) invasion and (E) colony formation capacity of GBM primary culture cells. (F) Representative photo-images (left) and histograms comparing the effect of 5 μM stattic and garcinol on the number and size of tumorspheres formed by the GBM primary culture cells.  *p<0.05, **p<0.01, ***p<0.001; GAPDH is loading control.

Also kindly see our revised Materials and Methods section, Lines 150-165.

2.3. Cell lines and Primary Culture Cell Culture

The human U-87 MG (ATCC® HTB-14™) (ATCC, Manassas, VA) and GBM8401 GBM cell lines used in the study were purchased from (Bioresource Collection Research Center, Hsinchu, Taiwan). The cell lines were cultured in Gibco DMEM (Cat. No. 11965175, Thermo Fisher Scientific, Inc. Waltham, MA, USA), supplemented with 10% fetal bovine serum (FBS) and 1% penicillin/streptomycin (Invitrogen, Life Technologies, Carlsbad, CA, USA) and incubated in 5% humidified CO2 incubator at 37oC. The cells were sub-cultured when they reached 80-90% confluency and the media changed every 48-72h. Patient-derived CD133+ GBM spheres was kindly provided by our collaborator Dr. Alexander T.H. Wu at Taipei Medical University. In brief, the patient-derived GBM cells were first sorted using established flow cytometric method. Once CD133+ cells were sorted, they were expanded in advanced DMEM/F12 (Gibco) mixed with Neurobasal TM-A medium (Gibco) (1:1) supplemented with B-27 (1×), FGF (20 ng/mL) and EGF (20 ng/mL); culturing under these conditions maintained CD133+ cell population and stemness (as well as TMZ-resistant), the tumor-initiating ability was demonstrated in vivo as described previously  (Wei et al., 2016).

The reference has been added in the line 838-841.

Wei, L., Su, Y. K., Lin, C. M., Chao, T. Y., Huang, S. P., Huynh, T. T., Jan, H. J., Whang-Peng J., Chiou J. F., Wu A. T., Hsiao M. (2016) Preclinical investigation of ibrutinib, a Bruton's kinase tyrosine (Btk) inhibitor, in suppressing glioma tumorigenesis and stem cell phenotypes. Oncotarget. Oct 25;7(43):69961-69975.

Q5: Reviewer #4: The rationale to look at miR181-d as the link between garcinol, STAT3/5 phosphorylation remains obscure – the authors do not explain why they have looked at this miR and how garcinol is supposed to alter the expression of this miR. Theseare major shortcomings of this work.

A4: We thank the reviewer for this comment. We have tried to address the reviewer’s lack of clarity on the rationale for the study of garcinol’s effect on miR-181d, and STAT3/5 activation in our revised work. Please kindly see our revised Introduction section, Lines 94-132.

Over the last 2 decades, there has been increase in the documentation of the critical roles of microRNA (miRNAs/miRs) in the malignantization and progression of many human malignancies, including GBM; by their ability to hamper or facilitate cancer initiation, miRNAs have emerged as therapeutically-relevant actionable biomolecules for anticancer therapy (Bhaskaran et al., 2019; Floyd & Purow, 2014). One such miRNA family is miRNA-181, with varied expression of its isoforms (miRNA-181a/b/c/d) being touted as independent predictors of clinical outcome in patients with different cancer types (Pop-Bica et al., 2018). A notable feature of treatment-resistant GBM is the enzyme O6-methylguanin-DNA methyltransferase (MGMT), a DNA-repair protein that removes alkyl group from the O6 position of alkyl groups, and consequently diminish the curative effects of chemotherapeutic agents (Hegi et al., 2005), conversely, the silencing of MGMT through promoter methylation is associated with prolonged OS and disease-free survival (DFS) in patients with GBM (Bell et al., 2018). Interestingly, recent studies show that MGMT activity is inversely correlated with the expression of microRNA-181d/miRNA-181d; as higher expression of miRNA-181d is positively correlated with improved OS in patients with GBM (Zhang et al., 2012). There is also accruing evidence that the prognostic implications of altered miRNA expression, is connected to their roles in modulation of stemness signaling such as Notch, Hedgehog, and JAK/STAT3 (Floyd & Purow, 2014; F. Yang et al., 2017), and the consequent acquisition of stem cell-like traits by GBM cells; Thus, the present study’s rationale for exploring the actionability of miRNA-181d in the context of STAT signaling in GBM.

Due to the accruing adverse reactions to most chemotherapeutic agents, phytochemicals and nutraceuticals have garnered interest as possible safe alternatives or adjuvants in cancer treatment (Saadat & Gupta, 2012). Building on previous works by our team demonstrating that Garcinol inhibits CSCs-like phenotype by suppressing the Wnt/β-catenin/STAT3 signaling axis in human non-small cell lung carcinoma (Huang et al., 2018), we now examined the probable effect of Garcinol on GBM stem cells (GBM-SCs), and the implication of same for sensitivity to conventional chemotherapy and better prognosis. Garcinol, a major bioactive constituent of the fruit Garcinia indica, has widely documented antioxidant and anticancer effects, and is chemically similar in structure to the well-known curcumin (Ashad et al., 2017; Saadat & Gupta, 2012). In fact, Hong et al. (Hong et al., 2007) demonstrated that garcinol significantly inhibited the growth of the colon cancer cells, and importantly, provided evidence that unlike conventional chemotherapeutics, garcinol preferentially targets cancerous cells; thus inhibiting cancer growth without adversely affecting the neighboring ‘normal’ non-cancerous cells (Hong et al., 2007). In breast cancer, garcinol was shown to inhibit STAT3-NF-kB signaling, resulting in reduced invasiveness, in vitro, and significantly attenuated tumor growth in NOD-SCID mice (Ahmad et al., 2012, 2010); thus, building a case for the preclinical investigation of the probable anti-GBM effect of garcinol in this stance. This present study, for the first time, to the best of our knowledge, investigated and documents the effect of garcinol on GBM, consistent with current theme-relevant knowledge, especially in the context of the therapeutic effects of garcinol alone or in synergism with conventional anticancer treatment modalities on GBM-SCs through the mediation of STAT3/5A signaling and miRNA-181d.

Also kindly see our revised discussion and conclusion section, Lines 617-673.

In the context of the GBM SCs-phenotypes including enhanced proliferation, oncogenicity, therapy resistance, recurrence and poor prognosis, we further demonstrated that garcinol significantly inhibited GBM8401, U87MG, and GBM primary culture cell viability in a dose-dependent manner, as well as suppresses the cell invasive and migratory potentials, thus, demonstrating the anti-proliferative and anti-metastatic efficacy of garcinol in GBM. This is consistent with documented robust growth-inhibitory effects of garcinol demonstrated against colon cancer and immortalized intestinal cells (Hong et al., 2007). Since STAT3 and STAT5A are implicated in the maintenance of the stem cell-like characteristics of GBM, we examined garcinol’s inhibitory effect on the GBM-SCs profile. Interestingly, low dose garcinol (≤ 5 μM) deregulated STAT3/5A signaling with repressed AKT and ERK crosstalk, and this was sufficient to significantly impeded GBM cell migration, invasion, clonogenicity, and tumorsphere formation, with associated increase in apoptotic index and nuclear expression of SOX2 and OCT4. Our findings are corroborated by recent evidence that STAT3 and STAT5 are constitutively activated in malignant cells, and that their persistent activation facilitates cancer development and progression by altering downstream gene expression through epigenetic modification, EMT induction, oncogenic modification of the tumor microenvironment, and enhancing of CSCs self-renewal and differentiation (Yuan et al., 2015), as well as evidence implicating high ERK1/2 activity in the acquisition and maintenance of SOX2-expressing Glioma stem cells (Kwon et al., 2017).

This study also demonstrated for the first time to the best of our knowledge, that garcinol-induced suppression of STAT3 and STAT5A is associated with significant upregulation of hsa-miR-181d expression, in vitro, ex vivo and in vivo; interestingly we also showed direct interaction between hsa-miR-181d and STAT3 or STAT5A protein. We posit that upon treatment with garcinol, miR-181d canonically represses the activation/phosphorylation of STAT3/5A in a JAK2-mediated manner, or as also documented herein, miR-181d non-canonically binds directly to the coding region of STAT3 or STAT5A mRNA, eliciting STAT3/5A degradation, and consequently impair activation of STAT3 or STAT5A in the GBM cells. This demonstrated tumor suppressor role of hsa-miR-181d is consistent with findings showing that overexpression of miR-181d significantly suppressed esophageal squamous cell carcinoma (ESCC) by downregulating Derlin-1, inhibiting cancerous cell proliferation, migration and cell cycle progression in vitro, as well as inhibiting tumorigenicity in vivo (Li D et al., 2016), as well as in glioma samples and cell lines, where ectopic expression of miR-181d suppressed proliferation and induced cell cycle arrest and apoptosis by targeting K-ras and Bcl-2 (Wang et al., 2012). This demonstrated garcinol-modulated miR-181d/STAT3/5A signaling axis is of therapeutic relevance, considering that well documented role of the JAK-STAT signaling pathway, and more particularly its molecular effectors namely STAT3 and STAT5 which act as a point of convergence for several signaling pathways in cancerous cells and oncogenic processes (Luo and Balko, 2019). It is worth mentioning however, that while we cannot fully explain how garcinol induced hsa-miR-181d inhibited the activation of STAT3 and STAT5, our data finds some corroboration in increasingly documented role of miRs in the (de)activation of the JAK-STAT signaling (Zhuang G et al., 2012, Lam et al, 2013, Liu X et al., 2018), and of particular interest is miR-204 which similarly had very insignificant effect on total STAT3 expression, but impaired STAT3 phosphorylation, consequently inducing cancerous cell apoptosis and suppressed cell proliferation, migration in vitro and tumor growth in vivo (Liu X et al., 2018). Moreover, recent report that the Kaposi’s sarcoma-associated herpes virus (KSHV) miRNAs impair the activation/phosphorylation of STAT3/5, and inhibit the activation of STAT3-dependent reporter upon IL6-treatment, also lend credence to our finding that hsa-miR-181d may bind directly with STAT3/5A and impair activation of the later (Ramlingam and Ziegelbauer, 2017).

Conclusion

Taken together, as depicted in our graphical abstract in Figure 7, the present study provides evidence that the constitutive activation of STAT3/5A in GBM is inversely correlated with suppressed hsa-miR-181d expression, and that JAK2-mediated garcinol-induced upregulation of hsa-miR-181d/STAT3 and hsa-miR-181d/5A ratios underlies the anti-GBM-SCs effect of garcinol in STAT3/5A-addicted GBM. These findings are of translational relevance as they highlight the therapeutic efficacy of a relatively novel small molecule inhibitor of STAT3/5A in the highly invasive and often therapy resistant GBM. In addition, findings documented in the present pilot study form a basis for further large cohort exploration of the preclinical feasibility and subsequent clinical applicability of garcinol-modulated hsa-miR-181d/STAT ratio as a therapeutic strategy in GBM.

The reference has been added in the line 815-816.

Ramalingam, D., & Ziegelbauer, J. M. (2017). Viral microRNAs Target a Gene Network, Inhibit STAT Activation, and Suppress Interferon Responses. Scientific reports, 7, 40813. doi:10.1038/srep40813

Q6: Reviewer #4: The result section toitle “4. Garcinol increases the expression of hsa-miR181d, which has inhibitory effects on STAT3 and STAT5” is absolutely not supported by the results: mir181d affects the PHOSPHORYLATION of STAT3 &5, not their expression- it also alters the expression of JAK2, ERK and Akt, which aall alter the phosphorylation of these STAT transcription factors- In view of these results, the authors should look at the interaction of mir181d with the genes of these kinases rather than between this miR and STAT genes.

A5: We thank the reviewer for this insightful comment. As suggested by the reviewer, we have now modified the sub-title in question. We have also incorporated the reviewer’s suggestion on the likely interaction between mir181d and the JAK2 kinase in our revised manuscript. Please kindly see our revised Results section, Lines 447-491.

3.4. Garcinol increases the expression of hsa-miR181d, which has inhibitory effects on STAT3 and STAT5 activation

Having established that garcinol impairs STAT3 and STAT5A activation, we probed for likely modulators and/or mediators of the interaction between garcinol and the STAT proteins. Hsa-miR-181d shown in Figure 4A, has been implicated in the worse OS of patients with GBM (Zhang et al., 2012). Consistent with this, using the Schrodinger PyMOL 2.3 molecular docking and visualization software (http://pymol.org) we demonstrated that hsa-miR-181d interacts with and binds directly to STAT3 (docking score = -254.49, ligand root mean square deviation (RMSD) = 195.10 Å) or STAT5A (docking score = -234.19, ligand RMSD = 143.04 Å), complementing earlier prediction that hsa-miR-181d binds with STAT3 with a mirSVR or PhastCons score of -0.26 or 0.69, respectively, while it binds with STAT5A with a mirSVR  or PhastCons score of -0.21 or 0.49, respectively (Figure 4A). Concomitantly, as shown in Figure 4A, results of our nucleotide complementarity analysis indicate that the 5’-UTR (untranslated region) of hsa-miR-181d binds to the 3’ UTR of JAK2 with a mirSVR or PhastCons score of -0.79 or 0.61, respectively.  Where the mirSVR shows the likelihood of hsa-miR-181d down-regulating the target mRNA STAT3 and STAT5A based on the sequence and structure features in the miRNA/mRNA predicted target sites. Moreover, the PhastCons score shows the likelihood that the predicted miRNA/mRNA binding nucleotides are conserved. In concordance, our qRT-PCR analysis of 5 μM garcinol-treated U87MG and GBM8401 cells showed that garcinol significantly induced higher expression of miR-181d in the U87MG (2.7-fold, p < 0.01) and GBM8401 (2.1-fold, p < 0.01) cells (Figure 4B). Furthermore, having implicated STAT3/5A in enhanced migration and invasiveness of U87MG or GBM8401 cells, we examined the probable effect of hsa-miR-181d on these metastatic phenotypes of GBM cells. Using the scratch wound-healing assay, we demonstrated that compared to the untreated control or syn-mir-treated cells, treatment with mir-181d inhibitor significantly enhanced the ability of the U87MG cells to migrate (4.64-fold, p<0.01), while treatment with the mir-181d-mimic elicited marked attenuation of migration (3.80-fold, p<0.01) (Figure 4C), which is reminiscent of suppressed migration induced by 5 M garcinol (4.17-fold, p<0.01) earlier.  Similarly, while treatment with mir-181d inhibitor significantly enhanced the invasiveness of the U87MG cells (4.63-fold, p<0.01), treatment with the mir-181d-mimic elicited profound suppression of invasion (4.15-fold, p<0.01) (Figures 4D), and this was akin to the effect of 5 μM garcinol (4.22-fold, p<0.01). To confirm a direct relationship between the STAT proteins and miR-181d expression, western blot analysis was done comparing samples exposed to mir-181d inhibitor, mir-181d-mimic, or mir-181d inhibitor/garcinol combination. The results showed that mir-181d inhibitor significantly enhanced the expression of p-STAT3, p-STAT5, N-cadherin and vimentin proteins, but suppressed E-cadherin protein expression compared to the control group, while for the mir-181d-mimic-treated cells, the p-STAT3, p-STAT5, N-cadherin, and vimentin protein expression levels were significantly lower, but E-cadherin was upregulated; For cells incubated with mir-181d inhibitor and 5 μM garcinol concomitantly, p-STAT3 and p-STAT5 protein expression levels were markedly higher than in the mir-181d-mimic group but lower than the mir-181d-inhibitor group (Figure 4E). Concomitantly, we observed that compared with the control group, syn-mir-treated cells, or even mir-181d inhibitor-treated cells, treatment with mir-181d-mimic markedly suppressed the expression of JAK2 protein (2.22-fold, p<0.01), akin to the effect elicited by treatment with concurrently with mir-181d inhibitor and 5 μM garcinol (2.94-fold, p<0.01), which is consistent with the results above and suggestive of a miR-181d-mediated JAK2-modulated phosphorylation of STAT3/5A. These results indicate that garcinol can activate mir-181d activity which suppresses JAK2-modulated STAT3/5A activation.

Also kindly see our revised Discussion section, Lines 635-662.

This study also demonstrated for the first time to the best of our knowledge, that garcinol-induced suppression of STAT3 and STAT5A is associated with significant upregulation of hsa-miR-181d expression, in vitro, ex vivo and in vivo; interestingly we also showed direct interaction between hsa-miR-181d and STAT3 or STAT5A protein. We posit that upon treatment with garcinol, miR-181d canonically represses the activation/phosphorylation of STAT3/5A in a JAK2-mediated manner, or as also documented herein, miR-181d non-canonically binds directly to the coding region of STAT3 or STAT5A mRNA, eliciting STAT3/5A degradation, and consequently impair activation of STAT3 or STAT5A in the GBM cells. This demonstrated tumor suppressor role of hsa-miR-181d is consistent with findings showing that overexpression of miR-181d significantly suppressed esophageal squamous cell carcinoma (ESCC) by downregulating Derlin-1, inhibiting cancerous cell proliferation, migration and cell cycle progression in vitro, as well as inhibiting tumorigenicity in vivo (Li D et al., 2016), as well as in glioma samples and cell lines, where ectopic expression of miR-181d suppressed proliferation and induced cell cycle arrest and apoptosis by targeting K-ras and Bcl-2 (Wang et al., 2012). This demonstrated garcinol-modulated miR-181d/STAT3/5A signaling axis is of therapeutic relevance, considering that well documented role of the JAK-STAT signaling pathway, and more particularly its molecular effectors namely STAT3 and STAT5 which act as a point of convergence for several signaling pathways in cancerous cells and oncogenic processes (Luo and Balko, 2019). It is worth mentioning however, that while we cannot fully explain how garcinol induced hsa-miR-181d inhibited the activation of STAT3 and STAT5, our data finds some corroboration in increasingly documented role of miRs in the (de)activation of the JAK-STAT signaling (Zhuang G et al., 2012, Lam et al, 2013, Liu X et al., 2018), and of particular interest is miR-204 which similarly had very insignificant effect on total STAT3 expression, but impaired STAT3 phosphorylation, consequently inducing cancerous cell apoptosis and suppressed cell proliferation, migration in vitro and tumor growth in vivo (Liu X et al., 2018). Moreover, recent report that the Kaposi’s sarcoma-associated herpes virus (KSHV) miRNAs impair the activation/phosphorylation of STAT3/5, and inhibit the activation of STAT3-dependent reporter upon IL6-treatment, also lend credence to our finding that hsa-miR-181d may bind directly with STAT3/5A and impair activation of the later (Ramlingam and Ziegelbauer, 2017).

Also kindly see our revised Conclusion section, Lines 663-673.

Conclusion

Taken together, as depicted in our graphical abstract in Figure 7, the present study provides evidence that the constitutive activation of STAT3/5A in GBM is inversely correlated with suppressed hsa-miR-181d expression, and that JAK2-mediated garcinol-induced upregulation of hsa-miR-181d/STAT3 and hsa-miR-181d/5A ratios underlies the anti-GBM-SCs effect of garcinol in STAT3/5A-addicted GBM. These findings are of translational relevance as they highlight the therapeutic efficacy of a relatively novel small molecule inhibitor of STAT3/5A in the highly invasive and often therapy resistant GBM.

Q7: Reviewer #4: In light of 4., the statement in the discussion that “We posit that upon treatment with garcinol, miR-181d binds directly to the coding region of STAT3 or STAT5A mRNA, eliciting STAT3/5A degradation, and consequently reduce the expression and/or activation of STAT3 or STAT5A in the GBM cells. “ is dfinitely not supported by the results.

A7: We thank the reviewer for this comment. We have however rephrased the statement to address the reviewer’s concern. Please kindly see our revised Discussion section, Lines 635-662.

This study also demonstrated for the first time to the best of our knowledge, that garcinol-induced suppression of STAT3 and STAT5A is associated with significant upregulation of hsa-miR-181d expression, in vitro, ex vivo and in vivo; interestingly we also showed direct interaction between hsa-miR-181d and STAT3 or STAT5A protein. We posit that upon treatment with garcinol, miR-181d canonically represses the activation/phosphorylation of STAT3/5A in a JAK2-mediated manner, or as also documented herein, miR-181d non-canonically binds directly to the coding region of STAT3 or STAT5A mRNA, eliciting STAT3/5A degradation, and consequently impair activation of STAT3 or STAT5A in the GBM cells. This demonstrated tumor suppressor role of hsa-miR-181d is consistent with findings showing that overexpression of miR-181d significantly suppressed esophageal squamous cell carcinoma (ESCC) by downregulating Derlin-1, inhibiting cancerous cell proliferation, migration and cell cycle progression in vitro, as well as inhibiting tumorigenicity in vivo (Li D et al., 2016), as well as in glioma samples and cell lines, where ectopic expression of miR-181d suppressed proliferation and induced cell cycle arrest and apoptosis by targeting K-ras and Bcl-2 (Wang et al., 2012). This demonstrated garcinol-modulated miR-181d/STAT3/5A signaling axis is of therapeutic relevance, considering that well documented role of the JAK-STAT signaling pathway, and more particularly its molecular effectors namely STAT3 and STAT5 which act as a point of convergence for several signaling pathways in cancerous cells and oncogenic processes (Luo and Balko, 2019). It is worth mentioning however, that while we cannot fully explain how garcinol induced hsa-miR-181d inhibited the activation of STAT3 and STAT5, our data finds some corroboration in increasingly documented role of miRs in the (de)activation of the JAK-STAT signaling (Zhuang G et al., 2012, Lam et al, 2013, Liu X et al., 2018), and of particular interest is miR-204 which similarly had very insignificant effect on total STAT3 expression, but impaired STAT3 phosphorylation, consequently inducing cancerous cell apoptosis and suppressed cell proliferation, migration in vitro and tumor growth in vivo (Liu X et al., 2018). Moreover, recent report that the Kaposi’s sarcoma-associated herpes virus (KSHV) miRNAs impair the activation/phosphorylation of STAT3/5, and inhibit the activation of STAT3-dependent reporter upon IL6-treatment, also lend credence to our finding that hsa-miR-181d may bind directly with STAT3/5A and impair activation of the later (Ramlingam and Ziegelbauer, 2017).

Q8: Reviewer #4: The title of the manuscript remains unsupported by the results and should be modified: “Aberrant Activation of Hsa-miR-181d/STAT3 and Hsa-miR-181d/STAT5A Ratios Mediate the Anticancer Effect of Garcinol in STAT3/5A-Addicted GBM” as The authors have not demonstrated any ‘aberrant activation of Has-miR-181-d/STAT3-5’ in GBM; Such an ‘activation’ certainly does not explain the effect of Garcinol (at best, an inhibiton of the phosphorylation of these STAT proteins, possibly mediated by a down regulation of several kinases (JAK2, ERK, Akt).

A8: We thank the reviewer for this comment. To allay the reviewer’s concern, we have now modified the title of the manuscript. Please kindly see title page, Lines 1-4.

Enhanced Hsa-miR-181d/p-STAT3 and Hsa-miR-181d/p-STAT5A Ratios Mediate the Anticancer Effect of Garcinol in STAT3/5A-addicted Glioblastoma

Q9: Reviewer #4: The authors fail to demonstrate tha GBM are ‘addicted’ to STAT3/5 activation (in particular, given their lack of survival experiments using xenografts to demonstrate the absence of resistance mechanisms.

A9: We thank the reviewer for this important comments. We have made included the ‘survival experiments using xenografts to demonstrate the absence of resistance mechanisms’ suggested by the reviewer.

Please see the updated figure 5C of the effect of garcinol on the survival of mice GBM xenograft models.

Please kindly see our revised Results section, Lines 506-527.

3.5. Garcinol inhibits tumor growth in GBM mice models through inversely correlated STAT3/5A and hsa-miR-181d expressions

Having shown that treatment with garcinol suppresses the cancer stem cell-like phenotype of U87MG and GBM8401 cells in vitro, to determine the probable suppressive effect of garcinol on the formation and growth of tumor, in vivo, we generated NOD/SCID mice xenograft models derived by inoculation with 1 × 106 U87MG cells subcutaneously in the hind-flank. Mice were randomly placed into control or garcinol treatment group. We demonstrated that treatment with 1 mg/kg garcinol significantly reduced the size of tumors formed in the treated mice, compared to the untreated control group (U87MG: ~7.1-fold smaller, p < 0.001 by week 4) (Figures 5A), without adversely affecting the mice body weight (Figure 5B). We also observed that mice treated with garcinol showed 100% survival as compared to 60% in the control counterparts, over the 4-week treatment period. (Figure 5C). In subsequent experiments using protein lysates derived from the tumors extracted from the untreated and garcinol-treated mice, we demonstrated that compared to the untreated control group, STAT3, pSTAT3, STAT5A, p-STAT5A, Ki-67, and Bcl-xL protein expression levels were concomitantly suppressed, while Bax expression was significantly enhanced (Figure 5D). Moreover, for tumors extracted from the 1 mg/kg garcinol-treated U87MG tumor-bearing mice, compared to the untreated control group, STAT3, or STAT5A mRNA expression were suppressed by 4-fold (p < 0.01) or 3.87-fold (p < 0.01), while miR-181d expression was enhanced by 3.52-fold (p < 0.01) in the U87MG mice treated with 1 mg/kg garcinol (Figure 5E). These findings indicate that garcinol inhibits tumorigenicity and growth of GBM by abrogating STAT3/5A signaling, and upregulating hsa-miR-181d, with concomitant suppression of Ki-67 proliferation index and enhancement of Bax/Bcl-xL apoptotic ratio, in vivo.

Please kindly see our updated Figure 5 and its legend, Lines 529-538.

Figure 5. Garcinol inhibits tumor growth in GBM mice models through inversely correlated STAT3/5A and hsa-miR-181d expressions. Representative image and graph showing the effect of garcinol on the (A) tumor volume and (B) body weight of U87MG-tumor-bearing mice. p-values were determined by 2-way ANOVA. (C) Kaplan-Meier Survival curve.  Mice treated with garcinol showed 100% survival as compared to 60% in the control counterparts, over the 4-week treatment period.  (D) Representative images and histograms of the differential expression of STAT3, pSTAT3, STAT5A, pSTAT5A, Ki-67, Bax, and Bcl-xL proteins level in tumors extracted from mice bearing U87MG cell-derived tumors, treated with or without garcinol. (E) Histograms showing the effect of garcinol treatment on STAT3, STAT5A and miR-181d expression levels in U87MG-tumor-bearing mice. *p<0.05, **p<0.01, ***p<0.001; GAPDH is loading control.

This manuscript is a resubmission of an earlier submission. The following is a list of the peer review reports and author responses from that submission.

Round 1

Reviewer 1 Report

In this paper Liu et al. described the impact of STAT3 and STAT5A in GBM and the capacity of garcinol to dampen STAT3 and STAT5A activation. For this, they provide pertinent clinical data and biological experiments that should be completed.

Major points

- The authors claimed that garcinol enhance apoptosis. However they only show a modification of the Bax/Bcl-XL ratio, which is not sufficient. The authors must perform annexin V/7-AAD (or annexin V/IP) experiments to prove that cells died by apoptosis. The best should be also to perform experiments with a caspase inhibitor, to be sure that it is a caspase-dependent cell death. Without these experiments, the authors cannot conclude that garcinol enhances apotposis.

- The authors performed woundhealing and invasion experiments. This should be completed with the analysis of migration-associated genes expression such as ZEB1, TWIST, ZEB2, SLUG, ….

- In figure 3, the expression of SOX2 and OCT4 was analysed by IF. This does not constitute a quantitative method as with garcinol tumorspher are smallest so it is logical to have less satining. The analysis of OCT4 and SOX2 expression should be performed by western blot or QPCR. Idem for STAT3 and STAT5A.

- The authors conclude that hsa-miR-181d is an inhibitor of STAT3 and STAT5A expression. However, only pSTAT3 and pSTAT5A (and not total STAT3 and STAT5A expression) were modified by the miR181d inhibitor or mimic. This leads to conclude that the effect of miR-181d mostly rely on an upstream target. The authors should study the capacity of miR-181d to decrease the expression of STAT3 and STAT5A phosphorylation inductors, such as JAK1, JAK2, IL-6R, … This will be more consistent as the effects observed were both on STAT3 and STAT5A.

Minor points

- In M&M can the authors explained how they count tumorpheres (size, number, …).

- In M&M can the authors explained how percentages of migrations were calculated.

- In M&M can the authors explained how Q-scores for IHC were calculated.

- In some figure legend, the time of exposure to treatments should be added.

- In paragraph 3.3, the authors said that they demostrated (line 358). It is not true as it is an analysis of the alignement of 2 sequences.

- In the discussion (line 430) the authors said they examined garcinol’s therapeutic effect … It is not true as they only provide in vitro experiments.

Author Response

Answers to the comments:

Point-by-point responses to reviewer’s comments:

We would like to thank the reviewer for the thorough reading of our manuscript as well as their valuable comments. We have followed their comments closely and feel that they have further improved the readability and appeal of our work, as well as strengthened the manuscript. Below are our point-by-point responses.

Q1: Reviewer #1: In this paper Liu et al. described the impact of STAT3 and STAT5A in GBM and the capacity of garcinol to dampen STAT3 and STAT5A activation. For this, they provide pertinent clinical data and biological experiments that should be completed.

A1: We sincerely thank the reviewer for the time taken to review our work and for the important suggestion made to improve the quality of our work.

Q2: Reviewer #1: The authors claimed that garcinol enhance apoptosis. However, they only show a modification of the Bax/Bcl-XL ratio, which is not sufficient. The authors must perform annexin V/7-AAD (or annexin V/IP) experiments to prove that cells died by apoptosis. The best should be also to perform experiments with a caspase inhibitor, to be sure that it is a caspase-dependent cell death. Without these experiments, the authors cannot conclude that garcinol enhances apotposis.

A2: We appreciate the reviewer’s insightful comment. We have now included the requested data of V/7-AAD (or annexin V/IP) experiments in our revised manuscript. Please kindly see our revised Results section, lines 345-381.

3.2. Garcinol significantly inhibits GBM cell viability and oncogenicity through induction of STAT3/5A signaling and enhanced apoptosis

Against the background of recent work demonstrating that garcinol inhibits CSCs-like phenotype of human non-small cell lung carcinoma by suppressing the Wnt/β-catenin/STAT3 signaling axis (Huang et al., 2018), we investigated the probable STAT signaling-mediated anti-GBM effect of garcinol (Figure 2A). Firstly, to provide some mechanistic insight, we demonstrated that treatment of U87MG or GBM8401 cells with  2.5 μM or 5 μM garcinol significantly downregulated the expression of p-STAT3, p-STAT5, p-ERK, and p-AKT (Figure 2B), Synchronous with the observed inhibition of STAT3, STAT5 and AKT signaling, garcinol significantly suppressed the viability of GBM4801 and U87MG cells, with 10 μM eliciting 51% or 25% reduced viability of U87MG or GBM8401 cells, respectively, and 40 μM eliciting  94.7% reduction of U87MG and GBM8401 cell viability, indicating a dose-dependent GBM cell killing effect (Figure 2C), and this reduced viability was associated with markedly enhanced  Bax/Bcl-xL apoptotic ratio, as 2.5 μM induced a 1.67-fold (p < 0.05) or  2.7-fold (p < 0.05) increase in U87MG or GBM8401 apoptotic ratio, while  5 μM increased the apoptotic ratio by 2.83-fold (p < 0.001) or 2.92-fold (p < 0.001) in the U87MG or GBM8401 cells, respectively (Figure 2D). In addition, using the Phycoerythrin (PE)-conjugated Annexin V/7- Amino-Actinomycin (7-AAD) staining, we demonstrated that compared to the cell death in the untreated control (U87MG: 1.6%, GBM8401: 4.0%) or 20 mM pan-caspase inhibitor benzyloxycarbonyl-Val-Ala-Asp-fluoromethyl ketone (Z-VAD-FMK)-treated negative control (U87MG: 3.2%, GBM8401: 10.0%), treatment with Garcinol enhanced U87MG cell death (2.5 mM:  10.5% or 5 mM: 41.2%), while 2.5 mM or 5 mM garcinol elicited 15.1% or 32.1% apoptosis of the GBM8401 cells, respectively (Figure 2E), indicating that the Garcinol-induced cell death was apoptotic and caspase-dependent. Since the highly invasive GBM spreads fast to surrounding brain tissue, thus, contributing to its documented lethality (Omuro & DeAngelis, 2013), we sought to understand if and how garcinol affects this invasive trait. We demonstrated that treatment with 2.5 μM or 5 μM dose-dependently suppressed the migration of the U87MG (~59%, p < 0.01 or 81%, p < 0.001, respectively) and GBM8401 (~48%, p < 0.01 or 76%, p < 0.001, respectively) cells at the 24 h time-point (Figure 2F). Similarly, 2.5 μM or 5 μM garcinol induced a 60% (p < 0.01) or ~80% (p < 0.001) reduction of U87MG invasive capacity, respectively, and 39% (p < 0.01) or 60% (p < 0.001) reduction in number of invaded GBM8401 cells (Figure 2G). Furthermore, in parallel assays to confirm the anticancer role of Garcinol, consistent with earlier results, we demonstrated that treatment with 2.5 μM or 5 μM Garcinol, significantly suppressed the expression of N-cadherin, vimentin and slug proteins, while conversely upregulating the expression of E-cadherin protein (Figure 2H),thus indicating that Garcinol attenuates epithelial-to-mesenchymal transition and the metastatic phenotype of GBM cells. Together, these data suggest that garcinol significantly inhibits GBM cell viability and oncogenicity through induction of STAT3/5A and associated signaling with enhanced apoptosis.

Please also kindly see our updated Figure 2 and its legend, lines 383-399.

Figure 2. Garcinol significantly inhibits GBM cell viability and oncogenicity through induction of STAT3/5A signaling and enhanced apoptosis. (A) Chemical structure of garcinol with molecular formula C38H50O6 and molecular weight 602.80 g/mol. (B) Representative western blot photo-images of the effect of 2.5 μM – 5 μM on the expression of p-STAT3, STAT3, p-STAT5, STAT5, p-ERK, ERK, p-AKT, and AKT proteins in GBM8401 or U87MG cells. (C) Graphical representation of the effect of 2.5 μM – 40 μM on the viability of GBM8401 or U87MG cells. (D) Representative western-blot photo-images showing the effect of 2.5 μM – 5 μM on the expression of Bax and Bcl-xL proteins in GBM8401 or U87MG cells. (E) Flow-cytometry data (upper) and graphical representation (lower) showing the effect of Garcinol on U87MG cells co-stained with PE-conjugated Annexin V and 7- AAD, compared with untreated or Z-VAD-FMK -treated negative control groups. Annexin V-stained Q4 cells are early apoptotic cells, whereas Q2 cells are late stage apoptotic (necrotic) cells. Apoptosis (%), sum of Q4+Q2; CTL, vehicle-treated; Neg CTL, pan-caspase inhibitor benzyloxycarbonyl-Val-Ala-Asp-fluoromethyl ketone (Z-VAD-FMK). Representative photo-images (upper) and graphical representation (lower) of the effect of 2.5 μM or 5 μM on the (F) migration and (G) invasion of GBM8401 or U87MG cells. (H) Representative western-blot photo-images showing the effect of 2.5 μM – 5 μM on the expression of E-cadherin, N-cadherin, vimentin and slug proteins in GBM8401 or U87MG cells. *p<0.05, **p<0.01, ***p<0.001; GAPDH is loading control.

Also kindly see our revised Materials & Methods section, lines 138-146.

2.1. Drugs and Chemicals

Garcinol (sc-200891A, HPLC purity ≥95%) and Z-VAD-FMK (sc-3067, HPLC purity ≥95%) purchased from Santa Cruz Biotechnology (Santa Cruz, CA, USA) was dissolved in dimethyl sulfoxide (DMSO) to prepare a 20 mM stock and stored at −20°C until use. For different assays, the stock was further diluted using cell growth medium as appropriate. Dimethyl sulfoxide (DMSO), served as vehicle and negative control. BD Pharmingen™ PE Annexin V apoptosis detection kit I (#559763) was purchased from BD Biosciences (San Jose, CA, USA). Unless otherwise indicated, all reagents were obtained from Gibco (Thermo Fisher Scientific, Life Technologies, Foster City, CA, USA).

Please also see our revised Materials & Methods section, lines 266-276

2.13. PE-Annexin V/7-AAD cell death assay

PE-Annexin V/7-AAD staining was used for detection of cell death using the BD FACSCanto™ II flow cytometry system (BD Biosciences, San Jose, CA, USA) following the manufacturer’s instructions. Briefly, afer washing 5x105 wild type, Z-VAD-FMK-treated or Garcinol-treated U87MG cells twice with PBS, and once with annexin V binding buffer, the cells were incubated with PE-labeled annexin V and  7-AAD at room temperature for 15 min and then analyzed using flow cytometry. The mitochondrial transmembrane potential (DΨm) was evaluated using the cationic dye JC-1 (Mitochondrial Membrane Potential Assay Kit, #ab113850, Abcam plc., Cambridge, UK) in accordance with the manufacturer’s instructions (BD Pharmingen); 1x106 U87MG cells were incubated with 10 mg/mL JC-1 at 37 °C in the dark for 15 min, and then analyzed using flow cytometry. All samples were assayed three times in triplicate.

Q3: Reviewer #1: The authors performed wound healing and invasion experiments. This should be completed with the analysis of migration-associated genes expression such as ZEB1, TWIST, ZEB2, SLUG.

A3: We sincerely thank the reviewer for this important comment. We agree with the reviewer that our data should be complemented “with the analysis of migration- associated genes expression such as ZEB1, TWIST, ZEB2, SLUG, …”. Please kindly see our revised Results section, lines 345-381.

3.2. Garcinol significantly inhibits GBM cell viability and oncogenicity through induction of STAT3/5A signaling and enhanced apoptosis

Against the background of recent work demonstrating that garcinol inhibits CSCs-like phenotype of human non-small cell lung carcinoma by suppressing the Wnt/β-catenin/STAT3 signaling axis (Huang et al., 2018), we investigated the probable STAT signaling-mediated anti-GBM effect of garcinol (Figure 2A). Firstly, to provide some mechanistic insight, we demonstrated that treatment of U87MG or GBM8401 cells with  2.5 μM or 5 μM garcinol significantly downregulated the expression of p-STAT3, p-STAT5, p-ERK, and p-AKT (Figure 2B), Synchronous with the observed inhibition of STAT3, STAT5 and AKT signaling, garcinol significantly suppressed the viability of GBM4801 and U87MG cells, with 10 μM eliciting 51% or 25% reduced viability of U87MG or GBM8401 cells, respectively, and 40 μM eliciting  94.7% reduction of U87MG and GBM8401 cell viability, indicating a dose-dependent GBM cell killing effect (Figure 2C), and this reduced viability was associated with markedly enhanced  Bax/Bcl-xL apoptotic ratio, as 2.5 μM induced a 1.67-fold (p < 0.05) or  2.7-fold (p < 0.05) increase in U87MG or GBM8401 apoptotic ratio, while  5 μM increased the apoptotic ratio by 2.83-fold (p < 0.001) or 2.92-fold (p < 0.001) in the U87MG or GBM8401 cells, respectively (Figure 2D). In addition, using the Phycoerythrin (PE)-conjugated Annexin V/7- Amino-Actinomycin (7-AAD) staining, we demonstrated that compared to the cell death in the untreated control (U87MG: 1.6%, GBM8401: 4.0%) or 20 mM pan-caspase inhibitor benzyloxycarbonyl-Val-Ala-Asp-fluoromethyl ketone (Z-VAD-FMK)-treated negative control (U87MG: 3.2%, GBM8401: 10.0%), treatment with Garcinol enhanced U87MG cell death (2.5 mM:  10.5% or 5 mM: 41.2%), while 2.5 mM or 5 mM garcinol elicited 15.1% or 32.1% apoptosis of the GBM8401 cells, respectively (Figure 2E), indicating that the Garcinol-induced cell death was apoptotic and caspase-dependent. Since the highly invasive GBM spreads fast to surrounding brain tissue, thus, contributing to its documented lethality (Omuro & DeAngelis, 2013), we sought to understand if and how garcinol affects this invasive trait. We demonstrated that treatment with 2.5 μM or 5 μM dose-dependently suppressed the migration of the U87MG (~59%, p < 0.01 or 81%, p < 0.001, respectively) and GBM8401 (~48%, p < 0.01 or 76%, p < 0.001, respectively) cells at the 24 h time-point (Figure 2F). Similarly, 2.5 μM or 5 μM garcinol induced a 60% (p < 0.01) or ~80% (p < 0.001) reduction of U87MG invasive capacity, respectively, and 39% (p < 0.01) or 60% (p < 0.001) reduction in number of invaded GBM8401 cells (Figure 2G). Furthermore, in parallel assays to confirm the anticancer role of garcinol, consistent with earlier results, we demonstrated that treatment with 2.5 M or 5 M garcinol, significantly suppressed the expression of N-cadherin, vimentin and slug proteins, while conversely upregulating the expression of E-cadherin protein (Figure 2H),thus indicating that garcinol attenuates epithelial-to-mesenchymal transition (EMT) and the metastatic phenotype of GBM cells. Together, these data suggest that garcinol significantly inhibits GBM cell viability and oncogenicity through induction of STAT3/5A and associated signaling with enhanced apoptosis.

Please also kindly see our updated Figure 2 and its legend, lines 383-399.

Figure 2. Garcinol significantly inhibits GBM cell viability and oncogenicity through induction of STAT3/5A signaling and enhanced apoptosis. (A) Chemical structure of garcinol with molecular formula C38H50O6 and molecular weight 602.80 g/mol. (B) Representative western blot photo-images of the effect of 2.5 μM – 5 μM on the expression of p-STAT3, STAT3, p-STAT5, STAT5, p-ERK, ERK, p-AKT, and AKT proteins in GBM8401 or U87MG cells. (C) Graphical representation of the effect of 2.5 μM – 40 μM on the viability of GBM8401 or U87MG cells. (D) Representative western-blot photo-images showing the effect of 2.5 μM – 5 μM on the expression of Bax and Bcl-xL proteins in GBM8401 or U87MG cells. (E) Flow-cytometry data (upper) and graphical representation (lower) showing the effect of Garcinol on U87MG cells co-stained with PE-conjugated Annexin V and 7- AAD, compared with untreated or Z-VAD-FMK -treated negative control groups. Annexin V-stained Q4 cells are early apoptotic cells, whereas Q2 cells are late stage apoptotic (necrotic) cells. Apoptosis (%), sum of Q4+Q2; CTL, vehicle-treated; Neg CTL, pan-caspase inhibitor benzyloxycarbonyl-Val-Ala-Asp-fluoromethyl ketone (Z-VAD-FMK). Representative photo-images (upper) and graphical representation (lower) of the effect of 2.5 μM or 5 μM on the (F) migration and (G) invasion of GBM8401 or U87MG cells. (H) Representative western-blot photo-images showing the effect of 2.5 μM – 5 μM on the expression of E-cadherin, N-cadherin, vimentin and slug proteins in GBM8401 or U87MG cells. *p<0.05, **p<0.01, ***p<0.001; GAPDH is loading control.

Q4: Reviewer #1: In figure 3, the expression of SOX2 and OCT4 was analysed by IF. This does not constitute a quantitative method as with garcinol tumorspher are smallest, so it is logical to have less satining. The analysis of OCT4 and SOX2 expression should be performed by western blot or QPCR. Idem for STAT3 and STAT5A.

A4: We thank the reviewer for this comment. We have now included the requested data in our revised manuscript. Please kindly see Results section, lines 400-422.

3.3. Garcinol negatively impacts GBM stem cell-like phenotypes

Understanding that the highly prevalent and malignant GBM harbors self-renewing, tumorigenic GBM-SCs that facilitate tumor initiation and resistance to therapy (Lathia et al., 2015; Beier, Schulz, & Beier, 2011). To assess the effects of garcinol on GBM-SCs, we performed tumorsphere and colony formation assays on the GBM8401 and U87MG cell lines. The results from the tumorsphere assay demonstrated that 5 μM garcinol significantly caused both cell lines to lose their ability to form GBM tumorspheres, quantitatively and qualitatively, with ~ 88% (p < 0.01) reduction in the number of U87MG or GBM8401 tumorspheres formed, and ~96% (p < 0.01) or 89% (p < 0.001) reduction in the U87MG or GBM8401 tumorsphere sizes, respectively (Figure 3A). Furthermore, because of the biological relevance of clonality in GBM-SCs origin (Lathia et al., 2015), we demonstrated that 2.5 μM - 5 μM garcinol significantly inhibited the ability of the GBM cells to form colonies, dose-dependently, as 2.5 μM reduced the number of formed U87MG or GBM8401 colonies by 49% (p < 0.05) or 36% (p < 0.05), respectively, while 5 μM induced a 75% (p < 0.01) or 72% (p < 0.01) reduction, respectively (Figure 3B). Contextually, the garcinol-induced inhibition of tumorsphere and colony formation potential was associated with significant and dose-dependent down-regulation of the nuclear expression of stemness proteins SOX2 and OCT4 (Figure 3C) as shown with immunofluorescence (IFC) assay, and this inhibitory effect was associated with significantly suppressed p-STAT3 and p-STAT5A immunofluorescence, in a dose-dependent manner (Figure 3D). Consistently, akin to the IFC results, western blot analyses also showed that treatment with garcinol downregulated SOX2 (2.5 μM: 1.92-fold, p<0.05; 5 μM: 2.78-fold, p<0.01) and OCT4 (2.5 μM: 2.63-fold, p<0.05; 5 μM: 5.88-fold, p<0.01) protein expression levels in U87MG tumorspheres with similar trend in the GBM8401 tumospheres (Figure 3E). These data are indicative of the negative influence of garcinol on the stem cell-like phenotypes of GBM cells.

Also kindly see our updated Figure 3 and its legend, lines 424-434.

Figure 3. Garcinol negatively impacts GBM stem cell-like phenotypes. (A) Representative photo-images (left) and histograms (middle, right) of the effect of 5 μM garcinol on the number and size of tumorspheres formed by U87MG or GBM8401 cells. (B) Representative images (upper) and graph (lower) of the effect of 2 μM or 5 μM garcinol on the number of colonies formed by U87MG or GBM8401 cells. (C) Representative photo-images showing the effect of 2 μM or 5 μM garcinol on the tumorsphere size and sub-cellular localization of SOX2 and OCT4 proteins in U87MG or GBM8401 cells. (D) Graphs showing how 2 μM or 5 μM affect the MIF of SOX2, OCT4, p-STAT3 and p-STAT5A in U87MG or GBM8401 cells. (E) Representative western blot images (left) and histograms (right) showing the effect of 2 μM or 5 μM garcinol on the expression of SOX2 and OCT4 proteins. *p<0.05, **p<0.01, ***p<0.001; MIF, median immunofluorescence; DAPI served as nuclear marker. GAPDH served as loading control.

Q5: Reviewer #1: The authors conclude that hsa-miR-181d is an inhibitor of STAT3 and STAT5A expression. However, only pSTAT3 and pSTAT5A (and not total STAT3 and STAT5A expression) were modified by the miR181d inhibitor or mimic. This leads to conclude that the effect of miR-181d mostly rely on an upstream target. The authors should study the capacity of miR-181d to decrease the expression of STAT3 and STAT5A phosphorylation inductors, such as JAK1, JAK2, IL-6R, … This will be more consistent as the effects observed were both on STAT3 and STAT5A.

A5: We sincerely thank the reviewer for this insightful suggestion. Indeed, as suggested by the reviewer, we observed a moderate down-modulation of JAK2 by miR-181d, akin to the effect of Garcinol after treatment with mir-181d inhibitor. Please kindly see our revised Results section, lines 436-477.

3.4. Garcinol increases the expression of hsa-miR181d, which has inhibitory effects on STAT3 and STAT5 expression

Having established that garcinol impairs STAT3 and STAT5A activation, we probed for likely modulators and/or mediators of the interaction between garcinol and the STAT proteins. Hsa-miR-181d shown in Figure 4A, has been implicated in the worse OS of patients with GBM (Zhang et al., 2012). Consistent with this, using the Schrodinger PyMOL 2.3 molecular docking and visualization software (http://pymol.org) we demonstrated that hsa-miR-181d interacts with and binds directly to STAT3 (docking score = -254.49, ligand root mean square deviation (RMSD) = 195.10 Å) or STAT5A (docking score = -234.19, ligand RMSD = 143.04 Å), complementing earlier prediction that hsa-miR-181d binds with STAT3 with a mirSVR or PhastCons score of -0.26 or 0.69, respectively, while it binds with STAT5A with a mirSVR  or PhastCons score of -0.21 or 0.49, respectively (Figure 4A).  Where the mirSVR shows the likelihood of hsa-miR-181d down-regulating the target mRNA STAT3 and STAT5A based on the sequence and structure features in the miRNA/mRNA predicted target sites. Moreover, the PhastCons score shows the likelihood that the predicted miRNA/mRNA binding nucleotides are conserved. In concordance, our qRT-PCR analysis of 5 μM garcinol-treated U87MG and GBM8401 cells showed that garcinol significantly induced higher expression of miR-181d in the U87MG (2.7-fold, p < 0.01) and GBM8401 (2.1-fold, p < 0.01) cells (Figure 4B). Furthermore, having implicated STAT3/5A in enhanced migration and invasiveness of U87MG or GBM8401 cells, we examined the probable effect of hsa-miR-281d on these metastatic phenotypes of GBM cells. Using the scratch wound-healing assay, we demonstrated that compared to the untreated control or syn-mir-treated cells, treatment with mir-181d inhibitor significantly enhanced the ability of the U87MG cells to migrate (4.64-fold, p<0.01), while treatment with the mir-181d-mimic elicited marked attenuation of migration (3.80-fold, p<0.01) (Figure 4C), which is reminiscent of suppressed migration induced by 5 mM garcinol (4.17-fold, p<0.01) earlier.  Similarly, while treatment with mir-181d inhibitor significantly enhanced the invasiveness of the U87MG cells (4.63-fold, p<0.01), treatment with the mir-181d-mimic elicited profound suppression of invasion (4.15-fold, p<0.01) (Figures 4D), and this was akin to the effect of 5 mM garcinol (4.22-fold, p<0.01). To confirm a direct relationship between the STAT proteins and miR-181d expression, western blot analysis was done comparing samples exposed to mir-181d inhibitor, mir-181d-mimic, or mir-181d inhibitor/garcinol combination. The results showed that mir-181d inhibitor significantly enhanced the expression of p-STAT3, p-STAT5, N-cadherin and vimentin proteins, but suppressed E-cadherin protein expression compared to the control group, while for the mir-181d-mimic-treated cells, the p-STAT3, p-STAT5, N-cadherin, and vimentin protein expression levels were significantly lower, but E-cadherin was upregulated; For cells incubated with mir-181d inhibitor and 5 μM garcinol concomitantly, p-STAT3 and p-STAT5 protein expression levels were markedly higher than in the mir-181d-mimic group but lower than the mir-181d-inhibitor group (Figure 4E). Concomitantly, we observed that compared with the control group, syn-mir-treated cells, or even mir-181d inhibitor-treated cells, treatment with mir-181d-mimic markedly suppressed the expression of JAK2 protein (2.22-fold, p<0.01), akin to the effect elicited by treatment with concurrently with mir-181d inhibitor and 5 mM garcinol (2.94-fold, p<0.01) (Figure 4E), which is consistent with the results above and suggestive of a miR-181d-mediated JAK2-modulated phosphorylation of STAT3/5A. These results indicate that garcinol can activate mir-181d activity which suppresses STAT3/5A activation.

Also kindly see our updated Figure 4 and its legend, lines 479-491.

Figure 4. Garcinol increases the expression of hsa-miR181d, which has inhibitory effects on STAT3 and STAT5 expression. (A) 3-dimensional visualization of direct interaction between hsa-miR-181d and STAT3 (left) or STAT5A (middle) and images showing the complementary sequence alignment of hsa-miR-181d with STAT3 (upper right) or STAT5A (lower right). The mirSVR and PhastCons scores are indicated. (B) Histograms of the effect of 5 μM on hsa-miR-181d expression in U87MG or GBM8401 cells. (C) Representative images (left) and histograms (right) showing the effect of Syn-mir-CTL, mir-181d-inhibitor, or mir-181d-mimic on the migration of U87MG cells over 24 h, as determined by wound-healing assay. (D) Representative images (left) and histograms (right) showing the effect of Syn-mir-CTL, mir-181d-inhibitor, or mir-181d-mimic on the invasion of U87MG cells. (E) Representative western-blot photo-images (left) and histograms (right) comparing the effect of Syn-mir-CTL, mir-181d-inhibitor, mir-181d-mimic, or garcinol on the expression level of JAK2, p-STAT3, STAT3, p-STAT5, STAT5, E-cadherin, N-cadherin, and vimentin proteins in U87MG or GBM8401 cells. *p<0.05, **p<0.01, ***p<0.001; GAPDH is loading control.

Q6: Reviewer #1: Minor points

6.1- In M&M can the authors explained how they count tumorpheres (size, number, …).

6.2- In M&M can the authors explained how percentages of migrations were calculated.

6.3- In M&M can the authors explained how Q-scores for IHC were calculated.

6.4- In some figure legend, the time of exposure to treatments should be added.

6.5- In paragraph 3.3, the authors said that they demostrated (line 358). It is not true as it is an analysis of the alignement of 2 sequences.

6.6- In the discussion (line 430) the authors said they examined garcinol’s therapeutic effect … It is not true as they only provide in vitro experiments.

A6: We sincerely appreciate the reviewer’s comment.

A6.1- As requested by the reviewer, we have now included notes on how tumorsphere number and size were determined in our revised “M&M”. Please kindly see our revised Materials & Methods section, lines 218-233.

2.8. Tumorsphere Formation Assay

For tumorsphere formation, U87MG and GBM8401 cells were cultured in Chemicon® serum-free HEScGRO medium for human embryonic stem cell culture (CAT. No. SCM020, Merck KGaA, Darmstadt, Germany) supplemented with 10 ng/mL human recombinant basic fibroblast growth factor (hbFGF; Invitrogen, Carlsbad, CA, USA), 20 ng/mL human epithelial growth factor (hEGF; Millipore, Bedford, MA, USA), B27 supplement (Invitrogen, Carlsbad, CA, USA), heparin (CAT. No. 07980; STEMCELL Technologies Inc., Interlab Co., Ltd, Taipei, Taiwan), and NeuroCultTM NS-A proliferation supplement (CAT. No. 05753, STEMCELL Technologies Inc., Interlab Co., Ltd, Taipei, Taiwan). The cells were seeded at a concentration of 1x103 cells/mL/well in 6-well ultra-low adhesion plates (Corning Inc., Corning, NY, USA) with or without 5 μM of garcinol and incubated at 37oC in 5% humidified CO2 atmosphere for 7–10 days. The anchorage-independent tumorspheres (≥ 90 μm in diameter) were counted under inverted phase contrast microscope at a magnification of 40X and photographed.Tumorsphere sizes were determined from 5 randomly-selected fields of the digital images acquired, using NIH ImageJ software (https://imagej.nih.gov/ij/). We calculated our tumorsphere formation efficiency (MFE%) using the formula: MFE (%) = [No. of tumorspheres per well]/[No. of seeded cells per well] x 100

6.2- As requested by the reviewer, we have now included notes on how percentages of migrations were calculated in our revised “M&M”. Please kindly see our revised Materials & Methods section, lines 251-259.

2.11. Wound-healing migration Assay

After U87MG and GBM8401 cells were seeded in 24-well plates (Corning, Corning, NY, USA) with DMEM with 10% FBS and incubated until 100% confluency, scratch-wounds were made along the median axis of the adherent monolayer cells using sterile 200 μL micropipette tips. The wells were carefully washed with PBS to remove detached cells and then incubated in new growth media containing 0, 2.5, or 5 μM of garcinol for 24 h or 48 h. Photographs of scratch-wound healing were taken at indicated time-point and under microscope with a 10× objective lens, and analyzed with the NIH ImageJ software (https://imagej.nih.gov/ij/). The percentage migration (M%) was calculated using the formula: M (%) = [denuded area at time ‘x’]/[denuded area at time 0] x 100.

6.3- As requested by the reviewer, we have now included notes on explained how Q-scores for IHC were calculated in our revised “M&M”. Please kindly see our revised Materials & Methods section, lines 189-205.

2.6. Immunohistochemical (IHC) Staining

The study was approved by the Joint Institutional Review Board (JIRB) of the Taipei Medical University –Shuang Ho Hospital (Approval number: N201903047). Tissue samples from patients with primary and recurrent GBM were obtained from the Taipei Medical University-Shuang Ho Hospital GBM cohort (n = 45). After de-waxing the paraffin-embedded 4 μm tissue sections using xylene for 5 min twice and re-hydrating with 100% ethanol twice for 5 min, 95% ethanol for 5 min and 80% ethanol for 5 min, 3% hydrogen peroxide (H2O2) (TA-125-H2O2Q, Thermo Fisher Scientific, Waltham, MA, USA) was used to block endogenous peroxidase activity for 10 min. The sections were then immersed in 10 mmol/L ethylenediaminetetraacetic acid (EDTA) for 3 min in a pressure cooker, then blocked with 10% normal serum. Thereafter, tissue samples were incubated with primary antibody against STAT3 (1:200) or STAT5 (1:200) at 4oC overnight, and then with biotin-labeled secondary antibody at room temperature for 1h. Sections were incubated in diaminobenzidine (DAB) and then counterstained with hematoxylin. Visualization was done under a light microscope. For staining determination, we used the Quick (Q) score formula: Q = [percentage of stained/positive cells] x [intensity of staining]. The maximum score for percentage/distribution of stained/positive cells was 100, while the intensity of staining was delineated into weak (1), moderate (2), or strong (3), making the obtainable maximum Q score = 300.

6.4- We believe this concern has now been addressed in our revised manuscript. Please kindly see our updated Figures and their revised legends.

6.5- May we politely point out that we believe the erudite reviewer is mistaken here, as contrary to his assertion that “It is not true as it is an analysis of the alignment of 2 sequences.”, what we showed is indeed a bioinformatics-aided prediction of direct interaction between STAT3/5A and hsa-miR-181d based on sequence complementarity and conservation across species, thus the indication of mirSVR and PhastCons scores. Nevertheless, to address the reviewer’s concern, we have reworded the statement. Please kindly see our revised Results section, lines 436-477.

3.4. Garcinol increases the expression of hsa-miR181d, which has inhibitory effects on STAT3 and STAT5 expression

Having established that garcinol impairs STAT3 and STAT5A activation, we probed for likely modulators and/or mediators of the interaction between garcinol and the STAT proteins. Hsa-miR-181d shown in Figure 4A, has been implicated in the worse OS of patients with GBM (Zhang et al., 2012). Consistent with this, using the Schrodinger PyMOL 2.3 molecular docking and visualization software (http://pymol.org) we demonstrated that hsa-miR-181d interacts with and binds directly to STAT3 (docking score = -254.49, ligand root mean square deviation (RMSD) = 195.10 Å) or STAT5A (docking score = -234.19, ligand RMSD = 143.04 Å), complementing earlier prediction that hsa-miR-181d binds with STAT3 with a mirSVR or PhastCons score of -0.26 or 0.69, respectively, while it binds with STAT5A with a mirSVR  or PhastCons score of -0.21 or 0.49, respectively (Figure 4A).  Where the mirSVR shows the likelihood of hsa-miR-181d down-regulating the target mRNA STAT3 and STAT5A based on the sequence and structure features in the miRNA/mRNA predicted target sites. Moreover, the PhastCons score shows the likelihood that the predicted miRNA/mRNA binding nucleotides are conserved. In concordance, our qRT-PCR analysis of 5 μM garcinol-treated U87MG and GBM8401 cells showed that garcinol significantly induced higher expression of miR-181d in the U87MG (2.7-fold, p < 0.01) and GBM8401 (2.1-fold, p < 0.01) cells (Figure 4B). Furthermore, having implicated STAT3/5A in enhanced migration and invasiveness of U87MG or GBM8401 cells, we examined the probable effect of hsa-miR-281d on this metastatic phenotype of GBM cells. Using the scratch wound-healing assay, we demonstrated that compared to the untreated control or syn-mir-treated cells, treatment with mir-181d inhibitor significantly enhanced the ability of the U87MG cells to migrate (4.64-fold, p<0.01), while treatment with the mir-181d-mimic elicited marked attenuation of migration (3.80-fold, p<0.01) (Figure 4C), which is reminiscent of suppressed migration induced by 5 mM garcinol (4.17-fold, p<0.01) earlier. Similarly, while treatment with mir-181d inhibitor significantly enhanced the invasiveness of the U87MG cells (4.63-fold, p<0.01), treatment with the mir-181d-mimic elicited profound suppression of invasion (4.15-fold, p<0.01) (Figures 4D), and this was akin to the effect of 5 mM garcinol (4.22-fold, p<0.01). To confirm a direct relationship between the STAT proteins and miR-181d expression, western blot analysis was done comparing samples exposed to mir-181d inhibitor, mir-181d-mimic, or mir-181d inhibitor/garcinol combination. The results showed that mir-181d inhibitor significantly enhanced the expression of p-STAT3, p-STAT5, N-cadherin and vimentin proteins, but suppressed E-cadherin protein expression compared to the control group, while for the mir-181d-mimic-treated cells, the p-STAT3, p-STAT5, N-cadherin, and vimentin protein expression levels were significantly lower, but E-cadherin was upregulated; For cells incubated with mir-181d inhibitor and 5 μM garcinol concomitantly, p-STAT3 and p-STAT5 protein expression levels were markedly higher than in the mir-181d-mimic group but lower than the mir-181d-inhibitor group (Figure 4E). Concomitantly, we observed that compared with the control group, syn-mir-treated cells, or even mir-181d inhibitor-treated cells, treatment with mir-181d-mimic markedly suppressed the expression of JAK2 protein (2.22-fold, p<0.01), akin to the effect elicited by treatment with concurrently with mir-181d inhibitor and 5 mM garcinol (2.94-fold, p<0.01) (Figure 4E), which is consistent with the results above and suggestive of a miR-181d-mediated JAK2-modulated phosphorylation of STAT3/5A. These results indicate that garcinol can activate mir-181d activity which suppresses STAT3/5A activation.

6.6- To allay the reviewer’s concern regarding “garcinol’s therapeutic effect …”, we have reworded the statement in question. Please kindly see our revised Discussion section, lines 558-575.

In the context of the GBM SCs-phenotypes including enhanced proliferation, oncogenicity, therapy resistance, recurrence and poor prognosis, we further demonstrated that garcinol significantly inhibited GBM8401 and U87MG cell viability in a dose-dependent manner, as well as suppresses the cell invasive and migratory potentials, thus, demonstrating the anti-proliferative and anti-metastatic efficacy of garcinol in GBM. This is consistent with documented robust growth-inhibitory effects of garcinol demonstrated against colon cancer and immortalized intestinal cells (Hong et al., 2007). Since STAT3 and STAT5A are implicated in the maintenance of the stem cell-like characteristics of GBM, we examined garcinol’s inhibitory effect on the GBM-SCs profile. Interestingly, low dose garcinol (≤ 5 μM) deregulated STAT3/5A signaling with repressed AKT and ERK crosstalk, and this was sufficient to significantly impeded GBM cell migration, invasion, clonogenicity, and tumorsphere formation, with associated increase in apoptotic index and nuclear expression of SOX2 and OCT4. Our findings are corroborated by recent evidence that STAT3 and STAT5 are constitutively activated in malignant cells, and that their persistent activation facilitates cancer development and progression by altering downstream gene expression through epigenetic modification, EMT induction, oncogenic modification of the tumor microenvironment, and enhancing of CSCs self-renewal and differentiation (Yuan et al., 2015), as well as evidence implicating high ERK1/2 activity in the acquisition and maintenance of SOX2-expressing Glioma stem cells (Kwon et al., 2017).

Reviewer 2 Report

This manuscript evaluates the role of a novel inhibitor Garcinol and miR181d on the expression of STAT3 and STAT5a in glioblastoma cells. They find that Garcinol is able to inhibit glioblastoma proliferation, migration and invasion using several in vitro assays. Overall the results are somewhat preliminary and several concerns should be addressed if the manuscript is to be accepted. A more robust set of expts are required. These are highlighted in the following points:

Specific points to address in regards to above:

The authors should determine if Garcinol displays efficacy in vivo using appropriate glioblastoma animal models. As Garcinol can inhibit several intracellular kinases the authors should attempt to identify whether it inhibits a specific set of receptors upstream. The authors show that miR181d can potentially bind STAT3 and STAT5a (Fig 4A). They also show that miR181d reduces the phosphorylation of STAT3 and STAT5a. However miR181d presumably reduces the expression of STAT3 and STAT5a which is not observed using an miR181d inhibitor and mimic (Fig 4C). It is therefore confusing as to how miR181d reduces phosphorylation of STAT3 and STAT5a without reducing their overall expression. The authors need to demonstrate that the miR181d inhibitor and mimic display increases/decreases of cell proliferation/migration and invasion.

Author Response

Q1: Reviewer #2: This manuscript evaluates the role of a novel inhibitor Garcinol and miR181d on the expression of STAT3 and STAT5a in glioblastoma cells. They find that Garcinol is able to inhibit glioblastoma proliferation, migration and invasion using several in vitro assays. Overall the results are somewhat preliminary, and several concerns should be addressed if the manuscript is to be accepted. A more robust set of expts are required. These are highlighted in the following points.

A1: We thank the reviewer for the time taken to review our work, the critical assessment of our findings and the helpful suggestions given to help us improve the acceptability and appeal of our work.

Q2: Reviewer #2:  Specific points to address in regard to above.

A2: We thank the reviewer for these comments.

Q2.1: Reviewer #2: The authors should determine if Garcinol displays efficacy in vivo using appropriate glioblastoma animal models.

A2.1: We thank the reviewer for these comments. Indeed, we had started the in vivo studies before our initial submission. We now include results of the in vivo studies as requested by the reviewer. Please kindly see our revised Materials and Methods section, lines 291-303.

2.15. Mice Tumor Xenograft Study

The animal study protocol was approved by the Animal Care and User Committee at Taipei Medical University (Affidavit of Approval of Animal Use Protocol # Taipei Medical University- LAC-2017-0512) consistent with the U.S. National Institutes of Health Guide for the Care and Use of Laboratory Animals. NOD/SCID mice purchased from BioLASCO (BioLASCO Taiwan Co., Ltd. Taipei, Taiwan) were all inoculated with 1 × 106 U87MG cells in their hind flank subcutaneously, and then randomly placed into untreated control (0.1% DMSO, 100 μL daily; n = 3) or 1 mg/kg garcinol-treated (1 mg/kg body weight, suspended in 0.1% DMSO, intraperitoneal [i.p.] injection, five times/week; n = 3) group. Tumor growth was measured bi-weekly for 4 weeks after tumor cell inoculation using calipers and tumor volume (v) calculated using the formula: v = l × w2 × 0.5, where l is longest diameter, and w is shortest diameter of tumor. Thereafter, the mice were humanely sacrificed, and tumor samples extracted for further comparative immunohistochemistry (IHC) and miRNA analyses.

Also see our revised Results section, lines 492-511.

3.5. Garcinol inhibits tumor growth in GBM mice models through inversely correlated STAT3/5A and hsa-miR-181d expressions

Having shown that treatment with garcinol suppresses the cancer stem cell-like phenotype of U87MG and GBM8401 cells in vitro, to determine the probable suppressive effect of garcinol on the formation and growth of tumor, in vivo, we generated NOD/SCID mice  xenograft models derived by inoculation with 1 × 106 U87MG cells subcutaneously in the hind-flank. Mice were randomly placed into control (n = 3) or garcinol treatment (n = 3) group. We demonstrated that treatment with 1 mg/kg garcinol significantly reduced the size of tumors formed in the treated mice, compared to the untreated control group (U87MG: ~7.1-fold smaller, p < 0.001 by week 4) (Figures 5A), without adversely affecting the mice body weight (Figure 5B). In subsequent experiments using protein lysates derived from the tumors extracted from the untreated and garcinol-treated mice, we demonstrated that compared to the untreated control group, STAT3, pSTAT3, STAT5A, p-STAT5A, Ki-67, and Bcl-xL protein expression levels were concomitantly suppressed, while Bax expression was significantly enhanced (Figure 5C). Moreover, for tumors extracted from the 1 mg/kg garcinol-treated U87MG tumor-bearing mice, compared to the untreated control group, STAT3, or STAT5A mRNA expression were suppressed by 4-fold (p < 0.01) or 3.87-fold (p < 0.01), while miR-181d expression was enhanced by 3.52-fold (p < 0.01) in the U87MG mice treated with 1 mg/kg garcinol (Figure 5D). These findings indicate that garcinol inhibits tumorigenicity and growth of GBM by abrogating STAT3/5A signaling, and upregulating hsa-miR-181d, with concomitant suppression of Ki-67 proliferation index and enhancement of Bax/Bcl-xL apoptotic ratio, in vivo.

Also kindly see our updated Figure 5 and its legend, lines 513-520.

Figure 5. Garcinol inhibits tumor growth in GBM mice models through inversely correlated STAT3/5A and hsa-miR-181d expressions. Representative image and graph showing the effect of garcinol on the (A) tumor volume and (B) body weight of U87MG-tumor-bearing mice. p-values were determined by 2-way ANOVA. (C) Representative images and histograms of the differential expression of STAT3, pSTAT3, STAT5A, pSTAT5A, Ki-67, Bax, and Bcl-xL proteins level in tumors extracted from mice bearing U87MG cell-derived tumors, treated with or without garcinol. (D) Histograms showing the effect of garcinol treatment on STAT3, STAT5A and miR-181d expression levels in U87MG-tumor-bearing mice. *p<0.05, **p<0.01, ***p<0.001; GAPDH is loading control.

Q2.2: Reviewer #2: As Garcinol can inhibit several intracellular kinases the authors should attempt to identify whether it inhibits a specific set of receptors upstream.

A2.2: We thank the reviewer for these comments. Please kindly see our revised Results section, lines 436-477.

3.4. Garcinol increases the expression of hsa-miR181d, which has inhibitory effects on STAT3 and STAT5 expression

Having established that garcinol impairs STAT3 and STAT5A activation, we probed for likely modulators and/or mediators of the interaction between garcinol and the STAT proteins. Hsa-miR-181d shown in Figure 4A, has been implicated in the worse OS of patients with GBM (Zhang et al., 2012). Consistent with this, using the Schrodinger PyMOL 2.3 molecular docking and visualization software (http://pymol.org) we demonstrated that hsa-miR-181d interacts with and binds directly to STAT3 (docking score = -254.49, ligand root mean square deviation (RMSD) = 195.10 Å) or STAT5A (docking score = -234.19, ligand RMSD = 143.04 Å), complementing earlier prediction that hsa-miR-181d binds with STAT3 with a mirSVR or PhastCons score of -0.26 or 0.69, respectively, while it binds with STAT5A with a mirSVR  or PhastCons score of -0.21 or 0.49, respectively (Figure 4A).  Where the mirSVR shows the likelihood of hsa-miR-181d down-regulating the target mRNA STAT3 and STAT5A based on the sequence and structure features in the miRNA/mRNA predicted target sites. Moreover, the PhastCons score shows the likelihood that the predicted miRNA/mRNA binding nucleotides are conserved. In concordance, our qRT-PCR analysis of 5 μM garcinol-treated U87MG and GBM8401 cells showed that garcinol significantly induced higher expression of miR-181d in the U87MG (2.7-fold, p < 0.01) and GBM8401 (2.1-fold, p < 0.01) cells (Figure 4B). Furthermore, having implicated STAT3/5A in enhanced migration and invasiveness of U87MG or GBM8401 cells, we examined the probable effect of hsa-miR-181d on these metastatic phenotypes of GBM cells. Using the scratch wound-healing assay, we demonstrated that compared to the untreated control or syn-mir-treated cells, treatment with mir-181d inhibitor significantly enhanced the ability of the U87MG cells to migrate (4.64-fold, p<0.01), while treatment with the mir-181d-mimic elicited marked attenuation of migration (3.80-fold, p<0.01) (Figure 4C), which is reminiscent of suppressed migration induced by 5 μM garcinol (4.17-fold, p<0.01) earlier.  Similarly, while treatment with mir-181d inhibitor significantly enhanced the invasiveness of the U87MG cells (4.63-fold, p<0.01), treatment with the mir-181d-mimic elicited profound suppression of invasion (4.15-fold, p<0.01) (Figures 4D), and this was akin to the effect of 5 mM garcinol (4.22-fold, p<0.01). To confirm a direct relationship between the STAT proteins and miR-181d expression, western blot analysis was done comparing samples exposed to mir-181d inhibitor, mir-181d-mimic, or mir-181d inhibitor/garcinol combination. The results showed that mir-181d inhibitor significantly enhanced the expression of p-STAT3, p-STAT5, N-cadherin and vimentin proteins, but suppressed E-cadherin protein expression compared to the control group, while for the mir-181d-mimic-treated cells, the p-STAT3, p-STAT5, N-cadherin, and vimentin protein expression levels were significantly lower, but E-cadherin was upregulated; For cells incubated with mir-181d inhibitor and 5 μM garcinol concomitantly, p-STAT3 and p-STAT5 protein expression levels were markedly higher than in the mir-181d-mimic group but lower than the mir-181d-inhibitor group (Figure 4E). Concomitantly, we observed that compared with the control group, syn-mir-treated cells, or even mir-181d inhibitor-treated cells, treatment with mir-181d-mimic markedly suppressed the expression of JAK2 protein (2.22-fold, p<0.01), akin to the effect elicited by treatment with concurrently with mir-181d inhibitor and 5 μM garcinol (2.94-fold, p<0.01), which is consistent with the results above and suggestive of a miR-181d-mediated JAK2-modulated phosphorylation of STAT3/5A. These results indicate that garcinol can activate mir-181d activity which suppresses STAT3/5A activation.   

Also kindly see our updated Figure 4 and its legend, lines 479-491.

Figure 4. Garcinol increases the expression of hsa-miR181d, which has inhibitory effects on STAT3 and STAT5 expression. (A) 3-dimensional visualization of direct interaction between hsa-miR-181d and STAT3 (left) or STAT5A (middle) and images showing the complementary sequence alignment of hsa-miR-181d with STAT3 (upper right) or STAT5A (lower right). The mirSVR and PhastCons scores are indicated. (B) Histograms of the effect of 5 μM on hsa-miR-181d expression in U87MG or GBM8401 cells. (C) Representative images (left) and histograms (right) showing the effect of Syn-mir-CTL, mir-181d-inhibitor, or mir-181d-mimic on the migration of U87MG cells over 24 h, as determined by wound-healing assay. (D) Representative images (left) and histograms (right) showing the effect of Syn-mir-CTL, mir-181d-inhibitor, or mir-181d-mimic on the invasion of U87MG cells. (E) Representative western-blot photo-images (left) and histograms (right) comparing the effect of Syn-mir-CTL, mir-181d-inhibitor, mir-181d-mimic, or garcinol on the expression level of JAK2, p-STAT3, STAT3, p-STAT5, STAT5, E-cadherin, N-cadherin, and vimentin proteins in U87MG or GBM8401 cells. *p<0.05, **p<0.01, ***p<0.001; GAPDH is loading control.

Q2.3: Reviewer #2: The authors show that miR181d can potentially bind STAT3 and STAT5a (Fig 4A). They also show that miR181d reduces the phosphorylation of STAT3 and STAT5a. However, miR181d presumably reduces the expression of STAT3 and STAT5a which is not observed using an miR181d inhibitor and mimic (Fig 4C). It is therefore confusing as to how miR181d reduces phosphorylation of STAT3 and STAT5a without reducing their overall expression. The authors need to demonstrate that the miR181d inhibitor and mimic display increases/decreases of cell proliferation/migration and invasion.

A2.3: We thank the reviewer for this comment and do understand the reviewer’s confusion, however, while we would have loved to see our data following the trend described by the reviewer, in the spirit of data integrity, we have indeed presented the data as is. We have added some more data and tried to provide some explanation in our results description. Please kindly see our revised Results section, lines 436-477. 

3.4. Garcinol increases the expression of hsa-miR181d, which has inhibitory effects on STAT3 and STAT5 expression

Having established that garcinol impairs STAT3 and STAT5A activation, we probed for likely modulators and/or mediators of the interaction between garcinol and the STAT proteins. Hsa-miR-181d shown in Figure 4A, has been implicated in the worse OS of patients with GBM (Zhang et al., 2012). Consistent with this, using the Schrodinger PyMOL 2.3 molecular docking and visualization software (http://pymol.org) we demonstrated that hsa-miR-181d interacts with and binds directly to STAT3 (docking score = -254.49, ligand root mean square deviation (RMSD) = 195.10 Å) or STAT5A (docking score = -234.19, ligand RMSD = 143.04 Å), complementing earlier prediction that hsa-miR-181d binds with STAT3 with a mirSVR or PhastCons score of -0.26 or 0.69, respectively, while it binds with STAT5A with a mirSVR  or PhastCons score of -0.21 or 0.49, respectively (Figure 4A).  Where the mirSVR shows the likelihood of hsa-miR-181d down-regulating the target mRNA STAT3 and STAT5A based on the sequence and structure features in the miRNA/mRNA predicted target sites. Moreover, the PhastCons score shows the likelihood that the predicted miRNA/mRNA binding nucleotides are conserved. In concordance, our qRT-PCR analysis of 5 μM garcinol-treated U87MG and GBM8401 cells showed that garcinol significantly induced higher expression of miR-181d in the U87MG (2.7-fold, p < 0.01) and GBM8401 (2.1-fold, p < 0.01) cells (Figure 4B). Furthermore, having implicated STAT3/5A in enhanced migration and invasiveness of U87MG or GBM8401 cells, we examined the probable effect of hsa-miR-181d on these metastatic phenotypes of GBM cells. Using the scratch wound-healing assay, we demonstrated that compared to the untreated control or syn-mir-treated cells, treatment with mir-181d inhibitor significantly enhanced the ability of the U87MG cells to migrate (4.64-fold, p<0.01), while treatment with the mir-181d-mimic elicited marked attenuation of migration (3.80-fold, p<0.01) (Figure 4C), which is reminiscent of suppressed migration induced by 5 μM garcinol (4.17-fold, p<0.01) earlier.  Similarly, while treatment with mir-181d inhibitor significantly enhanced the invasiveness of the U87MG cells (4.63-fold, p<0.01), treatment with the mir-181d-mimic elicited profound suppression of invasion (4.15-fold, p<0.01) (Figures 4D), and this was akin to the effect of 5 mM garcinol (4.22-fold, p<0.01). To confirm a direct relationship between the STAT proteins and miR-181d expression, western blot analysis was done comparing samples exposed to mir-181d inhibitor, mir-181d-mimic, or mir-181d inhibitor/garcinol combination. The results showed that mir-181d inhibitor significantly enhanced the expression of p-STAT3, p-STAT5, N-cadherin and vimentin proteins, but suppressed E-cadherin protein expression compared to the control group, while for the mir-181d-mimic-treated cells, the p-STAT3, p-STAT5, N-cadherin, and vimentin protein expression levels were significantly lower, but E-cadherin was upregulated; For cells incubated with mir-181d inhibitor and 5 μM garcinol concomitantly, p-STAT3 and p-STAT5 protein expression levels were markedly higher than in the mir-181d-mimic group but lower than the mir-181d-inhibitor group (Figure 4E). Concomitantly, we observed that compared with the control group, syn-mir-treated cells, or even mir-181d inhibitor-treated cells, treatment with mir-181d-mimic markedly suppressed the expression of JAK2 protein (2.22-fold, p<0.01), akin to the effect elicited by treatment with concurrently with mir-181d inhibitor and 5 μM garcinol (2.94-fold, p<0.01), which is consistent with the results above and suggestive of a miR-181d-mediated JAK2-modulated phosphorylation of STAT3/5A. These results indicate that garcinol can activate mir-181d activity which suppresses STAT3/5A activation.    

Also kindly see our updated Figure 4 and its legend, lines 479-491.

Figure 4. Garcinol increases the expression of hsa-miR181d, which has inhibitory effects on STAT3 and STAT5 expression. (A) 3-dimensional visualization of direct interaction between hsa-miR-181d and STAT3 (left) or STAT5A (middle) and images showing the complementary sequence alignment of hsa-miR-181d with STAT3 (upper right) or STAT5A (lower right). The mirSVR and PhastCons scores are indicated. (B) Histograms of the effect of 5 μM on hsa-miR-181d expression in U87MG or GBM8401 cells. (C) Representative images (left) and histograms (right) showing the effect of Syn-mir-CTL, mir-181d-inhibitor, or mir-181d-mimic on the migration of U87MG cells over 24 h, as determined by wound-healing assay. (D) Representative images (left) and histograms (right) showing the effect of Syn-mir-CTL, mir-181d-inhibitor, or mir-181d-mimic on the invasion of U87MG cells. (E) Representative western-blot photo-images (left) and histograms (right) comparing the effect of Syn-mir-CTL, mir-181d-inhibitor, mir-181d-mimic, or garcinol on the expression level of JAK2, p-STAT3, STAT3, p-STAT5, STAT5, E-cadherin, N-cadherin, and vimentin proteins in U87MG or GBM8401 cells. *p<0.05, **p<0.01, ***p<0.001; GAPDH is loading control.

Reviewer 3 Report

Aberrabt activation of Hsa-miR-181d/STAT3….

Liu et al

In this research article the authors studied the effect of Garnicol in the hsa-miR-181d/STAT3 pathway in BM cells.

The paper is well-written with some typos.

The fold change of STAT3 and STAT5A (Figure 1A) is very low 1.07-1.09. This could considered not clinically relevant.

The figure 1B should be better explained: the differences in months of OS for low vs. high STAT3/STAT5A expression.

The overall idea of the paper is that the Has-miR-181d inhibit the phosphorylation levels of STAT3/STAT5A; therefore in Figure 1C, the phosphorylation levels of these proteins should also be assessed in the patient samples.

The statement of line 274-274 is too stout here: you could say only: STAT3 and STAT5A expression correlate with…..Same: lines: 405-406, 417-418

In many sections of the manuscript the term: downregulation and expression are incorrectly used. For example line 294: ….downregulated the expression of p-STAT3 …regulation is related with gene expression. here you should say “decreased the phosphorylation levels of …. This is more related with activity. Same lies: 339, 369, 409.

Figure 2: there is no explanation why you measured ERK and AKT (Fugure 2B). Same for Figure 2D when you measured apoptosis: why Bax/Bcl-xl. In this sense if you want to demonstrate that apoptosis occurs you should measure at least caspase-3 activity, of anexin-V, caspases and/or PARP-1 cleavage.

Figure 3A is not clear: better images are necessary.

The Figure 4 is the most important one of the paper. However the experiments here were not well designed: In figure 5A you made alignments of the STAT3/STAT5A to miR-181. However, you never performed experiments to demonstrated that binding (luciferase reporter assays are critical here). Instead you suggested that garnicol upregulate the miR-181d expression (which could be correct but not assessed in the paper). In turn, miR-181d somehow inhibits the phosphorylation of STAT3/STAT5A. The “major” role of microRNAs is to bind to the 3’-UTR of mRNAs to avoid protein synthesis (in most cases). So, you are bringing here a “new concept” that miRNAs inhibit protein phosphorylation. If this is the case you need to perform in vitro experiment to demonstrate that statement. Otherwise the figure 4 and Figure 5 are incorrect.

I could think that miR-181d posttranscriptionally regulate some protein (protein kinase): Garnicol increase the miR-181d; then miR-181d reduce the protein kinase levels (posttranscriptionally). As a consequence this protein cannot phosphorylate STAT3/STAT5A and thus their phosphorylation levels are reduced.

Avoid to mention figures in the Discussion section: lines 415, 427, 434

The discussion should be focus in the miR-181/STAT3/STAT5 pathway

Author Response

Q1: Reviewer #3: Aberrabt activation of Hsa-miR-181d/STAT3….

Liu et al In this research article the authors studied the effect of Garnicol in the hsa-miR-181d/STAT3 pathway in GBM cells. The paper is well-written with some typos.

A1: We thank the reviewer for the time taken to review our work, the assessment of our findings and the helpful suggestions given to help us improve the acceptability and appeal of our work. We have now carefully reviewed our manuscript for likely typographical and grammatical error. 

Q2: Reviewer #3: The fold change of STAT3 and STAT5A (Figure 1A) is very low 1.07-1.09. This could considered not clinically relevant. 

A7: We sincerely thank the reviewer for this observation. After re-analysis of the data, we discovered it was indeed statistical error which we have now correctly appropriately in our revised manuscript. Please kindly see our revised Result section, lines 311-334.

3.1. STAT-3 and STAT-5 are highly expressed in primary and recurrent glioblastoma, and their expression negatively correlates with overall survival rates

Firstly, against the background that STAT3 and STAT5 to be highly expressed in GBM cell lines (Roos et al., 2018), we evaluated the expression profiles of STAT proteins GDC TCGA-GBM cohort of 173 samples. Our results showed a 2.226-fold (t-Test = 13.114, p = 5.83 x 10-8) or 2.681-fold (t-Test = 14.037, p = 5.19 x 10-8) increase in the gene expression of STAT3 in the primary and recurrent GBM samples, respectively, compared to non-tumor samples; similarly, compared to the non-tumor control, the median expression of STAT5A was elevated by 1.492-fold (t-Test = 2.211, p = 0.036) or 2.453-fold (t-Test = 4.081, p = 0.001) the primary and recurrent GBM samples, respectively (Figure 1A). In line with the above, using the Kaplan-Meier plots and median-based high/low dichotomization of gene expression, our survival analyses showed a significantly strong association between worse OS and high STAT3 (p < 0.015) or STAT5 (p < 0.008) expression levels, as demonstrated by ~15.8% or ~11.3% survival advantage in patients with high STAT3 or STAT5, respectively, compared to the low group by year 2 after diagnosis (Figure 1B, upper). Interestingly, we also demonstrated that compared with the ~20%, 11%, or 10% OS amongst patients with STAT3lowSTAT5Alow, STAT3lowSTAT5Ahigh, or STAT3highSTAT5Alow GBM, respectively, no patient with STAT3highSTAT5Ahigh GBM was alive by year 3 after diagnosis, suggesting a strong association between STAT3highSTAT5Ahigh and GBM-specific mortality (p < 0.007) (Figure 1B, lower). Moreover, consistent with the RNA expression, results of our immunohistochemical staining confirmed elevated STAT3, pSTAT3, STAT5A, and pSTAT5A protein expression levels in primary (n = 31) and recurrent (n = 14) GBM tissues compared to non-tumor tissues (STAT3/5A vs non-tumor: ~5.3-fold, p < 0.001; pSTAT3/5A vs non-tumor: ~9.0-fold, p < 0.001 ) (Figures 1C and 1D). These results indicate, at least in part, that enhanced expression and/or activation of STAT3 and STAT5A play a critical role in the development and recurrence of GBM.

Please also kindly see our updated Figure 1 and its legend, lines 336-344.

Figure 1. STAT-3 and STAT-5 are highly expressed in primary and recurrent glioblastoma, and their expression negatively correlates with overall survival rates. (A) Graphical representation of the differential expression of STAT3 and STAT5A in primary GBM, recurrent GBM or normal brain tissues from the GDC TCGA-GBM cohort. (B) Kaplan-Meier plots of the effect of differential STAT3 or STAT5A expression on OS of patients with GBM. (C) Representative IHC images showing the differential expression of STAT3, pSTAT3, STAT5A, and pSTAT5A proteins in primary GBM, recurrent GBM or normal brain tissues. (D) Graphical representation of the differential expression of STAT3, pSTAT3, STAT5A, pSTAT5A proteins in primary GBM, recurrent GBM or normal brain tissues. *p<0.05, **p<0.01, ***p<0.001; OS, overall survival.

Q3: Reviewer #3: The figure 1B should be better explained: the differences in months of OS for low vs. high STAT3/STAT5A expression. 

A3:  We appreciate the reviewer’s comment. We have now tried to address the issue raised in our revised manuscript. Please kindly see our revised Result section, lines 311-334.

3.1. STAT-3 and STAT-5 are highly expressed in primary and recurrent glioblastoma, and their expression negatively correlates with overall survival rates

Firstly, against the background that STAT3 and STAT5 to be highly expressed in GBM cell lines (Roos et al., 2018), we evaluated the expression profiles of STAT proteins GDC TCGA-GBM cohort of 173 samples. Our results showed a 2.226-fold (t-Test = 13.114, p = 5.83 x 10-8) or 2.681-fold (t-Test = 14.037, p = 5.19 x 10-8) increase in the gene expression of STAT3 in the primary and recurrent GBM samples, respectively, compared to non-tumor samples; similarly, compared to the non-tumor control, the median expression of STAT5A was elevated by 1.492-fold (t-Test = 2.211, p = 0.036) or 2.453-fold (t-Test = 4.081, p = 0.001) the primary and recurrent GBM samples, respectively (Figure 1A). In line with the above, using the Kaplan-Meier plots and median-based high/low dichotomization of gene expression, our survival analyses showed a significantly strong association between worse OS and high STAT3 (p < 0.015) or STAT5 (p < 0.008) expression levels, as demonstrated by ~15.8% or ~11.3% survival advantage in patients with high STAT3 or STAT5, respectively, compared to the low group by year 2 after diagnosis (Figure 1B, upper). Interestingly, we also demonstrated that compared with the ~20%, 11%, or 10% OS amongst patients with STAT3lowSTAT5Alow, STAT3lowSTAT5Ahigh, or STAT3highSTAT5Alow GBM, respectively, no patient with STAT3highSTAT5Ahigh GBM was alive by year 3 after diagnosis, suggesting a strong association between STAT3highSTAT5Ahigh and GBM-specific mortality (p < 0.007) (Figure 1B, lower). Moreover, consistent with the RNA expression, results of our immunohistochemical staining confirmed elevated STAT3, pSTAT3, STAT5A, and pSTAT5A protein expression levels in primary (n = 31) and recurrent (n = 14) GBM tissues compared to non-tumor tissues (STAT3/5A vs non-tumor: ~5.3-fold, p < 0.001; pSTAT3/5A vs non-tumor: ~9.0-fold, p < 0.001 ) (Figures 1C and 1D). These results indicate, at least in part, that enhanced expression and/or activation of STAT3 and STAT5A play a critical role in the development and recurrence of GBM.

Please also kindly see our updated Figure 1 and its legend, lines 336-344.

Figure 1. STAT-3 and STAT-5 are highly expressed in primary and recurrent glioblastoma, and their expression negatively correlates with overall survival rates. (A) Graphical representation of the differential expression of STAT3 and STAT5A in primary GBM, recurrent GBM or normal brain tissues from the GDC TCGA-GBM cohort. (B) Kaplan-Meier plots of the effect of differential STAT3 or STAT5A expression on OS of patients with GBM. (C) Representative IHC images showing the differential expression of STAT3, pSTAT3, STAT5A, and pSTAT5A proteins in primary GBM, recurrent GBM or normal brain tissues. (D) Graphical representation of the differential expression of STAT3, pSTAT3, STAT5A, pSTAT5A proteins in primary GBM, recurrent GBM or normal brain tissues. *p<0.05, **p<0.01, ***p<0.001; OS, overall survival.

Q4: Reviewer #3: The figure 1B should be better explained: the differences in months of OS for low vs. high STAT3/STAT5A expression. 

A4:  We really thank the reviewer for this comment. We have now included the requested data in our revised manuscript. Please kindly see our revised Result section, Pages 7-8, Lines 311-334.

3.1. STAT-3 and STAT-5 are highly expressed in primary and recurrent glioblastoma, and their expression negatively correlates with overall survival rates

Firstly, against the background that STAT3 and STAT5 to be highly expressed in GBM cell lines (Roos et al., 2018), we evaluated the expression profiles of STAT proteins GDC TCGA-GBM cohort of 173 samples. Our results showed a 2.226-fold (t-Test = 13.114, p = 5.83 x 10-8) or 2.681-fold (t-Test = 14.037, p = 5.19 x 10-8) increase in the gene expression of STAT3 in the primary and recurrent GBM samples, respectively, compared to non-tumor samples; similarly, compared to the non-tumor control, the median expression of STAT5A was elevated by 1.492-fold (t-Test = 2.211, p = 0.036) or 2.453-fold (t-Test = 4.081, p = 0.001) the primary and recurrent GBM samples, respectively (Figure 1A). In line with the above, using the Kaplan-Meier plots and median-based high/low dichotomization of gene expression, our survival analyses showed a significantly strong association between worse OS and high STAT3 (p < 0.015) or STAT5 (p < 0.008) expression levels, as demonstrated by ~15.8% or ~11.3% survival advantage in patients with high STAT3 or STAT5, respectively, compared to the low group by year 2 after diagnosis (Figure 1B, upper). Interestingly, we also demonstrated that compared with the ~20%, 11%, or 10% OS amongst patients with STAT3lowSTAT5Alow, STAT3lowSTAT5Ahigh, or STAT3highSTAT5Alow GBM, respectively, no patient with STAT3highSTAT5Ahigh GBM was alive by year 3 after diagnosis, suggesting a strong association between STAT3highSTAT5Ahigh and GBM-specific mortality (p < 0.007) (Figure 1B, lower). Moreover, consistent with the RNA expression, results of our immunohistochemical staining confirmed elevated STAT3, pSTAT3, STAT5A, and pSTAT5A protein expression levels in primary (n = 31) and recurrent (n = 14) GBM tissues compared to non-tumor tissues (STAT3/5A vs non-tumor: ~5.3-fold, p < 0.001; pSTAT3/5A vs non-tumor: ~9.0-fold, p < 0.001 ) (Figures 1C and 1D). These results indicate, at least in part, that enhanced expression and/or activation of STAT3 and STAT5A play a critical role in the development and recurrence of GBM.

Please also kindly see our updated Figure 1 and its legend, lines 336-344.

Figure 1. STAT-3 and STAT-5 are highly expressed in primary and recurrent glioblastoma, and their expression negatively correlates with overall survival rates. (A) Graphical representation of the differential expression of STAT3 and STAT5A in primary GBM, recurrent GBM or normal brain tissues from the GDC TCGA-GBM cohort. (B) Kaplan-Meier plots of the effect of differential STAT3 or STAT5A expression on OS of patients with GBM. (C) Representative IHC images showing the differential expression of STAT3, pSTAT3, STAT5A, and pSTAT5A proteins in primary GBM, recurrent GBM or normal brain tissues. (D) Graphical representation of the differential expression of STAT3, pSTAT3, STAT5A, pSTAT5A proteins in primary GBM, recurrent GBM or normal brain tissues. *p<0.05, **p<0.01, ***p<0.001; OS, overall survival.

Q5: Reviewer #3: The statement of line 274-274 is too stout here: you could say only: STAT3 and STAT5A expression correlate with…..Same: lines: 405-406, 417-418. 

A5:  We appreciate the reviewer’s comment. As suggested by the reviewer, we have now corrected this in our revised manuscript. Please kindly see our revised Result section, Pages 7-8, Lines 311-334.

3.1. STAT-3 and STAT-5 are highly expressed in primary and recurrent glioblastoma, and their expression negatively correlates with overall survival rates

Firstly, against the background that STAT3 and STAT5 to be highly expressed in GBM cell lines (Roos et al., 2018), we evaluated the expression profiles of STAT proteins GDC TCGA-GBM cohort of 173 samples. Our results showed a 2.226-fold (t-Test = 13.114, p = 5.83 x 10-8) or 2.681-fold (t-Test = 14.037, p = 5.19 x 10-8) increase in the gene expression of STAT3 in the primary and recurrent GBM samples, respectively, compared to non-tumor samples; similarly, compared to the non-tumor control, the median expression of STAT5A was elevated by 1.492-fold (t-Test = 2.211, p = 0.036) or 2.453-fold (t-Test = 4.081, p = 0.001) the primary and recurrent GBM samples, respectively (Figure 1A). In line with the above, using the Kaplan-Meier plots and median-based high/low dichotomization of gene expression, our survival analyses showed a significantly strong association between worse OS and high STAT3 (p < 0.015) or STAT5 (p < 0.008) expression levels, as demonstrated by ~15.8% or ~11.3% survival advantage in patients with high STAT3 or STAT5, respectively, compared to the low group by year 2 after diagnosis (Figure 1B, upper). Interestingly, we also demonstrated that compared with the ~20%, 11%, or 10% OS amongst patients with STAT3lowSTAT5Alow, STAT3lowSTAT5Ahigh, or STAT3highSTAT5Alow GBM, respectively, no patient with STAT3highSTAT5Ahigh GBM was alive by year 3 after diagnosis, suggesting a strong association between STAT3highSTAT5Ahigh and GBM-specific mortality (p < 0.007) (Figure 1B, lower). Moreover, consistent with the RNA expression, results of our immunohistochemical staining confirmed elevated STAT3, pSTAT3, STAT5A, and pSTAT5A protein expression levels in primary (n = 31) and recurrent (n = 14) GBM tissues compared to non-tumor tissues (STAT3/5A vs non-tumor: ~5.3-fold, p < 0.001; pSTAT3/5A vs non-tumor: ~9.0-fold, p < 0.001 ) (Figures 1C and 1D). These results indicate, at least in part, that enhanced expression and/or activation of STAT3 and STAT5A play a critical role in the development and recurrence of GBM.

Q6: Reviewer #3: In many sections of the manuscript the term: downregulation and expression are incorrectly used. For example line 294: .downregulated the expression of p-STAT3 …regulation is related with gene expression. here you should say “decreased the phosphorylation levels of …. This is more related with activity. Same lies: 339, 369, 409. 

A6:  We appreciate the reviewer’s comment. As suggested by the reviewer, we have now corrected this in our revised manuscript. Please kindly see our revised Result section.

Q7: Reviewer #3: Figure 2: there is no explanation why you measured ERK and AKT (Fugure 2B). Same for Figure 2D when you measured apoptosis: why Bax/Bcl-xl. In this sense if you want to demonstrate that apoptosis occurs you should measure at least caspase-3 activity, of anexin-V, caspases and/or PARP-1 cleavage. 

A7: We appreciate the reviewer’s insightful comment. We have now included the requested data of V/7-AAD (or annexin V/IP) experiments in our revised manuscript. Please kindly see our revised Results section, lines 345-381.

3.2. Garcinol significantly inhibits GBM cell viability and oncogenicity through induction of STAT3/5A signaling and enhanced apoptosis

Against the background of recent work demonstrating that garcinol inhibits CSCs-like phenotype of human non-small cell lung carcinoma by suppressing the Wnt/β-catenin/STAT3 signaling axis (Huang et al., 2018), we investigated the probable STAT signaling-mediated anti-GBM effect of garcinol (Figure 2A). Firstly, to provide some mechanistic insight, we demonstrated that treatment of U87MG or GBM8401 cells with  2.5 μM or 5 μM garcinol significantly downregulated the expression of p-STAT3, p-STAT5, p-ERK, and p-AKT (Figure 2B), Synchronous with the observed inhibition of STAT3, STAT5 and AKT signaling, garcinol significantly suppressed the viability of GBM4801 and U87MG cells, with 10 μM eliciting 51% or 25% reduced viability of U87MG or GBM8401 cells, respectively, and 40 μM eliciting  94.7% reduction of U87MG and GBM8401 cell viability, indicating a dose-dependent GBM cell killing effect (Figure 2C), and this reduced viability was associated with markedly enhanced  Bax/Bcl-xL apoptotic ratio, as 2.5 μM induced a 1.67-fold (p < 0.05) or  2.7-fold (p < 0.05) increase in U87MG or GBM8401 apoptotic ratio, while  5 μM increased the apoptotic ratio by 2.83-fold (p < 0.001) or 2.92-fold (p < 0.001) in the U87MG or GBM8401 cells, respectively (Figure 2D). In addition, using the Phycoerythrin (PE)-conjugated Annexin V/7- Amino-Actinomycin (7-AAD) staining, we demonstrated that compared to the cell death in the untreated control (U87MG: 1.6%, GBM8401: 4.0%) or 20 mM pan-caspase inhibitor benzyloxycarbonyl-Val-Ala-Asp-fluoromethyl ketone (Z-VAD-FMK)-treated negative control (U87MG: 3.2%, GBM8401: 10.0%), treatment with Garcinol enhanced U87MG cell death (2.5 mM:  10.5% or 5 mM: 41.2%), while 2.5 mM or 5 mM garcinol elicited 15.1% or 32.1% apoptosis of the GBM8401 cells, respectively (Figure 2E), indicating that the Garcinol-induced cell death was apoptotic and caspase-dependent. Since the highly invasive GBM spreads fast to surrounding brain tissue, thus, contributing to its documented lethality (Omuro & DeAngelis, 2013), we sought to understand if and how garcinol affects this invasive trait. We demonstrated that treatment with 2.5 μM or 5 μM dose-dependently suppressed the migration of the U87MG (~59%, p < 0.01 or 81%, p < 0.001, respectively) and GBM8401 (~48%, p < 0.01 or 76%, p < 0.001, respectively) cells at the 24 h time-point (Figure 2F). Similarly, 2.5 μM or 5 μM garcinol induced a 60% (p < 0.01) or ~80% (p < 0.001) reduction of U87MG invasive capacity, respectively, and 39% (p < 0.01) or 60% (p < 0.001) reduction in number of invaded GBM8401 cells (Figure 2G). Furthermore, in parallel assays to confirm the anticancer role of Garcinol, consistent with earlier results, we demonstrated that treatment with 2.5 μM or 5 μM Garcinol, significantly suppressed the expression of N-cadherin, vimentin and slug proteins, while conversely upregulating the expression of E-cadherin protein (Figure 2H),thus indicating that Garcinol attenuates epithelial-to-mesenchymal transition and the metastatic phenotype of GBM cells. Together, these data suggest that garcinol significantly inhibits GBM cell viability and oncogenicity through induction of STAT3/5A and associated signaling with enhanced apoptosis.

Please also kindly see our updated Figure 2 and its legend, lines 383-399.

Figure 2. Garcinol significantly inhibits GBM cell viability and oncogenicity through induction of STAT3/5A signaling and enhanced apoptosis. (A) Chemical structure of garcinol with molecular formula C38H50O6 and molecular weight 602.80 g/mol. (B) Representative western blot photo-images of the effect of 2.5 μM – 5 μM on the expression of p-STAT3, STAT3, p-STAT5, STAT5, p-ERK, ERK, p-AKT, and AKT proteins in GBM8401 or U87MG cells. (C) Graphical representation of the effect of 2.5 μM – 40 μM on the viability of GBM8401 or U87MG cells. (D) Representative western-blot photo-images showing the effect of 2.5 μM – 5 μM on the expression of Bax and Bcl-xL proteins in GBM8401 or U87MG cells. (E) Flow-cytometry data (upper) and graphical representation (lower) showing the effect of Garcinol on U87MG cells co-stained with PE-conjugated Annexin V and 7- AAD, compared with untreated or Z-VAD-FMK -treated negative control groups. Annexin V-stained Q4 cells are early apoptotic cells, whereas Q2 cells are late stage apoptotic (necrotic) cells. Apoptosis (%), sum of Q4+Q2; CTL, vehicle-treated; Neg CTL, pan-caspase inhibitor benzyloxycarbonyl-Val-Ala-Asp-fluoromethyl ketone (Z-VAD-FMK). Representative photo-images (upper) and graphical representation (lower) of the effect of 2.5 μM or 5 μM on the (F) migration and (G) invasion of GBM8401 or U87MG cells. (H) Representative western-blot photo-images showing the effect of 2.5 μM – 5 μM on the expression of E-cadherin, N-cadherin, vimentin and slug proteins in GBM8401 or U87MG cells. *p<0.05, **p<0.01, ***p<0.001; GAPDH is loading control.

Also kindly see our revised Materials & Methods section, lines 138-146.

2.1. Drugs and Chemicals

Garcinol (sc-200891A, HPLC purity ≥95%) and Z-VAD-FMK (sc-3067, HPLC purity ≥95%) purchased from Santa Cruz Biotechnology (Santa Cruz, CA, USA) was dissolved in dimethyl sulfoxide (DMSO) to prepare a 20 mM stock and stored at −20°C until use. For different assays, the stock was further diluted using cell growth medium as appropriate. Dimethyl sulfoxide (DMSO), served as vehicle and negative control. BD Pharmingen™ PE Annexin V apoptosis detection kit I (#559763) was purchased from BD Biosciences (San Jose, CA, USA). Unless otherwise indicated, all reagents were obtained from Gibco (Thermo Fisher Scientific, Life Technologies, Foster City, CA, USA).

Please also see our revised Materials & Methods section, lines 266-276

2.13. PE-Annexin V/7-AAD cell death assay

PE-Annexin V/7-AAD staining was used for detection of cell death using the BD FACSCanto™ II flow cytometry system (BD Biosciences, San Jose, CA, USA) following the manufacturer’s instructions. Briefly, afer washing 5x105 wild type, Z-VAD-FMK-treated or Garcinol-treated U87MG cells twice with PBS, and once with annexin V binding buffer, the cells were incubated with PE-labeled annexin V and  7-AAD at room temperature for 15 min and then analyzed using flow cytometry. The mitochondrial transmembrane potential (DΨm) was evaluated using the cationic dye JC-1 (Mitochondrial Membrane Potential Assay Kit, #ab113850, Abcam plc., Cambridge, UK) in accordance with the manufacturer’s instructions (BD Pharmingen); 1x106 U87MG cells were incubated with 10 mg/mL JC-1 at 37 °C in the dark for 15 min, and then analyzed using flow cytometry. All samples were assayed three times in triplicate.

Q8: Reviewer #3: Figure 3A is not clear: better images are necessary. 

A8:  We sincerely thank the reviewer for this comment. We have provided better high-resolution images for Figure 3. Please kindly see our updated Figure 3 and its legend, lines 424-434.

Figure 3. Garcinol negatively impacts GBM stem cell-like phenotypes. (A) Representative photo-images (left) and histograms (middle, right) of the effect of 5 μM garcinol on the number and size of tumorspheres formed by U87MG or GBM8401 cells. (B) Representative images (upper) and graph (lower) of the effect of 2 μM or 5 μM garcinol on the number of colonies formed by U87MG or GBM8401 cells. (C) Representative photo-images showing the effect of 2 μM or 5 μM garcinol on the tumorsphere size and sub-cellular localization of SOX2 and OCT4 proteins in U87MG or GBM8401 cells. (D) Graphs showing how 2 μM or 5 μM affect the MIF of SOX2, OCT4, p-STAT3 and p-STAT5A in U87MG or GBM8401 cells. (E) Representative western blot images (left) and histograms (right) showing the effect of 2 μM or 5 μM garcinol on the expression of SOX2 and OCT4 proteins. *p<0.05, **p<0.01, ***p<0.001; MIF, median immunofluorescence; DAPI served as nuclear marker. GAPDH served as loading control.

Q9 Reviewer #3: The Figure 4 is the most important one of the paper. However, the experiments here were not well designed: In figure 5A you made alignments of the STAT3/STAT5A to miR-181. However, you never performed experiments to demonstrated that binding (luciferase reporter assays are critical here). Instead you suggested that garnicol upregulate the miR-181d expression (which could be correct but not assessed in the paper). In turn, miR-181d somehow inhibits the phosphorylation of STAT3/STAT5A. The “major” role of microRNAs is to bind to the 3’-UTR of mRNAs to avoid protein synthesis (in most cases). So, you are bringing here a “new concept” that miRNAs inhibit protein phosphorylation. If this is the case, you need to perform in vitro experiment to demonstrate that statement. Otherwise the figure 4 and Figure 5 are incorrect. (fig 4B, fig 5F)

A9: We sincerely thank the reviewer for this important comment. We have now included data for the 3D visualization of the interaction between miR-181d and STAT3 or STAT5A in our revised manuscript. Please kindly see our revised Results section, lines 436-477. 

3.4. Garcinol increases the expression of hsa-miR181d, which has inhibitory effects on STAT3 and STAT5 expression

Having established that garcinol impairs STAT3 and STAT5A activation, we probed for likely modulators and/or mediators of the interaction between garcinol and the STAT proteins. Hsa-miR-181d shown in Figure 4A, has been implicated in the worse OS of patients with GBM (Zhang et al., 2012). Consistent with this, using the Schrodinger PyMOL 2.3 molecular docking and visualization software (http://pymol.org) we demonstrated that hsa-miR-181d interacts with and binds directly to STAT3 (docking score = -254.49, ligand root mean square deviation (RMSD) = 195.10 Å) or STAT5A (docking score = -234.19, ligand RMSD = 143.04 Å), complementing earlier prediction that hsa-miR-181d binds with STAT3 with a mirSVR or PhastCons score of -0.26 or 0.69, respectively, while it binds with STAT5A with a mirSVR  or PhastCons score of -0.21 or 0.49, respectively (Figure 4A).  Where the mirSVR shows the likelihood of hsa-miR-181d down-regulating the target mRNA STAT3 and STAT5A based on the sequence and structure features in the miRNA/mRNA predicted target sites. Moreover, the PhastCons score shows the likelihood that the predicted miRNA/mRNA binding nucleotides are conserved. In concordance, our qRT-PCR analysis of 5 μM garcinol-treated U87MG and GBM8401 cells showed that garcinol significantly induced higher expression of miR-181d in the U87MG (2.7-fold, p < 0.01) and GBM8401 (2.1-fold, p < 0.01) cells (Figure 4B). Furthermore, having implicated STAT3/5A in enhanced migration and invasiveness of U87MG or GBM8401 cells, we examined the probable effect of hsa-miR-181d on these metastatic phenotypes of GBM cells. Using the scratch wound-healing assay, we demonstrated that compared to the untreated control or syn-mir-treated cells, treatment with mir-181d inhibitor significantly enhanced the ability of the U87MG cells to migrate (4.64-fold, p<0.01), while treatment with the mir-181d-mimic elicited marked attenuation of migration (3.80-fold, p<0.01) (Figure 4C), which is reminiscent of suppressed migration induced by 5 μM garcinol (4.17-fold, p<0.01) earlier.  Similarly, while treatment with mir-181d inhibitor significantly enhanced the invasiveness of the U87MG cells (4.63-fold, p<0.01), treatment with the mir-181d-mimic elicited profound suppression of invasion (4.15-fold, p<0.01) (Figures 4D), and this was akin to the effect of 5 mM garcinol (4.22-fold, p<0.01). To confirm a direct relationship between the STAT proteins and miR-181d expression, western blot analysis was done comparing samples exposed to mir-181d inhibitor, mir-181d-mimic, or mir-181d inhibitor/garcinol combination. The results showed that mir-181d inhibitor significantly enhanced the expression of p-STAT3, p-STAT5, N-cadherin and vimentin proteins, but suppressed E-cadherin protein expression compared to the control group, while for the mir-181d-mimic-treated cells, the p-STAT3, p-STAT5, N-cadherin, and vimentin protein expression levels were significantly lower, but E-cadherin was upregulated; For cells incubated with mir-181d inhibitor and 5 μM garcinol concomitantly, p-STAT3 and p-STAT5 protein expression levels were markedly higher than in the mir-181d-mimic group but lower than the mir-181d-inhibitor group (Figure 4E). Concomitantly, we observed that compared with the control group, syn-mir-treated cells, or even mir-181d inhibitor-treated cells, treatment with mir-181d-mimic markedly suppressed the expression of JAK2 protein (2.22-fold, p<0.01), akin to the effect elicited by treatment with concurrently with mir-181d inhibitor and 5 μM garcinol (2.94-fold, p<0.01), which is consistent with the results above and suggestive of a miR-181d-mediated JAK2-modulated phosphorylation of STAT3/5A. These results indicate that garcinol can activate mir-181d activity which suppresses STAT3/5A activation.    

Also kindly see our updated Figure 4 and its legend, lines 479-491.

Figure 4. Garcinol increases the expression of hsa-miR181d, which has inhibitory effects on STAT3 and STAT5 expression. (A) 3-dimensional visualization of direct interaction between hsa-miR-181d and STAT3 (left) or STAT5A (middle) and images showing the complementary sequence alignment of hsa-miR-181d with STAT3 (upper right) or STAT5A (lower right). The mirSVR and PhastCons scores are indicated. (B) Histograms of the effect of 5 μM on hsa-miR-181d expression in U87MG or GBM8401 cells. (C) Representative images (left) and histograms (right) showing the effect of Syn-mir-CTL, mir-181d-inhibitor, or mir-181d-mimic on the migration of U87MG cells over 24 h, as determined by wound-healing assay. (D) Representative images (left) and histograms (right) showing the effect of Syn-mir-CTL, mir-181d-inhibitor, or mir-181d-mimic on the invasion of U87MG cells. (E) Representative western-blot photo-images (left) and histograms (right) comparing the effect of Syn-mir-CTL, mir-181d-inhibitor, mir-181d-mimic, or garcinol on the expression level of JAK2, p-STAT3, STAT3, p-STAT5, STAT5, E-cadherin, N-cadherin, and vimentin proteins in U87MG or GBM8401 cells. *p<0.05, **p<0.01, ***p<0.001; GAPDH is loading control.

Q10 Reviewer #3: I could think that miR-181d posttranscriptionally regulate some protein (protein kinase): Garnicol increase the miR-181d; then miR-181d reduce the protein kinase levels (posttranscriptionally). As a consequence this protein cannot phosphorylate STAT3/STAT5A and thus their phosphorylation levels are reduced..

A10: We sincerely thank the reviewer for this insightful comment. We do agree with the reviewer on this possibility and actually probed it. Please kindly see our updated Figure 4 and its legend, lines 479-491.

Figure 4. Garcinol increases the expression of hsa-miR181d, which has inhibitory effects on STAT3 and STAT5 expression. (A) 3-dimensional visualization of direct interaction between hsa-miR-181d and STAT3 (left) or STAT5A (middle) and images showing the complementary sequence alignment of hsa-miR-181d with STAT3 (upper right) or STAT5A (lower right). The mirSVR and PhastCons scores are indicated. (B) Histograms of the effect of 5 μM on hsa-miR-181d expression in U87MG or GBM8401 cells. (C) Representative images (left) and histograms (right) showing the effect of Syn-mir-CTL, mir-181d-inhibitor, or mir-181d-mimic on the migration of U87MG cells over 24 h, as determined by wound-healing assay. (D) Representative images (left) and histograms (right) showing the effect of Syn-mir-CTL, mir-181d-inhibitor, or mir-181d-mimic on the invasion of U87MG cells. (E) Representative western-blot photo-images (left) and histograms (right) comparing the effect of Syn-mir-CTL, mir-181d-inhibitor, mir-181d-mimic, or garcinol on the expression level of JAK2, p-STAT3, STAT3, p-STAT5, STAT5, E-cadherin, N-cadherin, and vimentin proteins in U87MG or GBM8401 cells. *p<0.05, **p<0.01, ***p<0.001; GAPDH is loading control.

Q11 Reviewer #3: Avoid mentioning figures in the Discussion section: lines 415, 427, 434

The discussion should be focus in the miR-181/STAT3/STAT5 pathway

A11: We thank the reviewer for this request. We have now corrected this in our revised manuscript. Please kindly see our revised Discussion section, lines 576-598.

This study also demonstrated for the first time to the best of our knowledge, that garcinol-induced suppression of STAT3 and STAT5A is associated with significant upregulation of hsa-miR-181d expression, in vitro and in vivo; interestingly we also showed direct interaction between hsa-miR-181d and STAT3 or STAT5A protein. We posit that upon treatment with garcinol, miR-181d binds directly to the coding region of STAT3 or STAT5A mRNA, eliciting STAT3/5A degradation, and consequently reduce the expression and/or activation of STAT3 or STAT5A in the GBM cells. This demonstrated tumor suppressor role of hsa-miR-181d is consistent with findings showing that overexpression of miR-181d significantly suppressed esophageal squamous cell carcinoma (ESCC) by downredulating Derlin-1, inhibiting cancerous cell proliferation, migration and cell cycle progression in vitro, as well as inhibiting tumorigenicity in vivo (Li D et al., 2016), as well as in glioma samples and cell lines, where ectopic expression of miR-181d suppressed proliferation and induced cell cycle arrest and apoptosis by targeting K-ras and Bcl-2 (Wang et al., 2012). This demonstrated garcinol-modulated miR-181d/STAT3/5A signaling axis is of therapeutic relevance, considering that well documented role of the JAK-STAT signaling pathway, and more particularly its molecular effectors namely STAT3 and STAT5 which act as a point of convergence for several signaling pathways in cancerous cells and oncogenic processes (Luo and Balko, 2019). It is worth mentioning however, that while we cannot fully explain how garcinol induced hsa-miR-181d inhibited the activation of STAT3 and STAT5, our data finds some corroboration in increasingly documented role of miRs in the (de)activation of the JAK-STAT signaling (Zhuang G et al., 2012, Lam et al, 2013, Liu X et al., 2018), and of particular interest is miR-204 which similarly had very insignificant effect on total STAT3 expression, but impaired STAT3 phosphorylation, consequently inducing cancerous cell apoptosis and suppressed cell proliferation, migration in vitro and tumor growth in vivo (Liu X et al., 2018)

Reviewer 4 Report

We read with interest the paper by Liu et al. entitled “Aberrant Activation of Hsa-miR-181d/STAT3 and Hsa-miR-181d/5A Ratios Mediate the Anticancer Effect of Garcinol in STAT3/5A-Addicted GBM”.

The authors assessed in this paper the potential antitumor effect of Garcinol on the survival, colony/sphere forming potential and and migration/invasion of GBM cells.

Although some of the findings are definitely interesting, this paper needs major revisions prior to being published.

Several major critics apply:

Not all experiments and analyses are (adequately) described in the material and methods section (e.g., the miRNA forced expression experiments are not even mentioned, the survival statistics and the definition of the cut-off values for high/low expression groups fore the survival analysis are not defined, the number of patients in the dataset for the gene expression experiments in the TCGA/GDC set is mentioned repeatedly with 173 patients, but the dataset on the website is only 172, the scoring method of the FFPE samples (and who did it) is not described, nor the cells (tumor? Microglia?) that stain positive); This work is performed on cell lines only (both commercially purchased) and not assessed on primary GBM cultures- the main findings should be reproduced on some primary cultures; These is no in vivo data on tumor models (ideally, survival experiments on in situ primary GBM xenografts ) in order to assess the durability of the effect and its potential relevance in patients; In the results: first paragraph: the authors state that “Interestingly, we also demonstrated that compared with the ~20%, 11%, or 10% OS amongst patients with STAT3lowSTAT5Alow, STAT3lowSTAT5Ahigh, or STAT3highSTAT5Alow GBM, respectively, no patient with STAT3highSTAT5Ahigh GBM was alive, suggesting a strong association between STAT3highSTAT5Ahigh and GBM-specific mortality (p < 0.007) “: what exactly was done here for an analysis? The thresholds are set arbitrarily and this seems at best cherry pikking the results to support an hypothesis- please discard; Paragraph 3.2: the authors conclude that there was an increased apoptosis following garcinol treatment, but do not show direct evidence of this mechanism, only a change in the expression ratios of pro vs anti-apoptotic Bcl family proteins- A TUNEL or FACS analysis of apoptosis is needed her to support the statement; The rationale to look at miR181-d as the link between garcinol, STAT3/5 phosphorylation remains obscure- given the fact that the expression of total STAT3  and STAT5 seem to remain unchanged, one would expect to look into a direct effect of garcinol on kinases (btw, ERK and Akt also vary and are MAJOR drivers of GBM proliferation and aopotosis resistance…)- Why on earth look into a miRNA, and how can the authors explain that the miRNA only targets the phosphorylation of STAT3/5, not the protein? Is the expression/editing of miR181-d under control of STAT3/5 itself? Can the authors provide time-course experiemnts to see if the control of mir181-d precedes the alteration of STAT3/5 phosphorylation? The discussion section should discuss the results and the mechanisms, not just speculate on unsupported conclusions- in particular, the statement that “Of translational relevance, we observed no significant difference in OS time between the STAT3highSTAT5Alow cohort and the STAT3lowSTAT5Ahigh cohort, suggesting there is no specific advantage in inhibiting the expression of either transcription factor, and highlighting probable therapeutic efficacy of parallel inhibition of both STAT3 and STAT5A. Thus, as indicated by the dismal OS of patients with STAT3highSTAT5high compared to the other groups (Figure 1), we, for the first time to the best of our knowledge demonstrate that concerted targeting of the aberrant expression and/or activity of STAT3/5A in GBM, both at protein and mRNA levels, is essential for any meaningful curative effect in STAT-based targeted therapy for patients with primary or even recurrent GBM.” Is totally unsupported by the results and is purely speculative. The title of the article is misleading as the authors have found a link between the phosphorylation of STAT3/5 and miR181-d, not STAT3/5 themselves- also, the ‘anticancer’ effect of garcinol is only evidenced in vitro.

Other, relevant critics/questions:

GBM that are re-operated tend in general to live longer than those that are not re-operated- however, in their cohort, the authors find that the recurrent tumors still show a major activation of STAT3 and 5- how can theytreconcile this with the survival data? (low stat = longer survival) in their results? Please discus; Figure 1: please define the Q scores? what were the numbers of patients in the FFPE cohort (primary vs recurrent)? were there paired samples from the same patients? if yes, where there any differences in the expression of P-STAT3/5 between primary and recurrent tumors? Paragraph 3.2: the authors conclude that there was an increased apoptosis following garcinol treatment, but do not show direct evidence of this mechanism, only a change in the expression ratios of pro vs anti-apoptotic Bcl family proteins- A TUNEL or FACS analysis of apoptosis is needed her to support the statement; While the decreases in invasion are proportionally bigger than the differences in survival at the same garcinol dosage, one cannot rule out that a significant proportion of the differences in invasion are due to a reduced cell proliferation- the crossing cells should be normalized with respect to the non crossing cells at the end of the experiments Scratch assays: please provide better pictures and discuss the role of proliferation as well; Figure 3C: the spheres do not seem to have the same size- please choose spheres of similar size or improve the picture in order to better see the size of the sphres (add phase contrast pictures)- indeed, spheres of variable size may have different expression of stem cell markers;

Author Response

Q1: Reviewer #4: We read with interest the paper by Liu et al. entitled “Aberrant Activation of Hsa-miR-181d/STAT3 and Hsa-miR-181d/5A Ratios Mediate the Anticancer Effect of Garcinol in STAT3/5A-Addicted GBM”.

The authors assessed in this paper the potential antitumor effect of Garcinol on the survival, colony/sphere forming potential and and migration/invasion of GBM cells.

Although some of the findings are definitely interesting, this paper needs major revisions prior to being published.

A1: We sincerely thank the reviewer for the time taken to review our work and for the important suggestions made to improve the quality of our work.

Q2: Reviewer #4: Not all experiments and analyses are (adequately) described in the material and methods section (e.g., the miRNA forced expression experiments are not even mentioned, the survival statistics and the definition of the cut-off values for high/low expression groups fore the survival analysis are not defined.  

A2: We thank the reviewer for these comments. Making use of the reviewer’s comments, we have now improved our Material and Methods section by including all initially omitted methods. Please kindly see our revised Materials & Methods section, lines 277-289.

2.14. Mir-181 transfection assay

U87MG cells (5 x 104) were seeded in 24-well plates 24 h  before transfection. Transfection with mi-181d control, Syn-mir, inhibitor or mimic was performed using Hiperfect Transfection Reagent (#301705, QiIAGEN Inc., Germantown, MD, USA) following the manufacturer’s instructions. miScript inhibitor negative control (#1027272), Syn-hsa-miRNA-181d-5p miScript miRNA mimic (#MSY0002821),  Anti- hsa-miRNA-181d-5p miScript miRNA inhibitor (#MIN0002821), and Hs_miR-181d_2 miScript Primer assay (#MS00031500) were purchased from QIAGEN Inc. (Germantown, MD, USA). Briefly, after adding 3 µl  Hiperfect transfection reagent to 100 µl FBS-free DMEM containing miR-181d mimic or negative control to a final concentration of 50 nmol/L, culture media volume in the wells were  adjusted to 600 µL using medium supplemented with 10% FBS. For evaluation of transfection efficiency, green fluorescent protein (GFP) expression was monitored under fluorescence microscope. We also measured miR181d  expression in transfected U87MG cells by real-time PCR.

Regarding “the definition of the cut-off values for high/low expression groups for the survival analysis”, Please kindly see our revised Results section, lines 311-334.

3.1. STAT-3 and STAT-5 are highly expressed in primary and recurrent glioblastoma, and their expression negatively correlates with overall survival rates

Firstly, against the background that STAT3 and STAT5 to be highly expressed in GBM cell lines (Roos et al., 2018), we evaluated the expression profiles of STAT proteins GDC TCGA-GBM cohort of 173 samples. Our results showed a 2.226-fold (t-Test = 13.114, p = 5.83 x 10-8) or 2.681-fold (t-Test = 14.037, p = 5.19 x 10-8) increase in the gene expression of STAT3 in the primary and recurrent GBM samples, respectively, compared to non-tumor samples; similarly, compared to the non-tumor control, the median expression of STAT5A was elevated by 1.492-fold (t-Test = 2.211, p = 0.036) or 2.453-fold (t-Test = 4.081, p = 0.001) the primary and recurrent GBM samples, respectively (Figure 1A). In line with the above, using the Kaplan-Meier plots and median-based high/low dichotomization of gene expression, our survival analyses showed a significantly strong association between worse OS and high STAT3 (p < 0.015) or STAT5 (p < 0.008) expression levels, as demonstrated by ~15.8% or ~11.3% survival advantage in patients with high STAT3 or STAT5, respectively, compared to the low group by year 2 after diagnosis (Figure 1B, upper). Interestingly, we also demonstrated that compared with the ~20%, 11%, or 10% OS amongst patients with STAT3lowSTAT5Alow, STAT3lowSTAT5Ahigh, or STAT3highSTAT5Alow GBM, respectively, no patient with STAT3highSTAT5Ahigh GBM was alive by year 3 after diagnosis, suggesting a strong association between STAT3highSTAT5Ahigh and GBM-specific mortality (p < 0.007) (Figure 1B, lower). Moreover, consistent with the RNA expression, results of our immunohistochemical staining confirmed elevated STAT3, pSTAT3, STAT5A, and pSTAT5A protein expression levels in primary (n = 31) and recurrent (n = 14) GBM tissues compared to non-tumor tissues (STAT3/5A vs non-tumor: ~5.3-fold, p < 0.001; pSTAT3/5A vs non-tumor: ~9.0-fold, p < 0.001 ) (Figures 1C and 1D). These results indicate, at least in part, that enhanced expression and/or activation of STAT3 and STAT5A play a critical role in the development and recurrence of GBM.

Q3: Reviewer #4: the number of patients in the dataset for the gene expression experiments in the TCGA/GDC set is mentioned repeatedly with 173 patients, but the dataset on the website is only 172.

A3: We thank the reviewer for these comments. However, we politely inform the reviewer that the number of STAT expression-relevant samples TCGA/GDC GBM dataset as at the time of initial submission was indeed 173 (not 172) samples and remains so as at the time of submitting this revised manuscript. Please kindly see our revised Materials and Methods section, lines 147-152.

2.2. Analyses of Cancer RNAseq Dataset

The Cancer Genome Atlas (TCGA) GDC-TCGA glioblastoma (GBM) cohort (n = 173) used for STAT3 and STAT5A gene expression profiling and correlative studies, was accessed, downloaded and analyzed using the University of California Santa Cruz (UCSC) Xena functional genomics explorer platform (https://xenabrowser.net/heatmap/#) . The dataset consists of non-tumor (n =5), primary GBM (n =155) and recurrent GBM (n=13).

Please also see our revised Results section, lines 311-334.

3.1. STAT-3 and STAT-5 are highly expressed in primary and recurrent glioblastoma, and their expression negatively correlates with overall survival rates

Firstly, against the background that STAT3 and STAT5 to be highly expressed in GBM cell lines (Roos et al., 2018), we evaluated the expression profiles of STAT proteins GDC TCGA-GBM cohort of 173 samples. Our results showed a 2.226-fold (t-Test = 13.114, p = 5.83 x 10-8) or 2.681-fold (t-Test = 14.037, p = 5.19 x 10-8) increase in the gene expression of STAT3 in the primary and recurrent GBM samples, respectively, compared to non-tumor samples; similarly, compared to the non-tumor control, the median expression of STAT5A was elevated by 1.492-fold (t-Test = 2.211, p = 0.036) or 2.453-fold (t-Test = 4.081, p = 0.001) the primary and recurrent GBM samples, respectively (Figure 1A). In line with the above, using the Kaplan-Meier plots and median-based high/low dichotomization of gene expression, our survival analyses showed a significantly strong association between worse OS and high STAT3 (p < 0.015) or STAT5 (p < 0.008) expression levels, as demonstrated by ~15.8% or ~11.3% survival advantage in patients with high STAT3 or STAT5, respectively, compared to the low group by year 2 after diagnosis (Figure 1B, upper). Interestingly, we also demonstrated that compared with the ~20%, 11%, or 10% OS amongst patients with STAT3lowSTAT5Alow, STAT3lowSTAT5Ahigh, or STAT3highSTAT5Alow GBM, respectively, no patient with STAT3highSTAT5Ahigh GBM was alive by year 3 after diagnosis, suggesting a strong association between STAT3highSTAT5Ahigh and GBM-specific mortality (p < 0.007) (Figure 1B, lower). Moreover, consistent with the RNA expression, results of our immunohistochemical staining confirmed elevated STAT3, pSTAT3, STAT5A, and pSTAT5A protein expression levels in primary (n = 31) and recurrent (n = 14) GBM tissues compared to non-tumor tissues (STAT3/5A vs non-tumor: ~5.3-fold, p < 0.001; pSTAT3/5A vs non-tumor: ~9.0-fold, p < 0.001 ) (Figures 1C and 1D). These results indicate, at least in part, that enhanced expression and/or activation of STAT3 and STAT5A play a critical role in the development and recurrence of GBM.

Please also kindly see our updated Figure 1 and its legend, lines 336-344.

Figure 1. STAT-3 and STAT-5 are highly expressed in primary and recurrent glioblastoma, and their expression negatively correlates with overall survival rates. (A) Graphical representation of the differential expression of STAT3 and STAT5A in primary GBM, recurrent GBM or normal brain tissues from the GDC TCGA-GBM cohort. (B) Kaplan-Meier plots of the effect of differential STAT3 or STAT5A expression on OS of patients with GBM. (C) Representative IHC images showing the differential expression of STAT3, pSTAT3, STAT5A, and pSTAT5A proteins in primary GBM, recurrent GBM or normal brain tissues. (D) Graphical representation of the differential expression of STAT3, pSTAT3, STAT5A, pSTAT5A proteins in primary GBM, recurrent GBM or normal brain tissues. *p<0.05, **p<0.01, ***p<0.001; OS, overall survival.

Q4: Reviewer #4: the scoring method of the FFPE samples (and who did it) is not described, nor the cells (tumor? Microglia?) that stain positive).

A4: We thank the reviewer for this very important comment. As implied by the reviewer, we have now described the IHC scoring method in our revised manuscript. Please kindly see our revised Materials and Methods section, lines 189-205.

2.6. Immunohistochemical (IHC) Staining

The study was approved by the Joint Institutional Review Board (JIRB) of the Taipei Medical University –Shuang Ho Hospital (Approval number: N201903047). Tissue samples from patients with primary and recurrent GBM were obtained from the Taipei Medical University-Shuang Ho Hospital GBM cohort (n = 45). After de-waxing the paraffin-embedded 4 μm tissue sections using xylene for 5 min twice and re-hydrating with 100% ethanol twice for 5 min, 95% ethanol for 5 min and 80% ethanol for 5 min, 3% hydrogen peroxide (H2O2) (TA-125-H2O2Q, Thermo Fisher Scientific, Waltham, MA, USA) was used to block endogenous peroxidase activity for 10 min. The sections were then immersed in 10 mmol/L ethylenediaminetetraacetic acid (EDTA) for 3 min in a pressure cooker, then blocked with 10% normal serum. Thereafter, tissue samples were incubated with primary antibody against STAT3 (1:200) or STAT5 (1:200) at 4oC overnight, and then with biotin-labeled secondary antibody at room temperature for 1h. Sections were incubated in diaminobenzidine (DAB) and then counterstained with hematoxylin. Visualization was done under a light microscope. For staining determination, we used the Quick (Q) score formula: Q = [percentage of stained/positive cells] x [intensity of staining] The maximum score for percentage/distribution of stained/positive cells was 100, while the intensity of staining was delineated into weak (1), moderate (2), or strong (3), making the obtainable maximum Q score = 300.

Q5: Reviewer #4: This work is performed on cell lines only (both commercially purchased) and not assessed on primary GBM cultures- the main findings should be reproduced on some primary cultures; These is no in vivo data on tumor models (ideally, survival experiments on in situ primary GBM xenografts ) in order to assess the durability of the effect and its potential relevance in patients. 

A5: We thank the reviewer for this comment. As suggested by the reviewer, we have now included our in vivo studies data in our revised manuscript. Please kindly see our revised Materials and Methods section, lines 218-233.

2.15. Mice Tumor Xenograft Study

The animal study protocol was approved by the Animal Care and User Committee at Taipei Medical University (Affidavit of Approval of Animal Use Protocol # Taipei Medical University- LAC-2017-0512) consistent with the U.S. National Institutes of Health Guide for the Care and Use of Laboratory Animals. NOD/SCID mice purchased from BioLASCO (BioLASCO Taiwan Co., Ltd. Taipei, Taiwan) were all inoculated with 1 × 106 U87MG cells in their hind flank subcutaneously, and then randomly placed into untreated control (0.1% DMSO, 100 μL daily; n = 3) or 1 mg/kg garcinol-treated (1 mg/kg body weight, suspended in 0.1% DMSO, intraperitoneal [i.p.] injection, five times/week; n = 3) group. Tumor growth was measured bi-weekly for 4 weeks after tumor cell inoculation using calipers and tumor volume (v) calculated using the formula: v = l × w2 × 0.5, where l is longest diameter, and w is shortest diameter of tumor. Thereafter, the mice were humanely sacrificed, and tumor samples extracted for further comparative immunohistochemistry (IHC) and miRNA analyses.

Also see our revised Results section, lines 492-511.

3.5. Garcinol inhibits tumor growth in GBM mice models through inversely correlated STAT3/5A and hsa-miR-181d expressions

Having shown that treatment with garcinol suppresses the cancer stem cell-like phenotype of U87MG and GBM8401 cells in vitro, to determine the probable suppressive effect of garcinol on the formation and growth of tumor, in vivo, we generated NOD/SCID mice  xenograft models derived by inoculation with 1 × 106 U87MG cells subcutaneously in the hind-flank. Mice were randomly placed into control (n = 3) or garcinol treatment (n = 3) group. We demonstrated that treatment with 1 mg/kg garcinol significantly reduced the size of tumors formed in the treated mice, compared to the untreated control group (U87MG: ~7.1-fold smaller, p < 0.001 by week 4) (Figures 5A), without adversely affecting the mice body weight (Figure 5B). In subsequent experiments using protein lysates derived from the tumors extracted from the untreated and garcinol-treated mice, we demonstrated that compared to the untreated control group, STAT3, pSTAT3, STAT5A, p-STAT5A, Ki-67, and Bcl-xL protein expression levels were concomitantly suppressed, while Bax expression was significantly enhanced (Figure 5C). Moreover, for tumors extracted from the 1 mg/kg garcinol-treated U87MG tumor-bearing mice, compared to the untreated control group, STAT3, or STAT5A mRNA expression were suppressed by 4-fold (p < 0.01) or 3.87-fold (p < 0.01), while miR-181d expression was enhanced by 3.52-fold (p < 0.01) in the U87MG mice treated with 1 mg/kg garcinol (Figure 5D). These findings indicate that garcinol inhibits tumorigenicity and growth of GBM by abrogating STAT3/5A signaling, and upregulating hsa-miR-181d, with concomitant suppression of Ki-67 proliferation index and enhancement of Bax/Bcl-xL apoptotic ratio, in vivo.

Also kindly see our updated Figure 5 and its legend, lines 513-520.

Figure 5. Garcinol inhibits tumor growth in GBM mice models through inversely correlated STAT3/5A and hsa-miR-181d expressions. Representative image and graph showing the effect of garcinol on the (A) tumor volume and (B) body weight of U87MG-tumor-bearing mice. p-values were determined by 2-way ANOVA. (C) Representative images and histograms of the differential expression of STAT3, pSTAT3, STAT5A, pSTAT5A, Ki-67, Bax, and Bcl-xL proteins level in tumors extracted from mice bearing U87MG cell-derived tumors, treated with or without garcinol. (D) Histograms showing the effect of garcinol treatment on STAT3, STAT5A and miR-181d expression levels in U87MG-tumor-bearing mice. *p<0.05, **p<0.01, ***p<0.001; GAPDH is loading control.

Q6: Reviewer #4: In the results: first paragraph: the authors state that “Interestingly, we also demonstrated that compared with the ~20%, 11%, or 10% OS amongst patients with STAT3lowSTAT5Alow, STAT3lowSTAT5Ahigh, or STAT3highSTAT5Alow GBM, respectively, no patient with STAT3highSTAT5Ahigh GBM was alive, suggesting a strong association between STAT3highSTAT5Ahigh and GBM-specific mortality (p < 0.007) “: what exactly was done here for an analysis? The thresholds are set arbitrarily and this seems at best cherry pikking the results to support an hypothesis- please discard

A6:  We thank the reviewer for this comment. We apologize for the lack of clarity initially. We have now rephrased the paragraph for clarity. Please kindly see our revised Results section, lines 311-334.

3.1. STAT-3 and STAT-5 are highly expressed in primary and recurrent glioblastoma, and their expression negatively correlates with overall survival rates

Firstly, against the background that STAT3 and STAT5 to be highly expressed in GBM cell lines (Roos et al., 2018), we evaluated the expression profiles of STAT proteins GDC TCGA-GBM cohort of 173 samples. Our results showed a 2.226-fold (t-Test = 13.114, p = 5.83 x 10-8) or 2.681-fold (t-Test = 14.037, p = 5.19 x 10-8) increase in the gene expression of STAT3 in the primary and recurrent GBM samples, respectively, compared to non-tumor samples; similarly, compared to the non-tumor control, the median expression of STAT5A was elevated by 1.492-fold (t-Test = 2.211, p = 0.036) or 2.453-fold (t-Test = 4.081, p = 0.001) the primary and recurrent GBM samples, respectively (Figure 1A). In line with the above, using the Kaplan-Meier plots and median-based high/low dichotomization of gene expression, our survival analyses showed a significantly strong association between worse OS and high STAT3 (p < 0.015) or STAT5 (p < 0.008) expression levels, as demonstrated by ~15.8% or ~11.3% survival advantage in patients with high STAT3 or STAT5, respectively, compared to the low group by year 2 after diagnosis (Figure 1B, upper). Interestingly, we also demonstrated that compared with the ~20%, 11%, or 10% OS amongst patients with STAT3lowSTAT5Alow, STAT3lowSTAT5Ahigh, or STAT3highSTAT5Alow GBM, respectively, no patient with STAT3highSTAT5Ahigh GBM was alive by year 3 after diagnosis, suggesting a strong association between STAT3highSTAT5Ahigh and GBM-specific mortality (p < 0.007) (Figure 1B, lower). Moreover, consistent with the RNA expression, results of our immunohistochemical staining confirmed elevated STAT3, pSTAT3, STAT5A, and pSTAT5A protein expression levels in primary (n = 31) and recurrent (n = 14) GBM tissues compared to non-tumor tissues (STAT3/5A vs non-tumor: ~5.3-fold, p < 0.001; pSTAT3/5A vs non-tumor: ~9.0-fold, p < 0.001 ) (Figures 1C and 1D). These results indicate, at least in part, that enhanced expression and/or activation of STAT3 and STAT5A play a critical role in the development and recurrence of GBM.

Please also kindly see our updated Figure 1 and its legend, lines 336-344.

Figure 1. STAT-3 and STAT-5 are highly expressed in primary and recurrent glioblastoma, and their expression negatively correlates with overall survival rates. (A) Graphical representation of the differential expression of STAT3 and STAT5A in primary GBM, recurrent GBM or normal brain tissues from the GDC TCGA-GBM cohort. (B) Kaplan-Meier plots of the effect of differential STAT3 or STAT5A expression on OS of patients with GBM. (C) Representative IHC images showing the differential expression of STAT3, pSTAT3, STAT5A, and pSTAT5A proteins in primary GBM, recurrent GBM or normal brain tissues. (D) Graphical representation of the differential expression of STAT3, pSTAT3, STAT5A, pSTAT5A proteins in primary GBM, recurrent GBM or normal brain tissues. *p<0.05, **p<0.01, ***p<0.001; OS, overall survival.

Q7: Reviewer #4: Paragraph 3.2: the authors conclude that there was an increased apoptosis following garcinol treatment, but do not show direct evidence of this mechanism, only a change in the expression ratios of pro vs anti-apoptotic Bcl family proteins- A TUNEL or FACS analysis of apoptosis is needed her to support the statement

A7:  We thank the reviewer for this insightful comment. As suggested by the reviewer, we have now included additional data in our revised manuscript. Please kindly see our revised Results section, lines 345-381.

3.2. Garcinol significantly inhibits GBM cell viability and oncogenicity through induction of STAT3/5A signaling and enhanced apoptosis

Against the background of recent work demonstrating that garcinol inhibits CSC-like phenotype of human non-small cell lung carcinoma by suppressing the Wnt/β-catenin/STAT3 signaling axis (Huang et al., 2018), we investigated the probable STAT signaling-mediated anti-GBM effect of garcinol (Figure 2A). Firstly, to provide some mechanistic insight, we demonstrated that treatment of U87MG or GBM8401 cells with 2.5 μM or 5 μM garcinol significantly downregulated the expression of p-STAT3, p-STAT5, p-ERK, and p-AKT (Figure 2B), Synchronous with the observed inhibition of STAT3, STAT5 and AKT signaling, garcinol significantly suppressed the viability of GBM4801 and U87MG cells, with 10 μM eliciting 51% or 25% reduced viability of U87MG or GBM8401 cells, respectively, and 40 μM eliciting  94.7% reduction of U87MG and GBM8401 cell viability, indicating a dose-dependent GBM cell killing effect (Figure 2C), and this reduced viability was associated with markedly enhanced  Bax/Bcl-xL apoptotic ratio, as 2.5 μM induced a 1.67-fold (p < 0.05) or  2.7-fold (p < 0.05) increase in U87MG or GBM8401 apoptotic ratio, while  5 μM increased the apoptotic ratio by 2.83-fold (p < 0.001) or 2.92-fold (p < 0.001) in the U87MG or GBM8401 cells, respectively (Figure 2D). In addition, using the Phycoerythrin (PE)-conjugated Annexin V/7- Amino-Actinomycin (7-AAD) staining, we demonstrated that compared to the cell death in the untreated control (U87MG: 1.6%, GBM8401: 4.0%) or 20 M pan-caspase inhibitor benzyloxycarbonyl-Val-Ala-Asp-fluoromethyl ketone (Z-VAD-FMK)-treated negative control (U87MG: 3.2%, GBM8401: 10.0%), treatment with Garcinol enhanced U87MG cell death (2.5 M:  10.5% or 5 M: 41.2%), while 2.5 M or 5 M garcinol elicited 15.1% or 32.1% apoptosis of the GBM8401 cells, respectively (Figure 2E), indicating that the Garcinol-induced cell death was apoptotic and caspase-dependent. Since the highly invasive GBM spreads fast to surrounding brain tissue, thus, contributing to its documented lethality (Omuro & DeAngelis, 2013), we sought to understand if and how garcinol affects this invasive trait. We demonstrated that treatment with 2.5 μM or 5 μM dose-dependently suppressed the migration of the U87MG (~59%, p < 0.01 or 81%, p < 0.001, respectively) and GBM8401 (~48%, p < 0.01 or 76%, p < 0.001, respectively) cells at the 24 h time-point (Figure 2F). Similarly, 2.5 μM or 5 μM garcinol induced a 60% (p < 0.01) or ~80% (p < 0.001) reduction of U87MG invasive capacity, respectively, and 39% (p < 0.01) or 60% (p < 0.001) reduction in number of invaded GBM8401 cells (Figure 2G). Furthermore, in parallel assays to confirm the anticancer role of garcinol, consistent with earlier results, we demonstrated that treatment with 2.5 M or 5 M garcinol, significantly suppressed the expression of N-cadherin, vimentin and slug proteins, while conversely upregulating the expression of E-cadherin protein (Figure 2H), thus indicating that garcinol attenuates epithelial-to-mesenchymal transition (EMT) and the metastatic phenotype of GBM cells. Together, these data suggest that garcinol significantly inhibits GBM cell viability and oncogenicity through induction of STAT3/5A and associated signaling with enhanced apoptosis.

Please kindly see our updated Figure 2 and its legend, lines 383-395.

Figure 2. Garcinol significantly inhibits GBM cell viability and oncogenicity through induction of STAT3/5A signaling and enhanced apoptosis. (A) Chemical structure of garcinol with molecular formula C38H50O6 and molecular weight 602.80 g/mol. (B) Representative western blot photo-images of the effect of 2.5 μM – 5 μM on the expression of p-STAT3, STAT3, p-STAT5, STAT5, p-ERK, ERK, p-AKT, and AKT proteins in GBM8401 or U87MG cells. (C) Graphical representation of the effect of 2.5 μM – 40 μM on the viability of GBM8401 or U87MG cells. (D) Representative western-blot photo-images showing the effect of 2.5 μM – 5 μM on the expression of Bax and Bcl-xL proteins in GBM8401 or U87MG cells. (E) Flow-cytometry data (upper) and graphical representation (lower) showing the effect of Garcinol on U87MG cells co-stained with PE-conjugated Annexin V and 7- AAD, compared with untreated or Z-VAD-FMK -treated negative control groups. Annexin V-stained Q4 cells are early apoptotic cells, whereas Q2 cells are late stage apoptotic (necrotic) cells. Apoptosis (%), sum of Q4+Q2; CTL, vehicle-treated; Neg CTL, pan-caspase inhibitor benzyloxycarbonyl-Val-Ala-Asp-fluoromethyl ketone (Z-VAD-FMK). Representative photo-images (upper) and graphical representation (lower) of the effect of 2.5 μM or 5 μM on the (F) migration and (G) invasion of GBM8401 or U87MG cells. (H) Representative western-blot photo-images showing the effect of 2.5 μM – 5 μM on the expression of E-cadherin, N-cadherin, vimentin and slug proteins in GBM8401 or U87MG cells. *p<0.05, **p<0.01, ***p<0.001; GAPDH is loading control.

Also kindly see our revised Materials and Methods section, lines 266-276.

2.13. PE-Annexin V/7-AAD cell death assay

PE-Annexin V/7-AAD staining was used for detection of cell death using the BD FACSCanto™ II flow cytometry system (BD Biosciences, San Jose, CA, USA) following the manufacturer’s instructions. Briefly, afer washing 5x105 wild type, Z-VAD-FMK-treated or Garcinol-treated U87MG cells twice with PBS, and once with annexin V binding buffer, the cells were incubated with PE-labeled annexin V and  7-AAD at room temperature for 15 min and then analyzed using flow cytometry. The mitochondrial transmembrane potential (DΨm) was evaluated using the cationic dye JC-1 (Mitochondrial Membrane Potential Assay Kit, #ab113850, Abcam plc., Cambridge, UK) in accordance with the manufacturer’s instructions (BD Pharmingen); 1x106 U87MG cells were incubated with 10 g/mL JC-1 at 37 °C in the dark for 15 min, and then analyzed using flow cytometry. All samples were assayed three times in triplicate.

Q8: Reviewer #4: The rationale to look at miR181-d as the link between garcinol, STAT3/5 phosphorylation remains obscure- given the fact that the expression of total STAT3  and STAT5 seem to remain unchanged, one would expect to look into a direct effect of garcinol on kinases (btw, ERK and Akt also vary and are MAJOR drivers of GBM proliferation and aopotosis resistance…)- Why on earth look into a miRNA, and how can the authors explain that the miRNA only targets the phosphorylation of STAT3/5, not the protein? Is the expression/editing of miR181-d under control of STAT3/5 itself? Can the authors provide time-course experiemnts to see if the control of mir181-d precedes the alteration of STAT3/5 phosphorylation? The discussion section should discuss the results and the mechanisms, not just speculate on unsupported conclusions- in particular, the statement that “Of translational relevance, we observed no significant difference in OS time between the STAT3highSTAT5Alow cohort and the STAT3lowSTAT5Ahigh cohort, suggesting there is no specific advantage in inhibiting the expression of either transcription factor, and highlighting probable therapeutic efficacy of parallel inhibition of both STAT3 and STAT5A. Thus, as indicated by the dismal OS of patients with STAT3highSTAT5high compared to the other groups (Figure 1), we, for the first time to the best of our knowledge demonstrate that concerted targeting of the aberrant expression and/or activity of STAT3/5A in GBM, both at protein and mRNA levels, is essential for any meaningful curative effect in STAT-based targeted therapy for patients with primary or even recurrent GBM.” Is totally unsupported by the results and is purely speculative.

A8:  We thank the reviewer for this comment. As suggested by the reviewer, we have now included additional data, including for a known upstream modulator of STAT activation in our revised manuscript. Please kindly see our revised Results section, lines 436-477.

3.4. Garcinol increases the expression of hsa-miR181d, which has inhibitory effects on STAT3 and STAT5 expression

Having established that garcinol impairs STAT3 and STAT5A activation, we probed for likely modulators and/or mediators of the interaction between garcinol and the STAT proteins. Hsa-miR-181d shown in Figure 4A, has been implicated in the worse OS of patients with GBM (Zhang et al., 2012). Consistent with this, using the Schrodinger PyMOL 2.3 molecular docking and visualization software (http://pymol.org) we demonstrated that hsa-miR-181d interacts with and binds directly to STAT3 (docking score = -254.49, ligand root mean square deviation (RMSD) = 195.10 Å) or STAT5A (docking score = -234.19, ligand RMSD = 143.04 Å), complementing earlier prediction that hsa-miR-181d binds with STAT3 with a mirSVR or PhastCons score of -0.26 or 0.69, respectively, while it binds with STAT5A with a mirSVR  or PhastCons score of -0.21 or 0.49, respectively (Figure 4A).  Where the mirSVR shows the likelihood of hsa-miR-181d down-regulating the target mRNA STAT3 and STAT5A based on the sequence and structure features in the miRNA/mRNA predicted target sites. Moreover, the PhastCons score shows the likelihood that the predicted miRNA/mRNA binding nucleotides are conserved. In concordance, our qRT-PCR analysis of 5 μM garcinol-treated U87MG and GBM8401 cells showed that garcinol significantly induced higher expression of miR-181d in the U87MG (2.7-fold, p < 0.01) and GBM8401 (2.1-fold, p < 0.01) cells (Figure 4B). Furthermore, having implicated STAT3/5A in enhanced migration and invasiveness of U87MG or GBM8401 cells, we examined the probable effect of hsa-miR-181d on these metastatic phenotypes of GBM cells. Using the scratch wound-healing assay, we demonstrated that compared to the untreated control or syn-mir-treated cells, treatment with mir-181d inhibitor significantly enhanced the ability of the U87MG cells to migrate (4.64-fold, p<0.01), while treatment with the mir-181d-mimic elicited marked attenuation of migration (3.80-fold, p<0.01) (Figure 4C), which is reminiscent of suppressed migration induced by 5 M garcinol (4.17-fold, p<0.01) earlier.  Similarly, while treatment with mir-181d inhibitor significantly enhanced the invasiveness of the U87MG cells (4.63-fold, p<0.01), treatment with the mir-181d-mimic elicited profound suppression of invasion (4.15-fold, p<0.01) (Figures 4D), and this was akin to the effect of 5 M garcinol (4.22-fold, p<0.01). To confirm a direct relationship between the STAT proteins and miR-181d expression, western blot analysis was done comparing samples exposed to mir-181d inhibitor, mir-181d-mimic, or mir-181d inhibitor/garcinol combination. The results showed that mir-181d inhibitor significantly enhanced the expression of p-STAT3, p-STAT5, N-cadherin and vimentin proteins, but suppressed E-cadherin protein expression compared to the control group, while for the mir-181d-mimic-treated cells, the p-STAT3, p-STAT5, N-cadherin, and vimentin protein expression levels were significantly lower, but E-cadherin was upregulated; For cells incubated with mir-181d inhibitor and 5 μM garcinol concomitantly, p-STAT3 and p-STAT5 protein expression levels were markedly higher than in the mir-181d-mimic group but lower than the mir-181d-inhibitor group (Figure 4E). Concomitantly, we observed that compared with the control group, syn-mir-treated cells, or even mir-181d inhibitor-treated cells, treatment with mir-181d-mimic markedly suppressed the expression of JAK2 protein (2.22-fold, p<0.01), akin to the effect elicited by treatment with concurrently with mir-181d inhibitor and 5 M garcinol (2.94-fold, p<0.01), which is consistent with the results above and suggestive of a miR-181d-mediated JAK2-modulated phosphorylation of STAT3/5A. These results indicate that garcinol can activate mir-181d activity which suppresses STAT3/5A activation.

See also our updated Figure 4 and legend, lines 479-491.

Figure 4. Garcinol increases the expression of hsa-miR181d, which has inhibitory effects on STAT3 and STAT5 expression. (A) 3-dimensional visualization of direct interaction between hsa-miR-181d and STAT3 (left) or STAT5A (middle) and images showing the complementary sequence alignment of hsa-miR-181d with STAT3 (upper right) or STAT5A (lower right). The mirSVR and PhastCons scores are indicated. (B) Histograms of the effect of 5 μM on hsa-miR-181d expression in U87MG or GBM8401 cells. (C) Representative images (left) and histograms (right) showing the effect of Syn-mir-CTL, mir-181d-inhibitor, or mir-181d-mimic on the migration of U87MG cells over 24 h, as determined by wound-healing assay. (D) Representative images (left) and histograms (right) showing the effect of Syn-mir-CTL, mir-181d-inhibitor, or mir-181d-mimic on the invasion of U87MG cells. (E) Representative western-blot photo-images (left) and histograms (right) comparing the effect of Syn-mir-CTL, mir-181d-inhibitor, mir-181d-mimic, or garcinol on the expression level of JAK2, p-STAT3, STAT3, p-STAT5, STAT5, E-cadherin, N-cadherin, and vimentin proteins in U87MG or GBM8401 cells. *p<0.05, **p<0.01, ***p<0.001; GAPDH is loading control.

Also kindly see our revised Discussion section, lines 576-598.

This study also demonstrated for the first time to the best of our knowledge, that garcinol-induced suppression of STAT3 and STAT5A is associated with significant upregulation of hsa-miR-181d expression, in vitro and in vivo; interestingly we also showed direct interaction between hsa-miR-181d and STAT3 or STAT5A protein. We posit that upon treatment with garcinol, miR-181d binds directly to the coding region of STAT3 or STAT5A mRNA, eliciting STAT3/5A degradation, and consequently reduce the expression and/or activation of STAT3 or STAT5A in the GBM cells. This demonstrated tumor suppressor role of hsa-miR-181d is consistent with findings showing that overexpression of miR-181d significantly suppressed esophageal squamous cell carcinoma (ESCC) by downredulating Derlin-1, inhibiting cancerous cell proliferation, migration and cell cycle progression in vitro, as well as inhibiting tumorigenicity in vivo (Li D et al., 2016), as well as in glioma samples and cell lines, where ectopic expression of miR-181d suppressed proliferation and induced cell cycle arrest and apoptosis by targeting K-ras and Bcl-2 (Wang et al., 2012). This demonstrated garcinol-modulated miR-181d/STAT3/5A signaling axis is of therapeutic relevance, considering that well documented role of the JAK-STAT signaling pathway, and more particularly its molecular effectors namely STAT3 and STAT5 which act as a point of convergence for several signaling pathways in cancerous cells and oncogenic processes (Luo and Balko, 2019). It is worth mentioning however, that while we cannot fully explain how garcinol induced hsa-miR-181d inhibited the activation of STAT3 and STAT5, our data finds some corroboration in increasingly documented role of miRs in the (de)activation of the JAK-STAT signaling (Zhuang G et al., 2012, Lam et al, 2013, Liu X et al., 2018), and of particular interest is miR-204 which similarly had very insignificant effect on total STAT3 expression, but impaired STAT3 phosphorylation, consequently inducing cancerous cell apoptosis and suppressed cell proliferation, migration in vitro and tumor growth in vivo (Liu X et al., 2018)

Q9: Reviewer #4: The title of the article is misleading as the authors have found a link between the phosphorylation of STAT3/5 and miR181-d, not STAT3/5 themselves- also, the ‘anticancer’ effect of garcinol is only evidenced in vitro.

A9:  We thank the reviewer for this comment. We believe with the added data; we have allayed the reviewer’s concern regarding the title and in vivo studies. Please see our updated Figure 5.

Q10: Reviewer #4: GBM that are re-operated tend in general to live longer than those that are not re-operated- however, in their cohort, the authors find that the recurrent tumors still show a major activation of STAT3 and 5- how can they treconcile this with the survival data? (low stat = longer survival) in their results? Please discus; Figure 1: please define the Q scores? what were the numbers of patients in the FFPE cohort (primary vs recurrent)? were there paired samples from the same patients? if yes, where there any differences in the expression of P-STAT3/5 between primary and recurrent tumors?.

A10:  We thank the reviewer for this important comment. We have made efforts to address the reviewer’s comments in our revised manuscript. While we are not sure we understand the first part of the comment, we have defined Q scores. Please see our revised Materials and Methods, Pages 4-5, Lines 189-205.

2.6. Immunohistochemical (IHC) Staining

The study was approved by the Joint Institutional Review Board (JIRB) of the Taipei Medical University –Shuang Ho Hospital (Approval number: N201903047). Tissue samples from patients with primary and recurrent GBM were obtained from the Taipei Medical University-Shuang Ho Hospital GBM cohort (n = 45). After de-waxing the paraffin-embedded 4 μm tissue sections using xylene for 5 min twice and re-hydrating with 100% ethanol twice for 5 min, 95% ethanol for 5 min and 80% ethanol for 5 min, 3% hydrogen peroxide (H2O2) (TA-125-H2O2Q, Thermo Fisher Scientific, Waltham, MA, USA) was used to block endogenous peroxidase activity for 10 min. The sections were then immersed in 10 mmol/L ethylenediaminetetraacetic acid (EDTA) for 3 min in a pressure cooker, then blocked with 10% normal serum. Thereafter, tissue samples were incubated with primary antibody against STAT3 (1:200) or STAT5 (1:200) at 4oC overnight, and then with biotin-labeled secondary antibody at room temperature for 1h. Sections were incubated in diaminobenzidine (DAB) and then counterstained with hematoxylin. Visualization was done under a light microscope. For staining determination, we used the Quick (Q) score formula: Q = [percentage of stained/positive cells] x [intensity of staining] The maximum score for percentage/distribution of stained/positive cells was 100, while the intensity of staining was delineated into weak (1), moderate (2), or strong (3), making the obtainable maximum Q score = 300.

Please kindly see our updated Figure 1 and its legend, Page 8, Lines 336-344.

Figure 1. STAT-3 and STAT-5 are highly expressed in primary and recurrent glioblastoma, and their expression negatively correlates with overall survival rates. (A) Graphical representation of the differential expression of STAT3 and STAT5A in primary GBM, recurrent GBM or normal brain tissues from the GDC TCGA-GBM cohort. (B) Kaplan-Meier plots of the effect of differential STAT3 or STAT5A expression on OS of patients with GBM. (C) Representative IHC images showing the differential expression of STAT3, pSTAT3, STAT5A, and pSTAT5A proteins in primary GBM, recurrent GBM or normal brain tissues. (D) Graphical representation of the differential expression of STAT3, pSTAT3, STAT5A, pSTAT5A proteins in primary GBM, recurrent GBM or normal brain tissues. *p<0.05, **p<0.01, ***p<0.001; OS, overall survival

Q11: Reviewer #4: Paragraph 3.2: the authors conclude that there was an increased apoptosis following garcinol treatment, but do not show direct evidence of this mechanism, only a change in the expression ratios of pro vs anti-apoptotic Bcl family proteins- A TUNEL or FACS analysis of apoptosis is needed her to support the statement;

A11:  We thank the reviewer for this comment. As suggested by the reviewer, we have now included additional data in our revised manuscript. Please kindly see our revised Results section, lines 345-381.

3.2. Garcinol significantly inhibits GBM cell viability and oncogenicity through induction of STAT3/5A signaling and enhanced apoptosis

Against the background of recent work demonstrating that garcinol inhibits CSC-like phenotype of human non-small cell lung carcinoma by suppressing the Wnt/β-catenin/STAT3 signaling axis (Huang et al., 2018), we investigated the probable STAT signaling-mediated anti-GBM effect of garcinol (Figure 2A). Firstly, to provide some mechanistic insight, we demonstrated that treatment of U87MG or GBM8401 cells with 2.5 μM or 5 μM garcinol significantly downregulated the expression of p-STAT3, p-STAT5, p-ERK, and p-AKT (Figure 2B), Synchronous with the observed inhibition of STAT3, STAT5 and AKT signaling, garcinol significantly suppressed the viability of GBM4801 and U87MG cells, with 10 μM eliciting 51% or 25% reduced viability of U87MG or GBM8401 cells, respectively, and 40 μM eliciting  94.7% reduction of U87MG and GBM8401 cell viability, indicating a dose-dependent GBM cell killing effect (Figure 2C), and this reduced viability was associated with markedly enhanced  Bax/Bcl-xL apoptotic ratio, as 2.5 μM induced a 1.67-fold (p < 0.05) or  2.7-fold (p < 0.05) increase in U87MG or GBM8401 apoptotic ratio, while  5 μM increased the apoptotic ratio by 2.83-fold (p < 0.001) or 2.92-fold (p < 0.001) in the U87MG or GBM8401 cells, respectively (Figure 2D). In addition, using the Phycoerythrin (PE)-conjugated Annexin V/7- Amino-Actinomycin (7-AAD) staining, we demonstrated that compared to the cell death in the untreated control (U87MG: 1.6%, GBM8401: 4.0%) or 20 M pan-caspase inhibitor benzyloxycarbonyl-Val-Ala-Asp-fluoromethyl ketone (Z-VAD-FMK)-treated negative control (U87MG: 3.2%, GBM8401: 10.0%), treatment with Garcinol enhanced U87MG cell death (2.5 M:  10.5% or 5 M: 41.2%), while 2.5 M or 5 M garcinol elicited 15.1% or 32.1% apoptosis of the GBM8401 cells, respectively (Figure 2E), indicating that the Garcinol-induced cell death was apoptotic and caspase-dependent. Since the highly invasive GBM spreads fast to surrounding brain tissue, thus, contributing to its documented lethality (Omuro & DeAngelis, 2013), we sought to understand if and how garcinol affects this invasive trait. We demonstrated that treatment with 2.5 μM or 5 μM dose-dependently suppressed the migration of the U87MG (~59%, p < 0.01 or 81%, p < 0.001, respectively) and GBM8401 (~48%, p < 0.01 or 76%, p < 0.001, respectively) cells at the 24 h time-point (Figure 2F). Similarly, 2.5 μM or 5 μM garcinol induced a 60% (p < 0.01) or ~80% (p < 0.001) reduction of U87MG invasive capacity, respectively, and 39% (p < 0.01) or 60% (p < 0.001) reduction in number of invaded GBM8401 cells (Figure 2G). Furthermore, in parallel assays to confirm the anticancer role of garcinol, consistent with earlier results, we demonstrated that treatment with 2.5 M or 5 M garcinol, significantly suppressed the expression of N-cadherin, vimentin and slug proteins, while conversely upregulating the expression of E-cadherin protein (Figure 2H), thus indicating that garcinol attenuates epithelial-to-mesenchymal transition (EMT) and the metastatic phenotype of GBM cells. Together, these data suggest that garcinol significantly inhibits GBM cell viability and oncogenicity through induction of STAT3/5A and associated signaling with enhanced apoptosis.

Please kindly see our updated Figure 2 and its legend, lines 383-395.

Figure 2. Garcinol significantly inhibits GBM cell viability and oncogenicity through induction of STAT3/5A signaling and enhanced apoptosis. (A) Chemical structure of garcinol with molecular formula C38H50O6 and molecular weight 602.80 g/mol. (B) Representative western blot photo-images of the effect of 2.5 μM – 5 μM on the expression of p-STAT3, STAT3, p-STAT5, STAT5, p-ERK, ERK, p-AKT, and AKT proteins in GBM8401 or U87MG cells. (C) Graphical representation of the effect of 2.5 μM – 40 μM on the viability of GBM8401 or U87MG cells. (D) Representative western-blot photo-images showing the effect of 2.5 μM – 5 μM on the expression of Bax and Bcl-xL proteins in GBM8401 or U87MG cells. (E) Flow-cytometry data (upper) and graphical representation (lower) showing the effect of Garcinol on U87MG cells co-stained with PE-conjugated Annexin V and 7- AAD, compared with untreated or Z-VAD-FMK -treated negative control groups. Annexin V-stained Q4 cells are early apoptotic cells, whereas Q2 cells are late stage apoptotic (necrotic) cells. Apoptosis (%), sum of Q4+Q2; CTL, vehicle-treated; Neg CTL, pan-caspase inhibitor benzyloxycarbonyl-Val-Ala-Asp-fluoromethyl ketone (Z-VAD-FMK). Representative photo-images (upper) and graphical representation (lower) of the effect of 2.5 μM or 5 μM on the (F) migration and (G) invasion of GBM8401 or U87MG cells. (H) Representative western-blot photo-images showing the effect of 2.5 μM – 5 μM on the expression of E-cadherin, N-cadherin, vimentin and slug proteins in GBM8401 or U87MG cells. *p<0.05, **p<0.01, ***p<0.001; GAPDH is loading control.

Also kindly see our revised Materials and Methods section, lines 266-276.

2.13. PE-Annexin V/7-AAD cell death assay

PE-Annexin V/7-AAD staining was used for detection of cell death using the BD FACSCanto™ II flow cytometry system (BD Biosciences, San Jose, CA, USA) following the manufacturer’s instructions. Briefly, afer washing 5x105 wild type, Z-VAD-FMK-treated or Garcinol-treated U87MG cells twice with PBS, and once with annexin V binding buffer, the cells were incubated with PE-labeled annexin V and  7-AAD at room temperature for 15 min and then analyzed using flow cytometry. The mitochondrial transmembrane potential (DΨm) was evaluated using the cationic dye JC-1 (Mitochondrial Membrane Potential Assay Kit, #ab113850, Abcam plc., Cambridge, UK) in accordance with the manufacturer’s instructions (BD Pharmingen); 1x106 U87MG cells were incubated with 10 g/mL JC-1 at 37 °C in the dark for 15 min, and then analyzed using flow cytometry. All samples were assayed three times in triplicate.

Q12: Reviewer #4: While the decreases in invasion are proportionally bigger than the differences in survival at the same garcinol dosage, one cannot rule out that a significant proportion of the differences in invasion are due to a reduced cell proliferation- the crossing cells should be normalized with respect to the non-crossing cells at the end of the experiments. Scratch assays: please provide better pictures and discuss the role of proliferation as well

A12:  We thank the reviewer for these comments. For our colony formation assay, we have used well established and documented protocol. In our revised manuscript, we have provided more representative, high resolution figures. For the Scratch wound-healing assay, please kindly see our updated Figure 2F.

Q13: Reviewer #4: Figure 3C: the spheres do not seem to have the same size- please choose spheres of similar size or improve the picture in order to better see the size of the sphres (add phase contrast pictures)- indeed, spheres of variable size may have different expression of stem cell markers; 

A13:  We sincerely thank the reviewer for this comment. As requested by the reviewer, we have provided more representative, high resolution figures. please kindly see our updated Figure 3C.
